# LUNGUAGE: A BENCHMARK FOR STRUCTURED AND SEQUENTIAL CHEST X-RAY INTERPRETATION

## ABSTRACT

Radiology reports convey detailed clinical observations and capture diagnostic reasoning that evolves over time. However, existing evaluation methods are limited to single-report settings and rely on coarse metrics that fail to capture fine-grained clinical semantics and temporal dependencies. We introduce LUNGUAGE, a benchmark dataset for structured radiology report generation that supports both single-report evaluation and longitudinal patient-level assessment across multiple studies. It contains 1,473 annotated chest X-ray reports, each reviewed by experts, and 186 of them contain longitudinal annotations to capture disease progression and inter-study intervals, also reviewed by experts. Using this benchmark, we develop a two-stage structuring framework that transforms generated reports into fine-grained, schema-aligned structured reports, enabling longitudinal interpretation. We also propose LUNGUAGESCORE, an interpretable metric that compares structured outputs at the entity, relation, and attribute level while modeling temporal consistency across patient timelines. These contributions establish the first benchmark dataset, structuring framework, and evaluation metric for sequential radiology reporting, with empirical results demonstrating that LUNGUAGESCORE effectively supports structured report evaluation. Code and data are available at:
`https://anonymous.4open.science/r/lunguage`

## 1 INTRODUCTION

Radiology reports play a critical role in diagnosis by recording patient history, describing imaging findings, documenting procedures, and noting temporal changes. However, because they are written in unstructured free text, reports vary widely in terminology, style, and level of detail across radiologists, complicating consistent computational interpretation and hindering automated systems for report generation and evaluation. To address these challenges, structuring frameworks have been developed to convert free-text reports into standardized, machine-friendly formats (Jain et al. (2021); Khanna et al. (2023); Wu et al. (2021); Zhang et al. (2023); Zhao et al. (2024)). While these frameworks improve representational consistency, current evaluation methods remain fundamentally limited in two key aspects: temporal reasoning and fine-grained clinical accuracy.

Temporal reasoning is central to radiologic interpretation, as diagnoses often depend on comparing current and prior studies to assess whether a finding has progressed. However, most evaluation protocols (Bannur et al. (2024); Huang et al. (2024); Jain et al. (2021); Khanna et al. (2023); Ostmeier et al. (2024); Smit et al. (2020); Wu et al. (2021); Yu et al. (2023a); Zhang et al. (2023); Zhao et al. (2024)) assess reports in isolation, without incorporating previous findings. This makes it impossible to determine whether temporal expressions—such as "no change," "improved," or "new"—are appropriate. For instance, the statement "no change in pneumonia" cannot be meaningfully evaluated without confirming whether pneumonia was present in prior studies.

Fine-grained clinical accuracy is equally critical. Reliable interpretation depends on attributes such as precise location (e.g., "carina above 3 cm") and lesion size (e.g., "2.5 cm"). These details are essential for diagnostic specificity and downstream decision-making, yet most evaluation protocols collapse them into broad categories. For instance, "2.5 cm right upper lobe nodule with spiculated margins" may be reduced to simply "nodule," and this loss of granularity makes it difficult to distinguish precise from incomplete outputs.

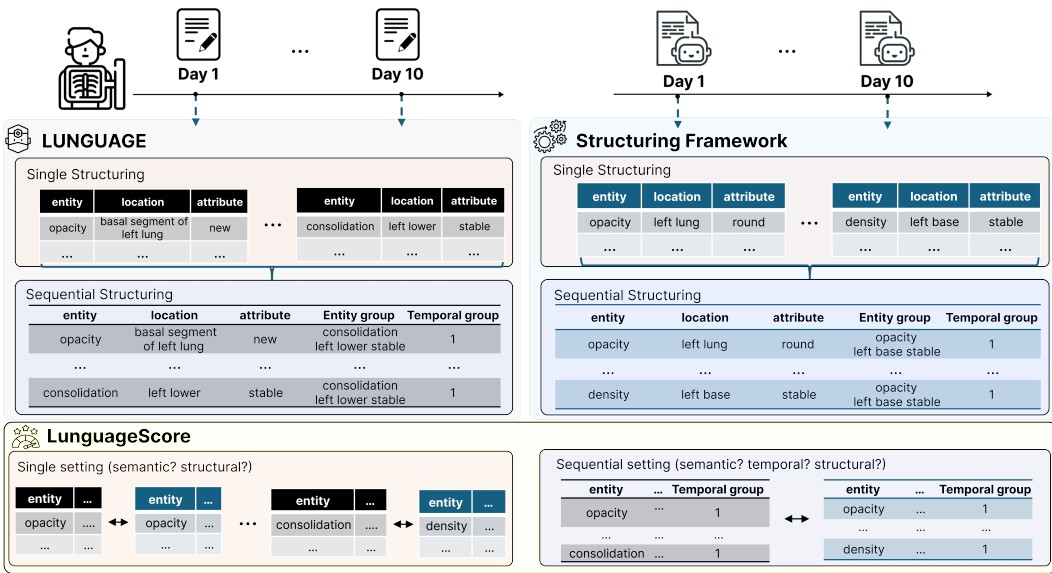

Figure 1: **Evaluation pipeline for radiology report generation.** We introduce the first evaluation framework for radiology report generation, enabling both detailed single-report assessment and comprehensive patient-level trajectory evaluation. On the left, we release LUNGUAGE, a radiologist-annotated benchmark of structured single and sequential chest X-ray reports. On the right, we develop a two-stage **structuring framework** that converts free-text into schema-aligned structures at both single and sequential levels. At the bottom, we present LUNGUAGESCORE, a clinically validated metric that jointly measures semantic accuracy, structural fidelity, and temporal alignment, providing clinically faithful evaluation.

Structuring frameworks have attempted to address these issues by extracting entities and relations from reports (Jain et al. (2021); Khanna et al. (2023); Wu et al. (2021); Zhang et al. (2023); Zhao et al. (2024)). Some extend this by tagging temporal descriptors such as "worsened" or "stable" (Khanna et al. (2023); Wu et al. (2021)). Yet, they remain restricted to single reports and rely only on explicitly stated expressions, without verifying consistency across time. Consequently, they cannot ensure whether findings align with prior studies or capture coherent clinical trajectories, and often miss the clinical granularity needed for precise diagnostic interpretation.

Recent report generation models have begun incorporating temporal inputs such as prior reports, imaging, or clinical indications (Bannur et al. (2024); Zhou et al. (2024)), enabling outputs that are more context-aware and temporally coherent. However, evaluation protocols have not kept pace. Generated reports are still judged at isolated timepoints rather than across a continuous timeline, making it impossible to assess whether models appropriately incorporated prior findings or preserved clinically important details at both temporal and semantic levels.

To address these limitations, we present the first evaluation pipeline for assessing radiology report generation in both single and sequential settings. Our contributions are threefold. **(1)** We construct LUNGUAGE, a fine-grained benchmark that establishes reliable ground truth for evaluation. It consists of **1,473 single reports** from 230 patients (annotated with 17,949 entities and 23,307 relation–attribute pairs across 18 clinically grounded relation types) and **186 sequential reports** from 30 patients (95,404 observation pairs across 2–14 reports per patient). These support longitudinal analysis through ENTITYGROUPS (linking the same finding across reports) and TEMPORALGROUPS (segmenting diagnostic episodes). **(2)** To enable automatic benchmarking on this scale, we develop a **structuring framework** that converts free text into *entity–relation–attribute* triplets and links them across time following the LUNGUAGE schema. It achieves high agreement with expert annotations (F1: 0.94 for entity–relation, 0.86 for full triplets, 0.69 for ENTITYGROUP, 0.87 for TEMPORALGROUP). **(3)** Building on this foundation, we introduce LUNGUAGESCORE, a clinically grounded metric that compares structured representations from generated and reference reports. Unlike prior approaches, this metric simultaneously captures semantic accuracy, structural fidelity, and temporal coherence. To our best knowledge, this is the first study to combine the highest schema granularity with explicit modeling of full diagnostic trajectories.

## 2 RELATED WORK

**Structuring Radiology Reports**   Radiology reports encode layered clinical semantics, spanning history, imaging observations, and diagnostic reasoning. Rule-based systems (Wu et al. (2021); Zhang et al. (2023)) achieve high precision in constrained settings but struggle to generalize due to linguistic variability. Supervised transformer-based methods (Jain et al. (2021); Khanna et al. (2023); Zhao et al. (2024)) are more flexible but depend heavily on the coverage and granularity of their annotation schema. Recently, prompting-based approaches have leveraged large language models (LLMs), such as GPT-4 (Achiam et al. (2023)) and open-source variants (Liu et al. (2024); Touvron et al. (2023)), to directly produce structured outputs from free text (Busch et al. (2024); Dorfner et al. (2024); Hartsock et al. (2025); Woźnicki et al. (2024)). While these models demonstrate strong few-shot performance, they remain prone to hallucinations, inconsistent terminology, and prompt sensitivity. To mitigate this, we employ a task-specific vocabulary and schema-aligned reference set, constraining outputs to valid clinical concepts and enhancing consistency through retrieval-augmented prompting. A detailed comparison is provided in Appendix A.4.

**Evaluation Metrics for Radiology Report Understanding**   Existing metrics fall into three categories: lexical, model-based, and structure-based. Lexical metrics (BLEU (Papineni et al. (2002)), ROUGE (Lin (2004)), METEOR (Banerjee & Lavie (2005))) rely on surface overlap and often miss clinical meaning. Model-based metrics (CheXbert (Smit et al. (2020)), BERTScore (Zhang et al. (2019))) capture semantic similarity but overlook fine-grained detail. Structure-based metrics (RadGraph-F1 (Jain et al. (2021)), RaTEScore (Zhao et al. (2024))) add granularity by matching entities and relations. Recent efforts emphasize clinical error detection: ReXVal (Yu et al. (2023b)) introduced expert-labeled errors, informing RadCliQ (Yu et al. (2023a)), which combines BERTScore and RadGraph-F1, while LLM-based metrics (GREEN (Ostmeier et al. (2024)), FineRadScore (Huang et al. (2024)), RadFact (Bannur et al. (2024)), CheXprompt (Zambrano Chaves et al. (2025))) aim to approximate expert judgment or factual correctness. However, most metrics still evaluate reports in isolation, overlooking temporal consistency across studies and neglecting attributes like location, extent, or progression. In contrast, our evaluation pipeline provides structured, temporally aligned evaluation over patient report sequences, enabling clinically faithful assessment across semantic, structural, and temporal dimensions.

## 3 LUNGUAGE: SINGLE AND SEQUENTIAL STRUCTURED REPORTS

Radiology reports vary in depth and nuance, with differences in phrasing, certainty, and cross-sentence connections that make structured interpretation challenging. They are commonly divided into *indication/history*, which provides contextual cues (e.g., "history of cough"), and *findings* and *impression*, which contain detailed descriptions and diagnostic reasoning (e.g., "left opacities likely consolidation or pneumonia"). To address this complexity, we present LUNGUAGE, a benchmark dataset of radiologist-annotated chest X-ray reports in two complementary versions: 1,473 single reports and 186 sequential reports. Reports were structured through a rigorous annotation process (Appendix A.3) guided by three principles: diagnostic source distinction (separating image-based from context-based findings), semantic precision (capturing descriptive cues such as certainty, status, and other fine-grained attributes including location, severity, and morphology), and longitudinal linkage (capturing temporal consistency through entity and temporal grouping). This design reflects physicians' clinical perspectives and supports both a single-report schema for fine-grained interpretation and a sequential schema for modeling patient-level diagnostic trajectories. Figure 2 illustrates these schemas.

### 3.1 SINGLE STRUCTURED REPORT: FINE-GRAINED SCHEMA AND ANNOTATION

We propose a single-report schema that captures the internal structure of radiology reports by structuring clinically relevant information into two units: **entities**, representing core clinical concepts, and **relations**, encoding their attributes and interconnections, enabling systematic modeling of both detailed descriptions and cross-sentence reasoning for fine-grained and clinically faithful interpretation.

**ENTITIES** are assigned to one of six clinically grounded categories based on their derivability from chest X-ray imaging: PF (PERCEPTUAL FINDINGS) for directly observable image features (e.g.,

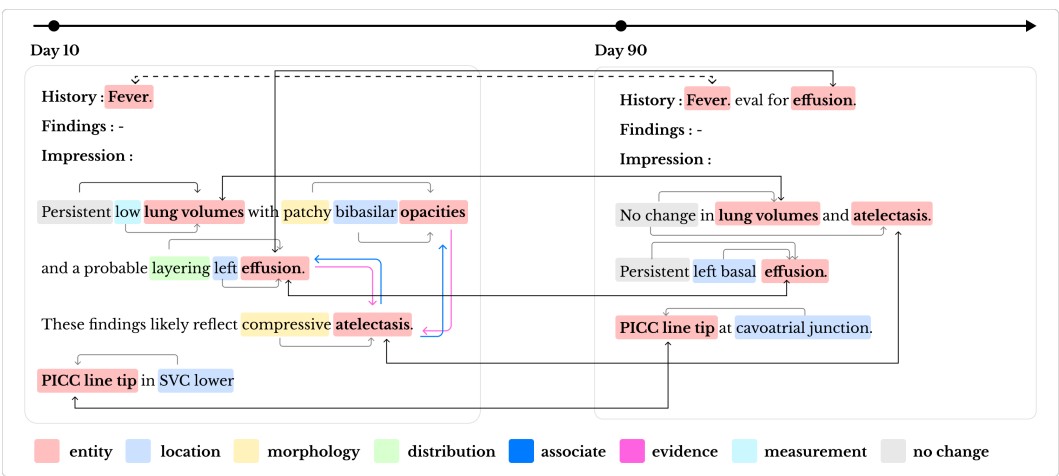

Figure 2: **Schema for Single and Sequential Report Structuring.** The figure shows two reports from the same patient at day 10 and day 90. For the single report schema (within each report), gray solid lines connect entities to attributes, while pink and blue solid lines represent inter-entity reasoning relations (ASSOCIATE, EVIDENCE). For the sequential schema (across reports), black solid lines denote entities in the same ENTITYGROUP (same clinical finding over time) and TEMPORALGROUP (same diagnostic episodes), while black dashed lines show entities in the same ENTITYGROUP but different TEMPORALGROUPS (different diagnostic episodes).

"lung," "opacity"); CF (CONTEXTUAL FINDINGS) for diagnoses inferred from external clinical context (e.g., "pneumonia"); OTH (OTHER OBJECTS) for mentioned devices or procedures (e.g., "ET tube"); COF (CLINICAL OBJECTIVE FINDINGS) for structured observations from non-imaging sources (e.g., lab tests); NCD (NON-CXR DIAGNOSIS) for diagnoses based on other modalities (e.g., "AIDS"); and PATIENT INFO for reported history or symptoms (e.g., "fever," "cough").

**RELATIONS** capture clinical properties and inter-entity connections, often spanning multiple sentences. The schema includes diagnostic stance (DXSTATUS, DXCERTAINTY); spatial and descriptive characteristics (LOCATION, MORPHOLOGY, DISTRIBUTION, MEASUREMENT, SEVERITY, COMPARISON); temporal dynamics (ONSET, IMPROVED, WORSENED, NOCHANGE, PLACEMENT); and contextual information (PASTHX, OTHERSOURCE, ASSESSMENTLIMITATIONS)[1]. It also includes two reasoning relations: ASSOCIATE (bidirectional links between related entities) and EVIDENCE (asymmetric support from a finding to a diagnosis). For example, in "left lung opacity suggests pneumonia," the schema identifies both ASSOCIATE between *opacity* and *pneumonia*, and EVIDENCE indicating that *pneumonia* is inferred from *opacity*. Full definitions can be found in Appendix A.1.

**Single Report Annotation Process**   We developed a two-stage annotation pipeline for 1,473 radiology reports from MIMIC-CXR (Johnson et al. (2019)) to ensure fine-grained and clinically grounded structuring of report language. The process began with schema design and initial structured drafts generated by GPT-4 (0613)[2], from which all candidate entity and relation terms were collected to build a comprehensive vocabulary. Four radiologists independently reviewed these terms in a blinded manner, resolving discrepancies through majority voting and consensus meetings, and referring to the Fleischner Society (Bankier et al. (2024)) terminology and UMLS (Bodenreider (2004)) mappings where appropriate. This stage unified terminology and eliminated category-level inconsistencies in advance. In the second stage, annotators revised all reports using this curated vocabulary, focusing on contextual interpretation and correction of potential LLM errors rather than category disputes. Reports were evenly divided among radiologists, who verified every *(entity, relation, attribute)* triplet, including cross-sentence relations such as ASSOCIATE and EVIDENCE. This two-step process yielded 17,949 entity instances and 23,307 relation instances, providing a

---

[1]**Abbreviations:** "Dx" stands for "diagnosis" and is used in relations such as DXSTATUS (i.e., positive or negative finding) and DXCERTAINTY (i.e., definitive or tentative). "Hx" in PASTHX stands for "history".

[2]All large language model (LLM) usage, including GPT-4, was conducted using HIPAA-compliant deployments provided by Azure and Fireworks AI.

reliable and clinically validated dataset for benchmarking structured report interpretation. Details of the vocabulary and annotation process are provided in Appendix A.2 and A.3.1.

## 3.2 SEQUENTIAL STRUCTURED REPORT: DISEASE TRAJECTORY SCHEMA AND ANNOTATION

Longitudinal reports often exhibit lexical variation, abstraction shifts, and inconsistent phrasing (Meystre et al. (2008); Wang et al. (2018)). The same pathology may be described differently across timepoints, such as "left opacity" and "left lower consolidation," with differences in wording and specificity that complicate semantic alignment and temporal reasoning. To address this, we introduce a schema that structures reports across patient timelines through two key components:

**ENTITYGROUPS** identify observations that refer to the same underlying clinical finding, even when expressed using different terms, anatomical references, or levels of abstraction. Within each patient, all observation pairs are compared to detect semantic equivalence, regardless of when they appear in the timeline, whether the finding is reported as present or absent (DXSTATUS), or whether it is stated definitively or tentatively (DXCERTAINTY). For example, "PICC line tip in lower SVC" and "at the cavoatrial junction" (Figure 2) may describe the same catheter tip location, reflecting inherent ambiguity in 2D imaging. Similarly, "lung volumes" reported as low on day 10 and described as "no change" on day 90 can be grouped to indicate persistent low lung volume.

**TEMPORALGROUPS** divide each ENTITYGROUP into distinct diagnostic episodes based on temporal distance, status shifts or certainty, and clinical change expressions (e.g., "worsening"). This approach captures clinically meaningful transitions in a patient's condition (Chapman et al. (2011); Savova et al. (2010)). For example, "fever" mentioned in both the day 10 and day 90 reports (Figure 2) appears in the "history" section but occurs far apart in time, so treating them as separate temporal groups better reflects clinical reasoning, whereas repeated descriptions of an effusion would remain in the same group. Together, these components support fine-grained evaluation of both semantic consistency and temporal coherence in longitudinal model outputs.

**Sequential Report Annotation Process** We annotated 186 chest X-ray reports from 30 patients—a subset of the 230-patient cohort used in single-report annotation—to construct a gold-standard dataset for patient-level longitudinal evaluation. The same four radiologists independently reviewed reports in chronological order, linking observations that referred to the same underlying finding into ENTITY-GROUPS (e.g., "pleural effusion right lung increasing") and dividing them into TEMPORALGROUPS (labeled 1, 2, . . . ) to distinguish diagnostic episodes. Terminology was normalized when appropriate (e.g., aligning "clavicle hardware" with "orthopedic side plate") while preserving abstraction and anatomical distinctions. Patients contributed 2–14 reports, with intervals spanning 1–1,200 days. For each patient, all observation pairs (29–141 per case) were compared, yielding 95,404 total comparisons. This rigorous process ensured both longitudinal consistency and clinically meaningful transitions such as resolution or recurrence. Further details are provided in Appendix A.3.2.

## 4 STRUCTURING FRAMEWORK FOR SINGLE AND SEQUENTIAL REPORTS

We develop a two-stage structuring framework that automatically structures radiology reports using the same schema as LUNGUAGE, generating radiologist-like structured outputs for consistent evaluation. The framework covers both single-report and longitudinal settings, producing representations for semantic, structural, and temporal evaluation (Figure 1).

**(i) Single Structuring Framework** To generate structured outputs from free text, we apply corpus-guided relation extraction with a LLM, which extracts *(entity, relation, attribute)* triplets aligned to our schema. The task requires handling both intra- and inter-sentential contexts and accommodating lexical variation without relying on templates. While LLMs can capture diverse phrasing and nuanced expressions, they are prone to hallucinations and inconsistencies (Busch et al. (2024); Dorfner et al. (2024); Hartsock et al. (2025); Woźnicki et al. (2024)). To mitigate errors, we guide the model with a curated vocabulary derived from our annotation corpus (Section 3.1). Details of the prompts and the vocabulary-matching algorithm are provided in Appendix B.1 and B.1.1.

**(ii) Sequential Structuring Framework** Building on the outputs from stage (i), we use the LLM to interpret report sequences over time. To address longitudinal variability, the model performs normalization and temporal aggregation. Each entity and its attributes are linearized into flattened text in chronological order relative to the initial study (e.g., "day 0: opacity right lung," "day 30: opacity

right basilar"). The LLM is guided with few-shot examples illustrating lexical variation, abstraction shifts (e.g., descriptive to diagnostic terms), and rephrasings of persistent devices. Using these, it determines whether observations across time represent the same underlying finding and whether they belong to a single temporal group. Decisions are guided by semantic similarity, anatomical alignment, and temporal continuity. Observations reflecting recurrence after resolution or clinically disconnected events are treated as distinct temporal groups. This process yields two outputs: ENTITY GROUPS and TEMPORAL GROUPS, consistent with Section 3.2. The format combines entity, location, and temporal pattern (e.g., "pleural effusion right lung no change"), with groups numbered sequentially (1, 2, 3, . . . ). This framework enables faithful structuring of longitudinal narratives, capturing clinically meaningful trajectories across report sequences. Full prompt examples are provided in Appendix B.2.

## 5 LUNGUAGESCORE: A FINE-GRAINED PATIENT-LEVEL METRIC

We propose **LUNGUAGESCORE**, a fine-grained metric for quantifying radiology report quality across semantic equivalence, temporal coherence, and attribute-level similarity. It captures clinically meaningful distinctions in terminology (e.g., "right clavicle hardware" vs. "orthopedic side plate"), longitudinal trends (e.g., resolution vs. decrease), and detailed attributes (e.g., 2.3 cm vs. 3.0 cm). These dimensions are integrated into a single similarity score that compares candidate and reference reports—either individually or as sequences—enabling patient-level evaluation.

**Evaluation Principles.** LUNGUAGESCORE is grounded in three clinical principles: **semantic sensitivity** captures concept-level equivalence across linguistic variation (Meystre et al. (2008); Wang et al. (2018)); **temporal coherence** ensures alignment with clinical timelines for assessing disease progression (Chapman et al. (2011); Savova et al. (2010)); and **structural granularity** evaluates fine-grained attributes critical for diagnosis (Demner-Fushman et al. (2009); Pons et al. (2016)). These principles enable clinically faithful evaluation suitable for real-world deployment.

**Evaluation Method.** Each patient is associated with a sequence of $T$ structured reports. The metric operates at the patient level and supports both single-report ($T = 1$) and sequential-report ($T > 1$) evaluations. In the **single-report** setting, evaluation is based on semantic and structural alignment, while in the **sequential-report** setting, temporal alignment is additionally incorporated to assess consistency across longitudinal disease trajectories. Formally, LUNGUAGESCORE evaluates similarity between predicted and gold reference sets of structured report findings as follows.

For each patient, we compare all predicted and gold reference findings across the entire sequence of reports. Let $\mathcal{S}^{\text{pred}} = (S_1^{\text{pred}}, \ldots, S_T^{\text{pred}})$ and $\mathcal{S}^{\text{gold}} = (S_1^{\text{gold}}, \ldots, S_T^{\text{gold}})$ denote the predicted and gold sequences for a given patient, where each $S_t^{(\cdot)}$ is the set of all structured findings at the $t$-th study. Pairwise similarity is computed over every possible pair of findings, pooled across all timepoints:

$$(f^{\text{pred}}, f^{\text{gold}}) \in \left( \bigcup_{t_p=1}^{T} S_{t_p}^{\text{pred}} \right) \times \left( \bigcup_{t_g=1}^{T} S_{t_g}^{\text{gold}} \right). \tag{1}$$

Each pair of findings is assigned a composite similarity score that captures alignment across semantic, temporal, and structural similarity dimensions, as defined below:

$$\text{MatchScore}(f^{\text{pred}}, f^{\text{gold}}) = \text{Semantic} \cdot (\text{Temporal if } T > 1) \cdot \text{Structural}. \tag{2}$$

**Semantic similarity** determines whether two findings express the same underlying clinical concept. Representation differs by setting: in single reports ($T = 1$), each finding is encoded as a linearized phrase combining the entity and all attributes (e.g., "opacity"-"left lung"-"nodular"-"slightly increased"); in sequential reports ($T > 1$), where findings must be tracked across time, we instead use ENTITYGROUP (Section 4). This enables lexically divergent but conceptually identical findings to align across multiple reports. Semantic similarity is then computed as the average cosine similarity between contextual embeddings from two domain-specific clinical BERT models—MedCPT and BioLORD (Jin et al. (2023); Remy et al. (2024))—selected for their ability to capture variability in chest X-ray language. Details of model selection are in Appendix C.3.

$$\text{Semantic}(f^{\text{pred}}, f^{\text{gold}}) = \text{cosine}(\text{Embed}(f^{\text{pred}}), \text{Embed}(f^{\text{gold}})) \tag{3}$$

**Temporal similarity** is defined only when $T > 1$ and captures alignment across timepoints. It ensures that findings are not only semantically similar but also temporally coherent with the patient's disease progression. To prevent matches across unrelated timepoints, LUNGUAGESCORE prioritizes findings that occur in the same study timepoint $t$ and TEMPORALGROUP. Temporal alignment receives the maximum score (= 1) when both study timepoint $t$ and TEMPORALGROUP match, and a reduced score when only one matches, for example, when a predicted finding belongs to the correct TEMPORALGROUP but appears in a different study. Final scores are computed using equal weights:

$$\text{Temporal}(f^{\text{pred}}, f^{\text{gold}}) = w_S \cdot \mathbf{1}[\text{S}(f^{\text{pred}}) = \text{S}(f^{\text{gold}})] + w_G \cdot \mathbf{1}[\text{G}(f^{\text{pred}}) = \text{G}(f^{\text{gold}})]. \quad (4)$$

where S refers to the study timepoint $t$, G refers to the TEMPORALGROUP of findings across time, and equal weights ($w_S = w_G = 0.5$) are used in our implementation.

**Structural similarity** evaluates individual attributes (e.g. LOCATION, MEASUREMENT...) between predicted and gold reference findings, enabling fine-grained comparison. Each attribute is assigned a normalized weight $w_{\text{attribute}}$ based on its clinical importance, as determined by experts, reflecting its role in decision making (see Appendix C.1). Similarity is computed as:

$$\text{Structural}(f^{\text{pred}}, f^{\text{gold}}) = \sum_{\text{attribute}} w_{\text{attribute}} \cdot \text{sim}(f^{\text{pred}}[\text{attribute}], f^{\text{gold}}[\text{attribute}]), \quad (5)$$

where $\text{sim}(\cdot)$ returns 1 for exact matches on binary attributes[3] and cosine similarity for non-binary attributes[4] using the average of MedCPT and BioLORD contextual encoders. This ensures that evaluation captures both overall correctness and clinically critical attribute accuracy.

**Set-level matching with partial credit.** We can compute the combined MatchScore by multiplying semantic, temporal, and structural similarity scores (Equations 3-5), as shown in Equation 2. We then perform optimal bipartite matching between predicted findings $i$ and gold reference findings $j$ using MatchScore $s_{ij}$ as edge weights, giving us sets of matched pairs $\{(f_m^{(pred)}, f_n^{(gold)})\}$, unmatched predicted findings $\{f_u^{(pred)}\}$, and unmatched gold reference findings $\{f_v^{(gold)}\}$. Matched pairs contribute similarity $s_{mn}$ to true positives (TP), with residual $(1 - s_{mn})$ assigned to false positives (FP) and negatives (FN). Unmatched findings incur penalties based on their most similar finding:

$$\text{TP} = \sum_{(m,n)} s_{mn}, \text{FP} = \sum_{(m,n)} (1 - s_{mn}) + \sum_u \left(1 - \max_j s_{uj}\right), \text{FN} = \sum_{(m,n)} (1 - s_{mn}) + \sum_v \left(1 - \max_i s_{iv}\right). \quad (6)$$

This formulation supports **partial credit** based on alignment strength. Full credit is awarded only when a finding aligns simultaneously at the semantic, temporal, and structural levels. Partial matches contribute proportionally to the score, while unmatched findings in either set are penalized as FP or FN. This scoring scheme enables nuanced evaluation that distinguishes minor misalignments from complete misses. The final F1 score is computed from these TP, FP, and FN counts using the standard formula. Additional illustrative examples are provided in Appendix C.2.

## 6 EXPERIMENTS

We conduct three experiments from complementary perspectives: (1) performance of the structuring framework, (2) diagnostic utility of LUNGUAGESCORE as a single-report evaluation metric, and (3) benchmarking of single- and longitudinal-report generation models with LUNGUAGESCORE.

### 6.1 STRUCTURING FRAMEWORK VALIDATION

We evaluate the **structuring framework** on LUNGUAGE, comprising 1,473 reports from 230 patients (1–15 studies each), including 30 patients with full longitudinal trajectories. Evaluation follows two stages: (i) single structuring, assessing localized semantic relations, and (ii) sequential structuring, evaluating consistency and organization of findings into clinical episodes across time.

---

[3]Binary attributes: DXSTATUS (positive/negative) and DXCERTAINTY (definitive/tentative)

[4]Non-binary attributes include: LOCATION, SEVERITY, ONSET, IMPROVED, WORSENED, PLACEMENT, NOCHANGE, MORPHOLOGY, DISTRIBUTION, MEASUREMENT, COMPARISON, PASTHX, OTHERSOURCE, ASSESSMENTLIMITATIONS

Table 1: Performance of various models under zero-shot and 5-shot settings. Left: single report structuring performance. Right: sequential report structuring performance.

| | | Single Structuring | | | | | | Sequential Structuring | | | | | |
| | | entity-relation | | | entity-relation-attribute | | | Entity Grouping | | | Temporal Grouping | | |
| Shot | Model | F1 | P | R | F1 | P | R | F1 | P | R | F1 | P | R |
|---|---|---|---|---|---|---|---|---|---|---|---|---|---|
| | GPT-4.1 | 0.91 | 0.83 | **1.00** | **0.78** | **0.79** | 0.77 | **0.67** | **0.68** | 0.71 | **0.84** | 0.83 | **0.86** |
| | Qwen3 | 0.73 | 0.58 | **1.00** | 0.62 | 0.53 | 0.75 | 0.51 | 0.43 | 0.65 | **0.84** | **0.87** | 0.82 |
| Zero | Deepseek-v3 | 0.87 | 0.76 | **1.00** | 0.76 | 0.72 | **0.80** | 0.43 | 0.30 | 0.76 | 0.81 | **0.87** | 0.75 |
| | Llama4-Maverick | 0.81 | 0.68 | **1.00** | 0.69 | 0.64 | 0.76 | 0.37 | 0.24 | **0.77** | 0.60 | **0.87** | 0.47 |
| | MedGemma-27b-text-it | 0.75 | 0.59 | **1.00** | 0.20 | 0.28 | 0.16 | 0.33 | 0.30 | 0.37 | 0.74 | 0.85 | 0.66 |
| | GPT-OSS-120b | **0.92** | **0.85** | **1.00** | 0.70 | 0.71 | 0.69 | 0.62 | 0.57 | 0.69 | 0.83 | 0.86 | 0.80 |
| | GPT-4.1 | **0.94** | **0.88** | **1.00** | **0.86** | **0.86** | **0.86** | **0.69** | **0.72** | 0.68 | **0.87** | 0.84 | **0.91** |
| | Qwen3 | 0.92 | 0.85 | **1.00** | 0.84 | 0.83 | 0.85 | 0.64 | 0.58 | 0.71 | 0.85 | 0.87 | 0.84 |
| 5-shot | Deepseek-v3 | 0.93 | **0.88** | **1.00** | **0.86** | 0.85 | **0.86** | 0.67 | 0.61 | 0.75 | 0.86 | 0.89 | 0.84 |
| | Llama4-Maverick | **0.94** | **0.88** | **1.00** | **0.86** | **0.86** | 0.85 | 0.54 | 0.39 | **0.87** | 0.63 | **0.92** | 0.48 |
| | MedGemma-27b-text-it | 0.90 | 0.82 | **1.00** | 0.81 | 0.80 | 0.82 | 0.55 | 0.50 | 0.61 | 0.82 | 0.87 | 0.78 |
| | GPT-OSS-120b | 0.90 | 0.83 | **1.00** | 0.81 | 0.79 | 0.83 | 0.66 | 0.60 | 0.74 | 0.83 | 0.86 | 0.80 |

**Single Structuring** We evaluate model performance on generating structured representations from individual reports by comparing predicted *(entity, relation, attribute)* triplets against expert annotations in LUNGUAGE. Using micro-averaged precision, recall, and F1 scores at both the entity–relation and full triplet levels, we assess GPT-4.1 Achiam et al. (2023) alongside several recent open-source LLMs Liu et al. (2024); OpenAI (2025); Sellergren et al. (2025); Touvron et al. (2023); Zheng et al. (2025) under the framework described in Section 4. As shown in Table 1, all models achieve perfect recall, with 5-shot prompting yielding F1 scores of 0.90–0.94 for entity–relation extraction and 0.81–0.86 for full triplets. Performance improves further with more few-shot examples, demonstrating the robustness of the framework despite the schema's complexity. Additional experiments, including vocabulary guidance, 10-shot prompting, and qualitative examples, are provided in Appendix B.3.

**Sequential Structuring** The second stage evaluates whether models can group temporally distributed findings into clinically meaningful categories, a task complicated by subtle semantic distinctions. For instance, "heart size" may group with "cardiomegaly," whereas "mediastinal silhouette" concerns shape and can remain normal despite cardiomegaly. Using micro-averaged F1 scores, we observe that zero-shot prompting already yields strong temporal grouping performance (F1 $\approx$ 0.80–0.84 for GPT-4.1 and other LLMs), whereas entity grouping is noticeably more variable across models, particularly among open-source LLMs. Providing five in-context examples stabilizes the predictions and consistently improves entity grouping, with GPT-4.1 reaching an F1 of 0.69 and most models exceeding 0.60, while temporal grouping remains high (F1 $\approx$ 0.82–0.87). As detailed in Appendix B.4, the remaining discrepancies in entity grouping mainly concern how finely entities are grouped, for example whether closely related lexical variants or attribute-specific mentions are merged or split. Consequently, because LUNGUAGESCORE (Section 5, Equation 3) relies on continuous semantic, temporal, and structural similarity over matched findings rather than exact group identity, predictions that differ only in grouping granularity still receive high similarity when they follow the same diagnostic trajectory.

## 6.2 EVALUATING LUNGUAGESCORE ALIGNMENT WITH RADIOLOGIST JUDGMENTS

We validate the diagnostic utility of **LUNGUAGESCORE** on the ReXVal dataset (Yu et al. (2023b)), which consists of 200 MIMIC-CXR report pairs annotated by six radiologists to benchmark alignment between automated metrics and expert judgments. As ReXVal contains only single reports, we evaluate the single-report version of LUNGUAGESCORE (semantic and structural alignment). We compare against BLEU, BERTScore, GREEN, FineRadScore, RaTEScore, RadGraph-F1, and RadGraph-XL F1, with implementation details in Appendix D.

Table 2 reports Kendall Tau and Pearson correlations between each metric and the number of radiologist-identified errors, where stronger alignment corresponds to more negative values. We also report 95% confidence intervals from 1,000 bootstrap resamples.

Our metric outperforms traditional structure- or semantics-based metrics (BLEU, BERTScore, RaTEScore, RadGraph-F1, RadGraph-XL F1) but falls slightly short of LLM-derived scores (FineRadScore, GREEN), which are explicitly tuned to ReXVal's error taxonomy. Nonetheless, LUNGUAGESCORE achieves performance close to these metrics while relying only on semantic and structural alignment rather than error-type supervision. Additional analyses in Appendix D show that LUNGUAGESCORE correlates strongly with all other metrics.

### 6.3 BENCHMARKING SINGLE-REPORT AND SEQUENTIAL REPORT GENERATION MODELS

We further validate **LUNGUAGESCORE** by benchmarking it against existing evaluation methods across a diverse set of report generation models, assessing its capacity to capture clinically meaningful differences at both single-report and patient-level scales. We categorize the models based on input modality: those utilizing only the *current timepoint single image* (Cvt2DistilGPT2 (Nicolson et al., 2023), RGRG (Tanida et al., 2023), MedGemma (Sellergren et al., 2025), Lingshu (Xu et al., 2025), CheXAgent (Chen et al., 2024)), and those incorporating the *current image plus prior context* (e.g., history section or prior image) (Medversa (Zhou et al., 2024), LIBRA (Zhang et al., 2025), MAIRA-2 (Bannur et al., 2024)).

**Radiology report generation** All models require frontal chest X-rays. MAIRA-2 additionally uses lateral images when available. RGRG and Cvt2DistilGPT2 generate findings sections, while Medversa, MedGemma, Lingshu, CheXAgent, and LIBRA produce full reports that include both findings and impression; we standardize all outputs into complete reports. The *single+prior* group is configured to consume both the current and a prior context. MAIRA-2 *(standard)* and Medversa also incorporate the history/indication text as contextual input. Implementation details are provided in Appendix E.

**Single-report setting** In this setting, we compare generated reports with ground-truth references on a study-by-study basis using the same patient as in the sequential evaluation, restricted to the 67 studies that contain frontal images.[5] Reference reports combine findings and impression. Table 3 summarizes results across metrics, including LUNGUAGESCORE. For LUNGUAGESCORE, we use LUNGUAGE as ground truth and compare against outputs from the structuring framework in Section 4. Overall, models that use both current and prior context perform better. MAIRA-2 (standard, highest-performing), Medversa, and LIBRA achieve higher scores than the single-image baselines on LunguageScore (single), indicating a clear benefit from incorporating longitudinal context even when evaluating single reports. Notably, while their advantage over single-image models is relatively modest under existing metrics, LunguageScore reveals a clearer gap between context-aware and single-image systems.

**Sequential Setting** We use the same reports as in the single-report setting but additionally include the history/indication section to provide context for patient trajectories. All models are evaluated in this sequential setting, including those that only take the current image as input. This is because the benchmark is organized around patient-level sequences: in routine practice, radiologists interpret each study in light of previous examinations, and the resulting reports describe how findings evolve over time rather than isolated snapshots. As a consequence, the reference reports form a temporally coherent longitudinal narrative for each patient. Running any model independently at each timepoint therefore induces its own predicted trajectory, which LUNGUAGESCORE can meaningfully compare against this coherent reference sequence.

Table 2: Kendall Tau and Pearson correlation coefficients (with 95% CIs) between single-report metrics and the total number of radiologist-annotated errors in each report, across the ReXVal dataset. Note that FineRadScore was inverted for comparability.

| Metric | Kendall Tau | Pearson |
|---|---|---|
| BLEU | -0.38 (-0.29, -0.48) | -0.53 (-0.45, -0.62) |
| BERTScore | -0.50 (-0.43, -0.57) | -0.63 (-0.55, -0.70) |
| GREEN | -0.63 (-0.56, -0.69) | -0.73 (-0.67, -0.78) |
| 1/FineRadScore | -0.69 (-0.63, -0.74) | -0.75 (-0.69, -0.80) |
| RaTEScore | -0.52 (-0.44, -0.59) | -0.63 (-0.55, -0.70) |
| RadGraph F1 | -0.57 (-0.50, -0.63) | -0.68 (-0.62, -0.74) |
| RadGraph-XL F1 | -0.53 (-0.45, -0.62) | -0.63 (-0.56, -0.72) |
| LUNGUAGESCORE | -0.58 (-0.52, -0.64) | -0.69 (-0.63, -0.74) |

---

[5]Studies without frontal images were excluded, which created gaps in sequential analyses for 5 of 10 patients.

Table 3: Evaluation of radiology report generation models using multiple metrics. Scores are averages with 95% CIs.

| Input | Model | Single-report setting | | | | | Sequential |
|---|---|---|---|---|---|---|---|
| | | RaTEScore | GREEN | 1/FineRadScore | RadGraph F1 | LUNGUAGESCORE | LUNGUAGESCORE |
| single | Cvt2DistilGPT2 | 0.491 (0.46, 0.52) | 0.240 (0.19, 0.29) | 0.167 (0.14, 0.20) | 0.179 (0.15, 0.21) | 0.367 (0.34, 0.40) | 0.371 (0.33, 0.41) |
| | RGRG | 0.547 (0.53, 0.57) | 0.266 (0.23, 0.30) | 0.139 (0.11, 0.17) | 0.264 (0.23, 0.29) | 0.406 (0.38, 0.43) | 0.391 (0.36, 0.42) |
| | MedGemma | 0.495 (0.48, 0.51) | 0.149 (0.10, 0.20) | 0.127 (0.11, 0.14) | 0.133 (0.12, 0.15) | 0.318 (0.30, 0.34) | 0.345 (0.32, 0.37) |
| | Lingshu | 0.483 (0.46, 0.50) | 0.173 (0.13, 0.22) | 0.141 (0.11, 0.17) | 0.150 (0.13, 0.18) | 0.344 (0.32, 0.37) | 0.356 (0.33, 0.38) |
| | CheXAgent | 0.528 (0.50, 0.55) | 0.241 (0.20, 0.29) | 0.131 (0.12, 0.14) | 0.228 (0.20, 0.26) | 0.380 (0.35, 0.41) | 0.388 (0.36, 0.42) |
| single + prior | Medversa | 0.543 (0.52, 0.57) | 0.314 (0.26, 0.37) | **0.183** (0.15, 0.22) | 0.238 (0.21, 0.27) | 0.409 (0.38, 0.44) | 0.410 (0.37, 0.45) |
| | LIBRA | 0.526 (0.50, 0.55) | 0.266 (0.22, 0.30) | 0.127 (0.12, 0.14) | 0.227 (0.20, 0.26) | 0.414 (0.38, 0.45) | 0.417 (0.38, 0.43) |
| | MAIRA-2 (standard) | **0.564** (0.54, 0.59) | **0.325** (0.28, 0.37) | 0.156 (0.14, 0.18) | **0.274** (0.25, 0.30) | **0.429** (0.40, 0.46) | **0.432** (0.41, 0.46) |
| | MAIRA-2 (cascade) | 0.547 (0.53, 0.57) | 0.299 (0.25, 0.34) | 0.171 (0.13, 0.21) | 0.233 (0.21, 0.26) | 0.419 (0.39, 0.45) | 0.416 (0.38, 0.45) |

As shown in Table 3, models that leverage prior context (*single + prior*) rank above the single-image models, with MAIRA-2 achieving the best performance and Medversa second. Models that omit this prior context (Cvt2DistilGPT2, RGRG, MedGemma, Lingshu, CheXAgent) perform worse; notably, when comparing the single-report and sequential evaluations, the LUNGUAGESCORE scores of RGRG and MAIRA-2 (cascade) decrease, whereas the scores of the other models improve. This pattern indicates that models which appear strong in the single-report setting can still produce temporally inconsistent diagnostic trajectories once their predictions are examined across the full patient sequence. LUNGUAGESCORE makes these temporal inconsistencies explicit and highlights the importance of evaluating report generators in the sequential setting. Further analysis of error sensitivity is provided in Section D.

# 7    CONCLUSION

This work introduces a comprehensive pipeline for evaluating radiology reports, grounded in LUN-GUAGE, a fine-grained benchmark for single and sequential structured chest X-ray reports. To our knowledge, it is the first benchmark and evaluation framework explicitly designed for longitudinal chest X-ray report generation and patient-level structured assessment. The dataset is intentionally designed as a dense, expert-verified evaluation resource, comprising 1,473 single reports and longitudinal patient trajectories with rich entity–attribute structure and temporal alignments curated by board-certified radiologists. Building on this foundation, we propose a two-stage LLM-based structuring framework that reliably maps free-text reports into schema-aligned representations across both single and sequential settings, and LUNGUAGESCORE, a clinically grounded metric that evaluates model outputs along semantic, structural, and temporal dimensions.

LUNGUAGESCORE operates on entity-centered representations that bundle attributes and temporal links for each finding, enabling joint assessment of diagnostic status, spatial and descriptive attributes, longitudinal change, and relevant context. Reports are first mapped by an LLM into schema-aligned structures, then evaluated by a fixed scoring function over semantic, temporal, and attribute fields, yielding transparent, attribute-wise interpretable scores that remain stable under small errors in the structured inputs. Empirically, LUNGUAGESCORE correlates well with radiologist-annotated errors on ReXVal in the single-report setting and, in the longitudinal setting, more clearly separates context-aware models from single-image baselines while reducing penalties for clinically appropriate but textually omitted findings. We expect our work to serve as a practical, interpretable testbed for fine-grained single-report and longitudinal evaluation, and we discuss remaining limitations and avenues for extension in Appendix F.

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

SUPPLEMENTARY CONTENTS

## A  LUNGUAGE DETAILS

**Dataset preparation**   LUNGUAGE aims to support patient-level evaluation of chest X-ray reports by modeling longitudinal diagnostic scenarios. To this end, we curated a benchmark dataset from the official test split of MIMIC-CXR, selecting patients with between 1 and 15 sequential studies. This yielded 230 patients with a total of 1,473 reports.

We followed the official MIMIC-CXR preprocessing protocol to extract structured text from each report. Specifically, we parsed the `history` (including "Indication"), `findings`, and `impression` sections. The history/indication field provides contextual information relevant to diagnostic reasoning, such as presenting symptoms (e.g., "fever," "fatigue," "cough") or evaluation intents (e.g., "rule out pneumonia"). In contrast, the findings and impression sections describe image-based observations and interpretations.

Section-level coverage across the dataset is summarized as:

- **History (i.e., Indication)**: 1,362 reports (92.5%)

- **Findings**: 1,224 reports (83.1%)

- **Impression**: 1,015 reports (68.9%)

Among the reports, 767 contained both findings and impression sections, 457 had findings only, 248 had impression only, and 1 contained only a history section. We excluded infrequently occurring sections such as `comparison` (often containing anonymized metadata using placeholders like "___"), and `technique` (e.g., "AP view"), as these appeared in fewer than 5% of cases and were not directly relevant to diagnostic content.

To preserve diagnostic integrity and linguistic variability, we retained all reports in their original form without content filtering. This includes templated reports (e.g., "No acute cardiopulmonary process") and incomplete notes. All reports were annotated using our schema-based pipeline with no preprocessing beyond section parsing. Structured reports were constructed by directly using the raw textual expressions from the original reports, rather than replacing them with normalized terms, to maintain alignment with the radiologists' source language.

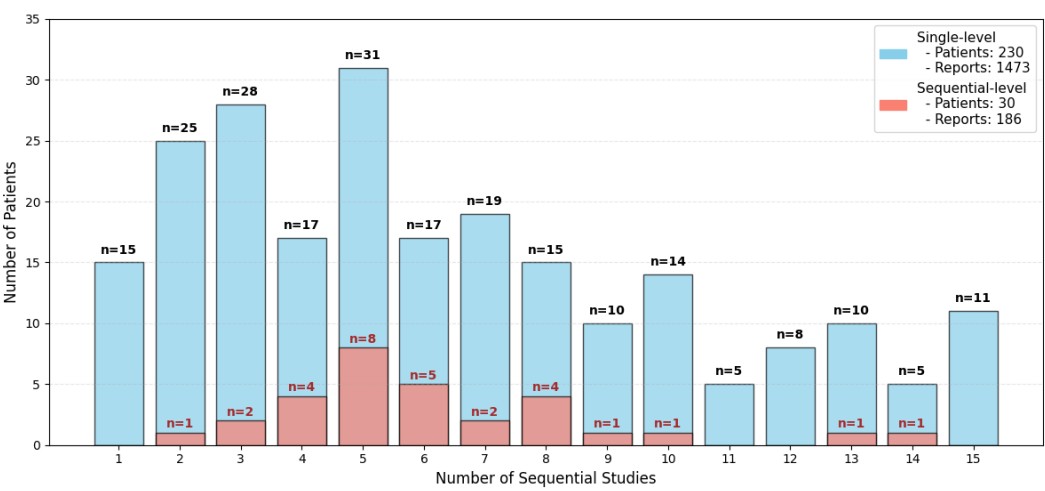

Figure A.1: **Distribution of the number of imaging studies per patient in LUNGUAGE.** Skyblue bars indicate the number of patients for each trajectory length (i.e., number of chest X-ray studies), reflecting the single-report annotation coverage. Salmon bars represent the subset of patients whose reports are also annotated at the longitudinal level. Values above the bars show the number of patients per group (n =), and for salmon bars, the number of patients with sequential annotations. The legend summarizes the total number of patients and reports included at each annotation level.

## A.1 SINGLE-REPORT SCHEMA: ENTITY AND RELATION DEFINITION

LUNGUAGE represents each radiology report as a structured collection of *(entity, relation, attribute)* triplets. This schema is designed to encode the diagnostic content of reports in a form that supports structured analysis, longitudinal reasoning, and machine-readable interpretation. It captures both observable features from chest X-ray (CXR) images and additional contextual elements embedded in clinical narratives.

### A.1.1 ENTITY TYPES

Entities represent clinically meaningful units such as findings, diagnoses, objects, or background context. Each entity is assigned one of six mutually exclusive Cat (category) labels, depending on whether it originates from the CXR image or external clinical sources. These six labels fall into two broad groups:

- **Chest X-ray Findings** are entities that can be directly visualized on the chest X-ray or inferred through image-based interpretation, possibly with minimal supporting context. These form the core of radiologic description and are divided into the following types:
  - **PF (Perceptual Findings)**: Visual features that are explicitly visible in the image and correspond to anatomical or pathological structures (e.g., "opacity", "pleural effusion", "pneumothorax"). These are the most direct and objective form of image evidence.
  - **CF (Contextual Findings)**: Diagnoses that require interpretation of visual findings in light of limited contextual knowledge (e.g., "pneumonia", "congestive heart failure"). These may involve reasoning beyond the image but still rely primarily on radiographic evidence.
  - **OTH (Other Objects)**: Non-anatomic elements such as medical devices, surgical hardware, or foreign materials visible on the image (e.g., "endotracheal tube", "central venous catheter", "foreign body"). These often require placement verification or complication monitoring.
- **Non Chest X-ray Findings** are entities that cannot be determined from the image alone and must be inferred from patient history, clinical documentation, or other diagnostic modalities:
  - **COF (Clinical Objective Findings)**: Structured clinical measurements or physical findings derived from sources such as laboratory tests or vital signs (e.g., "elevated white cell count", "low oxygen saturation"). These provide objective support for contextual interpretation.
  - **NCD (Non-CXR Diagnosis)**: Diagnoses that originate from non-CXR modalities (e.g., CT, MRI, serology) and are either mentioned for completeness or used to explain findings (e.g., "stroke", "AIDS").
  - **PATIENT INFO**: Historical or subjective patient information, such as symptoms or clinical background, that contributes to interpretation (e.g., "fever", "history of malignancy", "recent trauma").

Each entity is additionally annotated with the following attributes that define its diagnostic interpretation within the report:

- **DxStatus**: Indicates whether the entity is considered present or absent in the current study. This label is determined from report language and includes implications from stability or change. For example, "resolved effusion" is annotated as Positive, while "unchanged opacity" is Positive unless the prior state was normal, in which case it is Negative.
- **DxCertainty**: Reflects the level of confidence expressed by the radiologist, labeled as either Definitive or Tentative. Typical cues include phrases like "suggests", "cannot exclude", or "possibly indicative of", all leading to a tentative label.

### A.1.2 RELATION TYPES

Relations describe either attributes of a single entity or clinically relevant links between multiple entities. All relations must be grounded in the report text and can span across sentences within the same section.

**1. Diagnostic Reasoning** These relations connect semantically and clinically related entities. They encode the logic behind diagnostic interpretation.

- **Associate**: A bidirectional, non-causal relationship between entities that co-occur or are conceptually linked (e.g., "opacity" ↔ "consolidation"). When `Evidence` is used, a corresponding `Associate` is also required in the reverse direction.

- **Evidence**: A unidirectional relation in which a finding supports a diagnosis (e.g., "pneumonia" → "opacity").

**2. Spatial and Descriptive Attributes** These relations describe intrinsic visual characteristics of an entity as observed within a single chest X-ray image. Unlike temporal attributes, these do not require comparison with prior studies. Instead, they provide descriptive detail that refines the interpretation of a finding or object in terms of location, form, extent, intensity, and symmetry.

- **Location**: Specifies the anatomical or spatial position of the entity (e.g., "right upper lobe", "carina above 3 cm"). An entity may have multiple location labels, annotated as a comma-separated list (e.g., "right upper lobe, suprahilar"). Location applies to both disease findings and device placements (e.g., "fragmentation" of "sternal wires").

- **Morphology**: Describes the shape, form, or structural appearance of the entity (e.g., "nodular", "linear", "reticular", "confluent"). Morphological terms help differentiate types of opacities or identify characteristic patterns of pathology.

- **Distribution**: Refers to the anatomical spread or pattern of the entity (e.g., "focal", "diffuse", "multifocal", "bilateral"). This helps characterize whether the finding is localized or widespread, and whether it follows typical anatomical distributions.

- **Measurement**: Captures quantitative properties such as size, count, or volume (e.g., "2.5 cm", "few", "multiple"). These descriptors are typically numerical or ordinal and assist in severity grading or follow-up comparison.

- **Severity**: Reflects the degree of abnormality or clinical impact, often based on radiologic intensity or extent (e.g., "mild", "moderate", "severe", "marked").

- **Comparison**: Indicates asymmetry or difference across anatomical sides or regions within the same image (e.g., "left greater than right", "right lung appears denser"). This is distinct from temporal comparison and only refers to spatial contrasts visible in the current image.

**3. Temporal Change** These relations capture how an entity has changed over time by comparing the current study to previous imaging or known clinical baselines. Temporal attributes are essential for longitudinal interpretation and reflect disease progression, treatment response, or clinical stability. Unlike static descriptors, these attributes require temporal context and often imply clinical decision points.

- **Onset**: Indicates the timing or duration of a finding as described in the report (e.g., "acute", "subacute", "chronic", "new"). These descriptors suggest whether a condition has recently appeared or has been long-standing.

- **Improved**: Signals that a finding has regressed or resolved compared to a prior state (e.g., "resolved effusion", "decreased consolidation"). It is typically associated with positive treatment response or natural recovery.

- **Worsened**: Indicates that the condition has progressed, increased in extent, or become more severe over time (e.g., "enlarging opacity", "increased pleural effusion"). This is often associated with disease progression or complications.

- **No Change**: Describes a finding that has remained stable since a prior study (e.g., "unchanged opacity", "persistent nodule"). Although these are annotated as `Positive` by default, they are marked as `Negative` if the prior state was normal (i.e., continued absence of disease).

- **Placement**: Applies specifically to entities labeled as `OTH` (devices). It describes both the position (e.g., "in expected position", "malpositioned") and temporal actions involving the device (e.g., "inserted", "withdrawn", "removed"). This attribute is crucial for monitoring device-related interventions over time.

**4. Contextual Information** This category captures auxiliary information that influences the interpretation of findings but is not a primary descriptor of the radiologic appearance. These relations provide critical contextual cues—such as modality constraints, patient factors, or historical references—that support diagnostic interpretation. While not visual in the conventional sense, they are essential for accurately situating radiologic findings within the broader clinical scenario.

- **Past Hx**: Refers to the patient's prior medical or surgical history that contextualizes current findings (e.g., "status post lobectomy", "known tuberculosis"). These mentions often justify or explain current observations or exclude certain diagnoses.

- **Other Source**: Indicates that part of the reported information is derived from modalities other than chest X-ray (e.g., "seen on CT", "confirmed on MRI"). This distinction is important when findings cannot be visualized directly on the image being interpreted.

- **Assessment Limitations**: Describes technical or procedural factors that constrain the radiologist's ability to interpret the image accurately (e.g., "poor inspiration", "rotated patient position", "limited view due to overlying hardware"). These limitations help qualify the certainty or completeness of the report's conclusions.

## A.2 TASK-SPECIFIC VOCABULARY CONSTRUCTION

To systematically capture the range of descriptive, temporal, spatial, and contextual attributes in radiologic reporting, we constructed a structured vocabulary of relation terms grounded in all schema-defined relation types instantiated in LUNGUAGE. The process followed four stages: (1) automatic candidate extraction, (2) expert review and refinement, (3) hierarchical organization into clinically meaningful subcategories, and (4) normalization of lexical variants. This pipeline was designed to maximize coverage while ensuring clinical interpretability and internal consistency.

**Candidate extraction.** We first piloted schema and prompt designs on 100 sample reports, iteratively refining them before applying the finalized schema to the full set of 1,473 reports (see Appendix A.3 for details). Using GPT-4 (Achiam et al. (2023)), we produced initial structured outputs and extracted candidate terms corresponding to each relation type. This step emphasized high recall to capture the breadth of linguistic variation present in free-text radiology reports and provided a basis for analyzing hierarchical consistency across categories.

**Expert review and refinement.** Four board-certified physicians independently reviewed the candidate vocabularies for each relation category, verifying accurate categorization and eliminating spurious or ambiguous expressions. Disagreements were adjudicated through consensus meetings (Figure A.6), prioritizing clinical interpretability and reproducibility. This process was especially important for borderline cases such as distinguishing between `Condition` terms under MORPHOLOGY and subtle gradations of SEVERITY, or between `filed-of-view limitations` and `patient-related limitations`.

**Hierarchical organization of entity, location, and attribute taxonomies.** All vocabularies were organized hierarchically to reflect radiologic conventions and enable reasoning across different levels of granularity. A comprehensive overview of the entity taxonomy (Figure A.2), location taxonomy (Figure A.3), and attribute taxonomy (Figure A.4), together with representative examples, is provided in Table A.1. They can be grouped into three major taxonomies:

- **Entity taxonomy.** Entities were first assigned to one of six mutually exclusive `Cat` labels: `PF` (Perceptual Findings), `CF` (Contextual Findings), `COF` (Clinical Objective Findings), `NCD` (Non-CXR Diagnosis), `OTH` (Other Objects), and `PATIENT INFO` (Patient Information). Within each label, entities were further classified into subcategories such as *Diagnostic Observations*, *Anatomical Entities*, *Diseases and Disorders*, *Medical Devices*, or *Symptoms & Signs*. Representative examples include: "opacity" and "right hilum" (`PF`), "pneumonia" and "congestive heart failure" (`CF`), "oxygen saturation" (`COF`), "stroke" (`NCD`), "central venous catheter" (`OTH`), and "fever" or "chronic dyspnea" (`PATIENT INFO`). Normalization ensured consistent representation, while diverse raw expressions were linked at the lowest level (e.g., "pneumonia" → "PNA," "pneumonias").

- **Location taxonomy.** The most extensive vocabulary, comprising 546 terms, was organized into hierarchical paths that mirror clinical localization practices. High-level systems included

*respiratory* (229), *musculoskeletal* (84), *cardiovascular* (73), and *others* (160). Examples of hierarchical paths include: "lung → lobe → right → upper," "heart → chamber → atrium → left," "spine → thoracic → vertebra → T4." This structuring enables reasoning from coarse system-level interpretation to fine-grained anatomical localization.

- **Attribute taxonomy.** Attributes were systematically organized into descriptive and temporal axes. MORPHOLOGY (205) was divided into *shape and structure*, *texture and density*, and *condition*. Temporal change included ONSET (57), IMPROVED (118), WORSENED (102), and NO CHANGE (138), each stratified into graded interpretations (e.g., "moderate improvement," "minimal worsening"). Device-related metadata were captured under PLACEMENT (74), describing both positional accuracy (e.g., "malpositioned") and procedural changes (e.g., "removed," "repositioned"). Additional axes included MEASUREMENT (139), SEVERITY (86), DISTRIBUTION (37), and COMPARISON (44). Auxiliary types captured contextual but clinically relevant information: ASSESSMENT LIMITATIONS (233; e.g., "rotated patient," "poor inspiration"), OTHER SOURCE (55; e.g., CT, MRI), and PAST HX (39; e.g., "status post," "history of malignancy"). Our vocabulary was restricted to relation types that correspond to lexically explicit attributes. Four relation types—EVIDENCE, ASSOCIATE, DXSTATUS, and DXCERTAINTY—were excluded. These relations are critical to the annotation schema but represent pragmatic inference rather than explicit lexical expressions. For instance, EVIDENCE and ASSOCIATE encode reasoning links between entities, often spanning sentences, while DXSTATUS and DXCERTAINTY capture interpretive stance (e.g., presence vs. absence, tentative vs. definitive).

**Normalization.** The resulting vocabulary includes 14 relation types derived from lexical evidence, each normalized to a preferred set of terms and organized into semantically coherent subcategories. We additionally performed UMLS mapping wherever possible to align relation terms with existing biomedical ontologies, while preserving terms that fall outside conventional coverage. This ensured both lexical consistency and clinical validity, supporting future integration. Beyond its role in structuring chest X-ray reports, this vocabulary provides a reusable lexicon for tasks such as query expansion, ontology alignment, multimodal grounding, and patient-level reasoning, thereby establishing a clinically grounded and internally consistent taxonomy of radiologic language. Detailed construction procedures are explained in Appendix A.3.

**Comparison with prior resources and applications.** Compared to prior resources such as Rad-Graph (Jain et al. (2021)), our vocabulary introduces a substantially more fine-grained taxonomy. RadGraph defines only two entity types—*Anatomy* and *Observation*—and represents descriptive information indirectly through coarse relations such as *modify* or *suggestive of*. In contrast, our schema explicitly differentiates attributes such as MORPHOLOGY into *shape and structure*, *texture and density*, and *condition*, and provides graded subtypes for both temporal progression and severity. This level of granularity better reflects the linguistic practices of radiologists and enables more nuanced downstream evaluation.

Table A.1: Vocabulary Overview Taxonomy

**Entity Categories**

| Category | Subcategory | Example terms |
|---|---|---|
| pf (Perceptual Findings) | Diagnostic Observations, Anatomical Entities, Diseases and Disorders | opacity, right hilum |
| cf (Contextual Findings) | Diseases and Disorders, Diagnostic Observations | congestive heart failure, pneumonia |
| cof (Clinical Objective Findings) | Diagnostic Observations, Diseases and Disorders | oxygen saturation, anti pd1 antibody |
| ncd (Non-CXR Diagnosis) | Diseases and Disorders, Diagnostic Observations | stroke, seizure disorder |
| oth (Other Objects) | Medical Devices, Procedures & Surgeries, Treatment & Medications | central venous catheter, lobectomy |
| patient info | Symptoms & Signs, Diseases & Disorders, Treatment & Medications, Procedures & Surgeries | fever, cough, chronic dyspnea |

**Attribute Categories: Spatial and Descriptive**

| Category | Subcategory | Example terms |
|---|---|---|
| severity | Extreme, Significant, Moderate, Mild, Minimal | moderate, severe |
| measurement | Size, Quantity, Normality | 2.5 cm, multiple |
| morphology | Shape & Structure, Texture & Density, Condition | nodular, reticular |
| distribution | Pattern, Extent, General Description | diffuse, focal |
| comparison | Location & Laterality, Degree & Description | left greater than right |

**Attribute Categories: Temporal Change**

| Category | Subcategory | Example terms |
|---|---|---|
| onset | Acute/Sudden, Chronic/Long-term, Progressive | acute, chronic |
| improved | Extreme, Significant, Moderate, Mild, Minimal | resolved, decreased |
| worsened | Extreme, Significant, Moderate, Mild, Minimal | enlarging, increased |
| no change | No Change, Minimal Change | unchanged, persistent |
| placement | Standard Position, Repositioning, New Placement, Removal, Nonstandard Position | inserted, malpositioned |

**Attribute Categories: Contextual Information**

| Category | Subcategory | Example terms |
|---|---|---|
| assessment limitations | Evaluation, Field-of-View, Patient-Related, Technical | poor inspiration, rotated patient |
| other source | Image, Signal, External Source | CT, MRI |
| past hx | Past Hx | status post, known |

**Location Taxonomy and Coverage**

| Top-level Category | Category distribution (%) | Example Anatomical Sites | Max Depth | Example Location Paths |
|---|---|---|---|---|
| Respiratory | ≈42% | Lungs, pleura, bronchi, thoracic wall | up to 7 | *lung > lobes > right > upper, pleura > left > upper* |
| Cardiovascular | ≈13% | Heart chambers & valves, aorta, vena cava, jugular/supra-cardiac veins | up to 6 | *vessels > aorta > arch, heart > chambers > atrium > right, veins > jugular > internal > right* |
| Musculoskeletal | ≈15% | Spine (cervical—lumbar), ribs, clavicle, shoulder & acromioclavicular joints | up to 6 | *spine > thoracic, bones > ribs > left, joints > shoulder > right* |
| Abdominal | ≈6% | Stomach, bowel segments, abdominal quadrants, sub-diaphragmatic spaces | up to 6 | *stomach > fundus, quadrants > right, organs > intestines > duodenum* |
| Mediastinum | ≈4% | Paratracheal, carinal, paramediastinal compartments | up to 5 | *paratracheal > right, paramediastinal_region > right, carina* |
| Other structures / Descriptors | ≈19% | Axilla, neck, extremities, directional descriptors, device placements | up to 5 | *axilla > left, neck > lower, medical_device* |

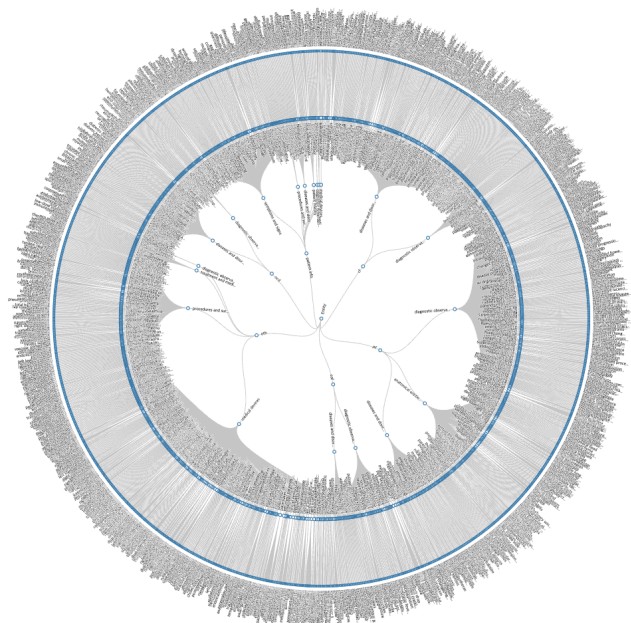

(a) Entity taxonomy (simplified). Shows only *Category* (PF, CF, OTH, COF, NCD, PATIENT INFO) and *Subcategory* (e.g., Diagnostic Observation, Anatomical Entity, Disease and Disorder).

(b) Entity taxonomy (full). Extends the simplified view by adding *Normalized terms* (canonical forms) and *Raw terms* (report expressions). For example, "pneumonia" (PF, Diagnostic Observation) is normalized to a standard form and may appear in reports as "PNA" or "pneumonia."

Figure A.2: **Entity taxonomy.** Comparison between simplified and full versions. The simplified taxonomy shows only up to Subcategory, while the full taxonomy additionally captures normalized terms and raw report expressions.

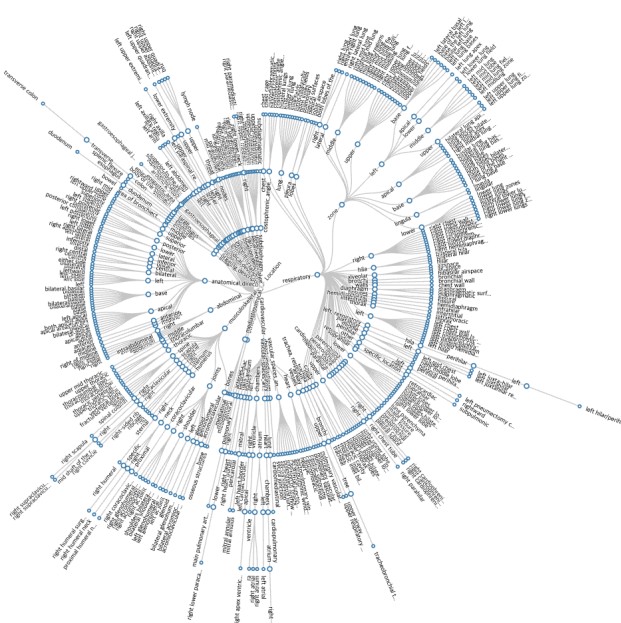

(a) Location taxonomy (simplified). Shows only up to depth 2 of the hierarchy, where broad anatomical systems (e.g., *respiratory*, *cardiovascular*) are subdivided into major regions.

(b) Location taxonomy (full). Extends the simplified view into a full hierarchical tree, reaching the lowest level of specificity as expressed in raw reports. For example, the *respiratory* system branches into "lung" → "left lung" → "left lower lobe," capturing fine-grained terms systematically.

Figure A.3: **Location taxonomy.** Comparison between simplified and full versions. The simplified taxonomy displays only the top levels of anatomical systems, while the full taxonomy represents the entire structured hierarchy down to the raw report expressions.

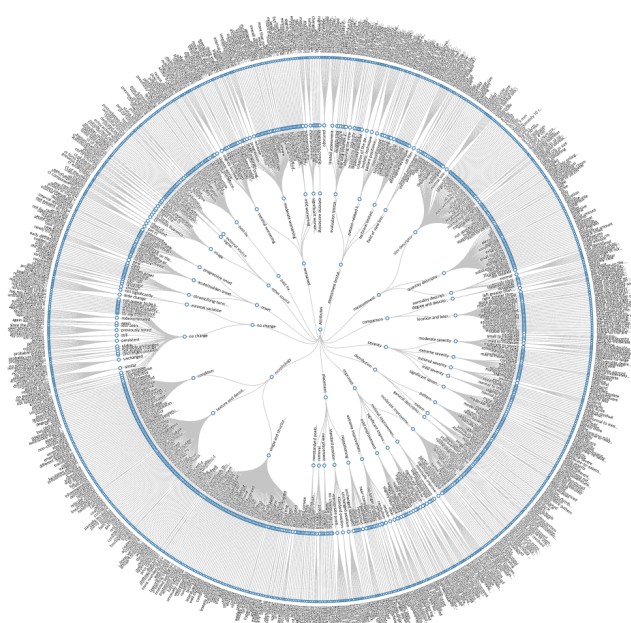

(a) Attribute taxonomy (simplified). Shows only *Category* (e.g., Severity, Morphology, Distribution, Temporal change, Contextual information) and their *Subcategories* (e.g., "Extreme–Minimal" scale for Severity, "Acute/Chronic" for Onset).

(b) Attribute taxonomy (full). Extends the simplified view by adding *Normalized terms* and *Raw report terms*. For example, the category *Improved* → subcategory *Minimal improvement* has normalized terms like "minimally improve" and maps to diverse raw expressions such as "somewhat better," "somewhat improved," or "slightly improved."

Figure A.4: **Attribute taxonomy.** Comparison between simplified and full versions. The simplified taxonomy presents categories and subcategories only, while the full taxonomy systematically incorporates normalized forms and raw report expressions used in clinical texts.

## A.3 ANNOTATION PROCESS

This section outlines the protocol used to ensure consistency, clinical accuracy, and transparency in our structured annotation process (Figure A.5). A key principle of our design was to adopt a *vocabulary-first approach*. Instead of starting directly with report-level annotations, we first extracted all unique candidate terms from the structured outputs and asked radiologists to evaluate them in isolation. Each annotator independently assessed whether a term was correctly categorized under the schema (keep, normalize, reassign, or remove) and proposed refinements when necessary. This strategy served two purposes: (i) it ensured that the schema categories were well defined and clinically meaningful before applying them at scale, and (ii) it minimized inconsistencies during report annotation by locking a shared vocabulary in advance. Consensus was reached through majority voting and resolution of edge cases, producing a stable schema–vocabulary foundation upon which high-quality single and sequential report annotations were later built.

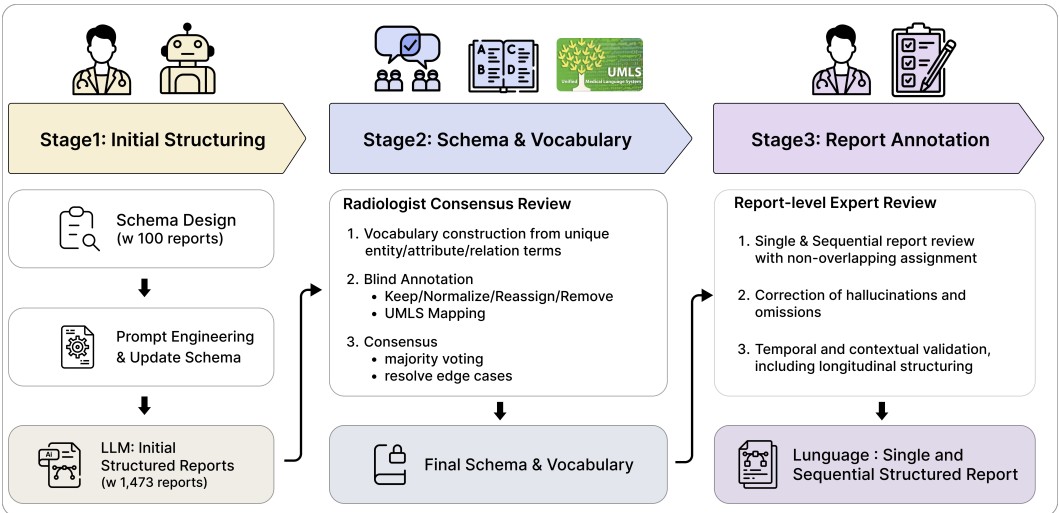

Figure A.5: **Annotation protocol.** The process begins with **Stage1: Initial Structuring**, where radiologists design an initial schema based on 100 chest X-ray reports and iteratively update it through prompt engineering. A large language model (LLM) then generates preliminary structured outputs from 1,473 reports, from which candidate vocabulary terms are extracted. In **Stage2: Schema & Vocabulary**, four board-certified radiologists conduct blinded annotation on all extracted terms (keep, normalize, reassign, or remove), supplemented with UMLS term mapping for interoperability. Consensus is reached through majority voting and resolution of edge cases, after which the schema and vocabulary are refined and locked. In **Stage3: Report Annotation**, radiologists perform expert review of non-overlapping subsets of single and sequential reports. This step includes correction of hallucinations and omissions, validation of temporal and contextual dependencies (including longitudinal structuring), and linking of cross-sentence relations such as ASSOCIATE and EVIDENCE. The pipeline yields a final schema and vocabulary as well as high-quality single and sequential structured reports (LUNGUAGE benchmark).

**Schema and Vocabulary Development and Validation.** The construction of the LUNGUAGE schema followed a vocabulary-first process that combined expert-driven refinement with large-scale automatic extraction. We began by randomly sampling 100 chest X-ray reports and manually drafting an initial schema. This draft was iteratively refined through prompt engineering: categories and relations were adjusted while repeatedly checking coverage against the same 100 reports. Once a stable structure was established, the process was scaled to all 1,473 reports using schema-guided prompts to a large language model, which produced preliminary structured drafts. These drafts were not treated as final structure reports, but instead served to systematically collect the entire lexical space of candidate terms across the ENTITY, ATTRIBUTE, and RELATION fields. This vocabulary-first approach ensured that the schema was grounded in actual report variation while also capturing rare or unconventional descriptors.

Figure A.6: **Example of consensus protocol for vocabulary validation.** This figure illustrates the vocabulary validation process using the *morphology* category as an example. For illustration, only the results from two reviewers are shown, although in the real protocol, all four radiologists independently reviewed every candidate term. The column *Lemma (ori)* represents raw vocabulary extracted from the LLM. Reviewers evaluated each term to determine whether it should remain in the category, be reassigned, merged as a synonym, or removed, and also identified appropriate synonyms. Green highlights indicate terms where reviewer opinions diverged, requiring consensus discussion. While not shown in this example, the full protocol also incorporated UMLS mappings and subcategory assignments.

The candidate vocabulary was then subjected to blinded review by four board-certified radiologists (an example is shown in Figure A.6). For each term, annotators determined whether to *keep*, *normalize*, *reassign*, or *remove*, and assigned it to an appropriate schema category (e.g., PF, CF, OTH). Disagreements were resolved through majority voting, while ambiguous cases were recorded in an *edge-case log* and revisited during consensus meetings. To promote alignment with established clinical practice and interoperability, raw terms were cross-checked against Fleischner Society terminology (Bankier et al. (2024)), particularly during normalization, and mapped to UMLS concepts when appropriate (e.g., UMLS TERM (CODE: C12345)).

Figure A.7: **UMLS mapping tool.** A custom interface was developed to retrieve candidate concepts via the UMLS API. Radiologists manually reviewed suggested mappings to select the most semantically aligned concepts, while terms without clear matches were explicitly marked as unmapped (–).

We developed a custom UMLS mapping tool (Figure A.7) to retrieve candidate concepts via the UMLS API, and radiologists manually reviewed these candidates to select the most semantically

aligned concepts. Terms without a clear correspondence were not forced into external standards; instead, they were explicitly marked as unmapped (– (CODE: -)). All mappings, including unmapped entries, were preserved in the final vocabulary for transparency and reuse.

Through this iterative adjudication and validation process, the vocabulary evolved from a raw set of extracted terms into a clinically coherent schema encoding core entity categories, relation types (e.g., DxCERTAINTY, ASSOCIATE, EVIDENCE), and detailed attributes (e.g., MORPHOLOGY, MEASUREMENT, TECHNICAL LIMITATION). The outcome was a locked schema and vocabulary that balanced comprehensive coverage with clinical precision, establishing the foundation for all subsequent annotation in the LUNGUAGE benchmark.

**Report Annotation and Validation.** After the schema and vocabulary were finalized, structured report annotation proceeded under a standardized workflow designed to ensure both consistency and clinical reliability. Radiologists worked with shared resources, including (i) a finalized schema document defining all entity, relation, and attribute types and their origin (image-derived vs. context-derived), (ii) an evolving *edge-case log* recording ambiguous examples and their resolutions, and (iii) a set of standardized decision rules for term normalization and schema assignment.

Using a custom annotation interface (Figure A.8), annotators reviewed non-overlapping subsets of reports to avoid redundancy. Their responsibilities extended beyond sentence-level checks to include linking observations across sentences (e.g., ASSOCIATE, EVIDENCE), validating temporal and contextual dependencies, and identifying hallucinations or omissions introduced during LLM-based structuring. This two-stage pipeline—automatic structuring followed by expert curation—yielded high-quality annotations for all 1,473 single reports and 186 longitudinal cases over a six-month period, with radiologists contributing multiple hours per week and participated in weekly review meetings. The resulting resource demonstrates annotation consistency and clinical reliability, providing a validated foundation for the LUNGUAGE benchmark.

### A.3.1 SINGLE REPORT ANNOTATION DETAILS

**Single Structured Report Statistics**

- Total number of reports: 1,473 chest X-ray reports
- Total number of patients: 230
- Number of imaging studies per patient: Ranges from 1 to 15
- Total number of annotated entities: 17,949
- Total number of annotated relation–attribute pairs: 23,307

To construct a clinically reliable gold-standard dataset, we implemented a structured annotation pipeline that reviewed and refined the initial triplets (entity-relation-attribute) generated by GPT-4 (0613). Unlike the vocabulary construction phase—which focused on individual terms without considering report context—this stage involved section-by-section review of all structured outputs in each report to ensure contextual accuracy and logical consistency.

All 1,473 chest X-ray reports in LUNGUAGE were divided evenly among annotators. Each annotator independently reviewed approximately one-quarter of the dataset, ensuring balanced coverage and minimizing reviewer bias across the annotated corpus. Within each report, annotators examined the structured outputs across the `history/indication`, `findings`, and `impression` sections. The goal was to verify whether the extracted *(entity, relation, attribute)* triplets accurately captured the meaning of the source text and aligned with the predefined schema.

This review explicitly included schema elements that require contextual interpretation and cannot be evaluated at the lexical level alone—namely, DxSTATUS, DxCERTAINTY, ASSOCIATE, and EVIDENCE. These attributes reflect interpretive judgments, such as identifying when an "opacity" supports a diagnosis of "pneumonia" or whether two entities should be linked through an associative relation. Annotators verified whether such relations were correctly inferred from the surrounding text and whether the attributes assigned to each entity (e.g., presence, uncertainty, temporal change) matched the narrative context.

To support this process, we developed a custom annotation interface (Figure A.8) that displayed the original report text alongside GPT-4's predicted triplets and an editable table of structured fields.

Each sentence in the report was paired with its associated annotations, including entity category, relation type, and all relevant attributes. Annotators could directly add, edit, remove, or merge entries to reflect clinically accurate interpretations. For example, terms like "ground glass opacity"—which could be mistakenly split—were merged into a single PF (perceptual finding) entity based on how radiologists commonly use the phrase. Annotation was conducted separately for each section (`history`, `findings`, `impression`), and the interface supported sentence-level review within each section to ensure consistent entity–relation mappings when terms appeared across multiple sentences.

As a result of this process, the finalized gold-standard dataset includes 17,949 validated entities and 23,307 relation instances. These annotations encompass both explicit descriptive attributes and contextually inferred diagnostic relationships, providing a robust benchmark for evaluating schema-based information extraction systems in chest radiograph interpretation. As an illustration, Figure A.9 shows the knowledge graph representation of a single annotated report drawn from the dataset.

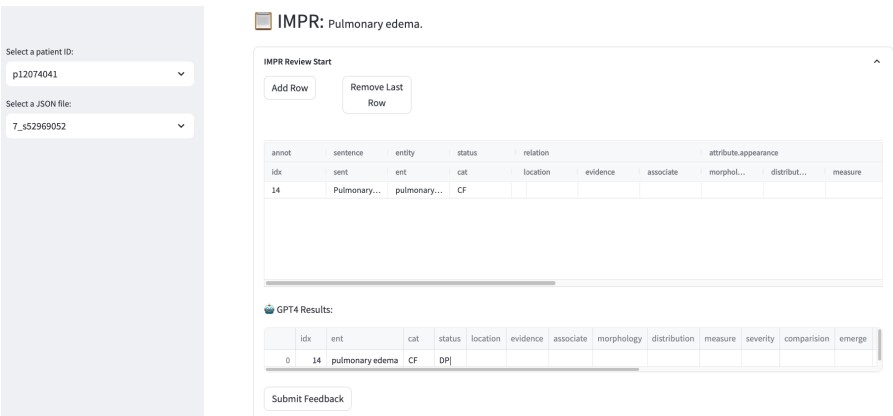

Figure A.8: Annotation interface used during gold dataset construction. Annotators reviewed GPT-4-generated triplets per report section and refined the entity–relation structure to ensure schema correctness and contextual validity.

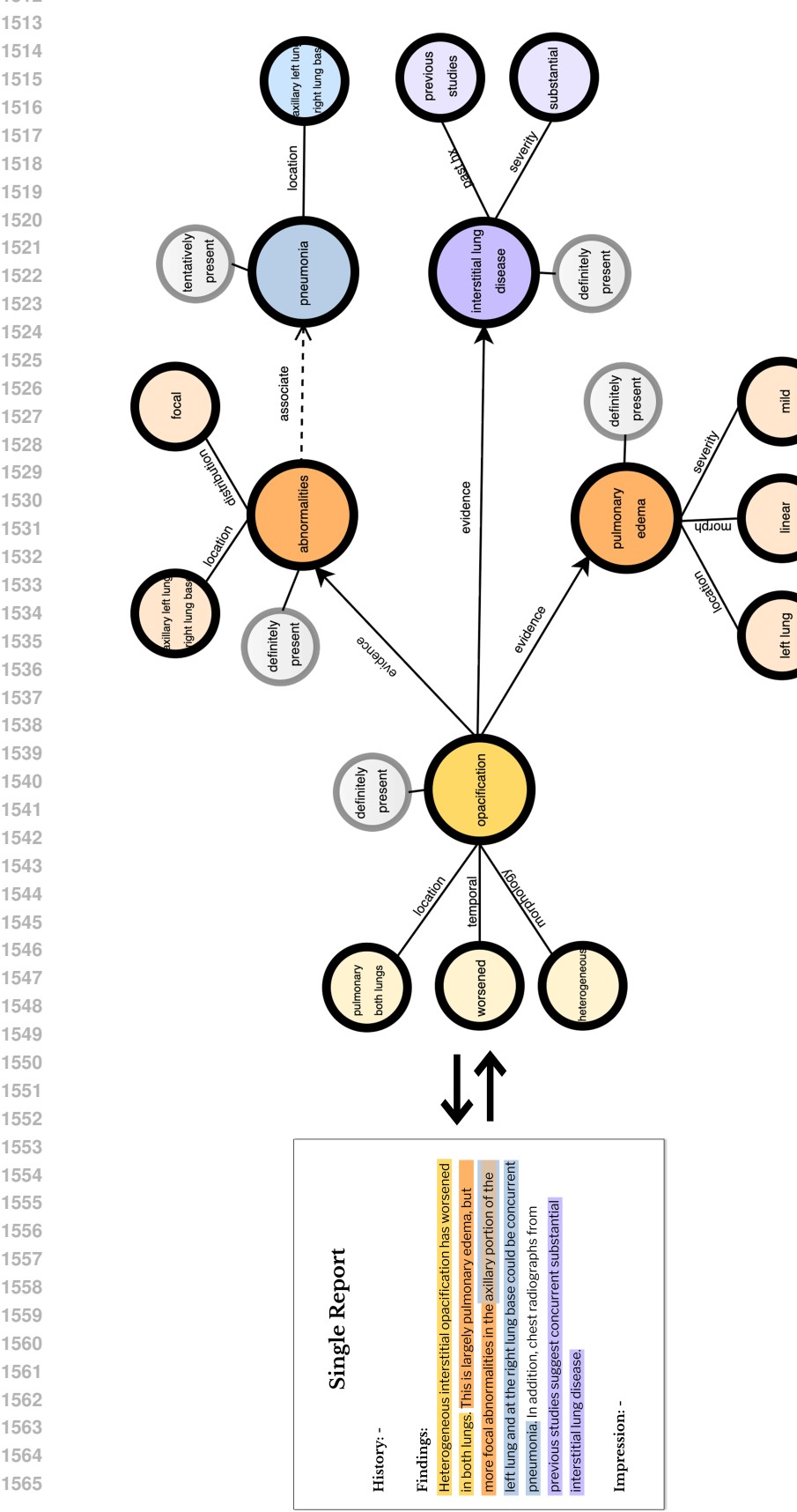

Figure A.9: **Structured Report as Knowledge Graph .** The figure illustrates how the findings section of a single chest X-ray report (bottom box) is structured under our schema. The text is decomposed into entities and their relations, including entity–entity links (e.g., associate, evidence) and entity–attribute links (e.g, status, certainty, location, severity, morphology). Connecting these components yields a knowledge graph (KG) that makes explicit both the relationships among entities and the descriptive properties attached to each entity.

### A.3.2 SEQUENTIAL REPORT ANNOTATION DETAILS

**Sequential Structured Report Statistics**

- Total number of reports: 186 chest X-ray reports

- Total number of patients: 30 (subset of the 230-patient cohort)

- Reports per patient: between 2 and 14

- Time intervals between reports: from 1 day to 1,200 days

- Observation pairs: 95,404 in total. This number comes from comparing every possible pair of annotated entities (observations) within each patient trajectory (i.e., all $\binom{n}{2}$ combinations). For example, a patient trajectory containing 34 entities yields 561 pairwise comparisons, while a dense case with 141 entities results in 9,870 pairs.

In contrast to the single-report structuring phase, which focused on refining schema-based annotations within individual reports, the sequential annotation phase aimed to assess the longitudinal consistency of entity-level interpretations across temporally ordered reports from the same patient. This required global comparisons across all sections—`history`, `findings`, and `impression`—integrating entity–relation triplets into clinically coherent sequences.

Unlike earlier phases that processed each report independently, this step involved exhaustive pairwise comparisons of all annotated expressions across time. Annotators judged whether lexically distinct phrases referred to the same underlying clinical entity by examining radiological terminology, anatomical location, temporal modifiers (e.g., "resolving", "unchanged"), and diagnostic specificity. Expressions identified as referring to the same finding were grouped together; otherwise, they were assigned to separate entity groups.

To further structure these entity groups, we assessed whether each represented a single episode of care or multiple distinct episodes. This required examining the temporal order and interval between observations. Intervals were computed using the `StudyDate` metadata from MIMIC-CXR, and episode boundaries were assigned based on temporal coherence—considering factors such as time gaps, patterns of resolution or worsening, and recurrence of findings.

For example, a progression from "moderate left effusion" (day 0) to "small effusion" (day 14) and "trace effusion" (day 45) was treated as a single resolving episode. However, a subsequent "moderate effusion" on day 180 was regarded as a separate episode, while all entities assigned to either episode are grouped into the same Entity Group. Similarly, "right lower lobe opacity" followed by "resolving infiltrate" was interpreted as one episode, whereas a new "opacity" on day 150 initiated a different episode. This process was applied to 186 chest X-ray reports from 30 patients, yielding longitudinal annotations that capture consistent entity grouping across lexical variations and clinically coherent organization of episodes based on temporal reasoning.

To better characterize the annotation results, we summarize the distribution of entity groupings and temporal episodes in Table A.2. The columns report:

- **# Reports:** The total number of reports per patient sequence.

- **Entity Group Distribution:** The number of findings assigned to each entity group (#Group), after normalization and longitudinal reasoning. Some groups consist of a single unique expression, while others aggregate multiple semantically related terms.

- **Temporal Group Distribution:** The number of findings assigned to each temporal group (#Group), where each group represents a distinct clinical episode.

Table A.2: Distribution of entity groups and temporal groups across annotated patient sequences.

| Subject ID | # Reports | Entity Group Distribution (#Group:Count) | Temporal Group Distribution (#Group:Count) |
|---|---|---|---|
| p10046166 | 6 | 1:26, 2:3, 3:3, 5:1, 7:1 | 1:32, 2:2 |
| p10274145 | 5 | 1:19, 2:11, 3:2, 4:3 | 1:33, 2:2 |
| p10523725 | 9 | 1:36, 2:6, 3:3, 4:2, 5:2, 7:2 | 1:47, 2:2, 3:1, 6:1 |
| p10532326 | 5 | 1:36, 2:6, 6:1 | 1:42, 2:1 |
| p10885696 | 8 | 1:33, 2:9, 3:5, 5:1, 6:2, 12:1 | 1:45, 2:2, 3:4 |
| p10886362 | 10 | 1:26, 2:3, 3:6, 4:4, 6:1, 7:1, 9:1, 13:1 | 1:39, 2:4 |
| p10959054 | 7 | 1:31, 2:6, 3:2, 4:2, 5:1, 6:1, 9:1 | 1:37, 2:5, 3:2 |
| p11540283 | 5 | 1:26, 2:5, 3:1, 4:2 | 1:33, 4:1 |
| p11607628 | 8 | 1:11, 2:2, 3:2, 4:4, 5:1, 6:2, 7:1, 8:1, 10:1 | 1:23, 2:2 |
| p11879886 | 6 | 1:27, 2:10, 3:3, 4:3, 5:2, 6:1 | 1:39, 2:4, 3:2, 4:1 |
| p12433421 | 13 | 1:49, 2:6, 3:10, 5:1, 7:1, 17:1 | 1:66, 2:2 |
| p12966004 | 3 | 1:21, 2:10, 4:1, 5:1 | 1:27, 2:5, 3:1 |
| p15094735 | 2 | 1:8, 2:5, 3:2, 4:1 | 1:13, 2:3 |
| p15109122 | 4 | 1:11, 2:1, 3:4, 4:1 | 1:16, 2:1 |
| p15207316 | 4 | 1:16, 2:6, 3:4 | 1:26 |
| p15272972 | 5 | 1:10, 2:3, 3:3, 4:2, 5:1 | 1:17, 2:2 |
| p15321868 | 6 | 1:24, 2:5, 3:2, 4:1, 5:2 | 1:32, 2:2 |
| p15446959 | 5 | 1:29, 2:7, 3:3, 4:2 | 1:37, 2:4 |
| p15881535 | 3 | 1:17, 2:2, 3:2, 5:1 | 1:20, 2:2 |
| p16059470 | 4 | 1:29, 2:5, 3:3, 6:1 | 1:37, 2:1 |
| p17270742 | 5 | 1:25, 2:4, 3:5, 4:2, 5:1 | 1:30, 2:4, 3:3 |
| p17288844 | 6 | 1:42, 2:8, 3:2, 4:1, 5:1 | 1:54 |
| p17396677 | 4 | 1:18, 2:3, 3:3, 4:1 | 1:23, 2:2 |
| p17720924 | 8 | 1:30, 2:8, 3:5, 4:1, 5:1 | 1:41, 2:2, 4:2 |
| p17962324 | 5 | 1:37, 2:4, 3:3, 4:1 | 1:43, 2:1, 3:1 |
| p18079481 | 14 | 1:34, 2:10, 3:3, 4:2, 6:3, 7:3, 8:1 | 1:43, 2:10, 3:3 |
| p18417750 | 7 | 1:41, 2:10, 3:5, 9:1 | 1:47, 2:7, 3:1, 6:2 |
| p18517718 | 6 | 1:15, 2:2, 3:4, 4:2, 5:1, 7:1 | 1:25 |
| p18570152 | 5 | 1:23, 2:4, 3:2, 4:1, 5:1, 6:2 | 1:25, 2:4, 3:3, 4:1 |
| p19150427 | 8 | 1:42, 2:7, 3:2, 4:1, 6:1 | 1:49, 2:2, 3:1, 4:1 |

Across the 30 patients in the sequential evaluation phase, the number of temporal groups assigned to a single entity group ranged from 1 to 6, indicating that some findings were observed in multiple distinct clinical episodes over time. Likewise, the number of distinct entity groups varied significantly. Most entity groups consisted of a single mention, but some aggregated up to 17 lexically different expressions. For example, subject p12433421 exhibited the most diverse entity grouping, with 17 distinct phrases all referring to variations of pleural effusion (e.g., "effusion," "pleural effusion," "pleural effusion left") unified under one normalized cluster. Similarly, subjects p10523725 and p18417750 exhibited high temporal discontinuity, with single entity groups spanning up to 6 distinct episodes (e.g., recurrent dyspnea separated by periods of resolution). These results highlight the complexity and variability of radiologic expression in longitudinal reporting, and underscore the necessity of models and metrics capable of robustly handling both semantic variation and episodic continuity in time-aware clinical tasks.

## A.4 Detailed Comparison with Existing Structured-Report Datasets

To clarify the design of our schema, we compare it against prior structured-report datasets using a common criterion. For fair comparison, all resources are re-aligned to a unified Named Entity Recognition (NER) / Relation Extraction (RE) standard: core clinical concepts are retained as **entities**, while descriptive aspects (e.g., status, measurement, severity) are recategorized as **relation labels**. This alignment can differ from the original dataset definitions. For example, RadGraph (Jain et al. (2021)) defines *Anatomy* and *Observation* as entities, but further subdivides observations into three uncertainty levels (*Definitely Present*, *Uncertain*, *Definitely Absent*), yielding four entity categories. In our comparison, however, only *Anatomy* and *Observation* are retained as entities, while the uncertainty levels are reassigned as *relation labels* linked to the observation entity. Consequently, the counts in Table A.3 may not exactly match those in the original papers, since all datasets are reorganized under a single consistent criterion to enable direct side-by-side comparison.

Table A.3: Comparison of schema coverage across major structured-report datasets. Numbers indicate the count of categories after re-aligning each schema under a unified NER/RE schema. **Entities** represent the number of core concept types defined in each dataset. **Entity-Entity relations** are semantic links between two entities (e.g., *located at*, *associate*, *evidence*). **Entity-Attribute relations** describe properties attached to a single entity (e.g., *status*, *certainty*, *location*, *severity*, *morphology*). **Sequential relation** indicates whether temporal continuity across multiple reports is explicitly modeled. Because each dataset originally adopted its own definitions, the numbers here may not match the original papers exactly.

| Dataset | #Entities | #Entity-Entity Relations | #Entity-Attribute Relations | Sequential Relation |
|---|---|---|---|---|
| RadGraph | 2 | 3 | 2 | × |
| RadGraph-XL | 2 | 3 | 3 | × |
| RadGraph-2 | 3 | 3 | 9 | × |
| Rate-NER | 3 | 0 | 1 | × |
| CAD-Chest | 1 | 0 | 4 | × |
| **LUNGUAGESCORE** | **6** | **2** | **16** | ✓ |

### (1) Entity-level diagnostic source

Existing structured-report datasets define only a limited range of entity types and do not explicitly distinguish whether a concept is directly inferable from chest radiographs. For example, RadGraph (Jain et al. (2021)) and RadGraph-XL (Delbrouck et al. (2024)) include only *anatomy* and *observation*, while RadGraph-2 (Khanna et al. (2023)) adds *device*. Rate-NER (Zhao et al. (2024)) defines *anatomy*, *abnormality*, and *disease*, whereas CAD-Chest (Zhang et al. (2023)) reduces coverage to a single entity type (*disease*).

This design mixes image-grounded findings (e.g., opacity, atelectasis) with contextual or diagnostic terms (e.g., pneumonia, heart failure). As a result, models may hallucinate unsupported content or be penalized unfairly when generating clinically valid but context-dependent descriptors.

Our schema introduces six categories with an explicit separation by visual inferability: **PF**, **CF**, and **OTH** represent image-grounded entities, while **COF**, **NCD**, and **PATIENT INFO** capture information that cannot be reliably inferred from the image itself. This distinction improves training relevance, enables fairer evaluation, and reduces hallucination risk.

### (2) Relation-level semantic precision

Relation definitions in prior datasets are generally coarse. RadGraph variants restrict entity–entity links to three types (*modify*, *located at*, *suggestive of*) and entity–attribute links to *status* and *certainty* (later including *measurement*). Rate-NER and CAD-Chest define only one to four relation categories in total.

Such limited taxonomies conflate distinct clinical reasoning cues. For instance, RadGraph's *suggestive of* combines associative reasoning (e.g., "right lung opacity may be nipple shadow") with evidential claims (e.g., "opacity suggests pneumonia"), which are clinically distinct. Temporal descriptors such

as "improved," "worsened," or "no change" are also collapsed into generic status labels, obscuring clinically meaningful differences.

Our schema expands this design to **18 relation labels** that provide more fine-grained semantic distinctions. This includes entity–entity relations such as *associate* and *evidence*, as well as a rich set of entity–attribute relations covering *status*, *certainty*, *severity*, *location*, *morphology*, *measurement*, *onset*, *improved*, *worsened*, *no change*, *placement*, *past history*, *other source*, and *assessment limitation*. This expanded taxonomy enables more precise error analysis and ensures that distinct error types (e.g., incorrect severity vs. incorrect negation) are penalized appropriately.

### (3) PATIENT-LEVEL LONGITUDINAL LINKAGE

All prior structured-report datasets treat each report in isolation, preventing validation of temporal descriptors or consistency across multiple studies. They cannot verify whether a reported improvement aligns with prior findings, nor can they unify lexical variants (e.g., "opacity" vs. "consolidation") across timepoints.

Our schema explicitly incorporates longitudinal structure through two constructs: **ENTITYGROUPS** and **TEMPORALGROUPS**. **ENTITYGROUPS** unify lexically different mentions of the same clinical finding across reports of a patient, ensuring consistent recognition of semantically equivalent terms. **TEMPORALGROUPS** segment patient trajectories into clinical episodes, reflecting disease progression, resolution, or chronic persistence.

Together, these mechanisms allow evaluation to assess whether generated findings remain temporally coherent across multiple visits, aligning with radiological practice where longitudinal comparison is central. By introducing sequential linkage, our schema enables structured evaluation of longitudinal reasoning—an ability absent from prior resources.

## B TWO-STAGE STRUCTURING FRAMEWORK DETAILS

### B.1 SINGLE STRUCTURING PROMPT

**Prompt Template for Single Structuring**

```
You are a high-precision relation-extraction engine for chest X-ray report sections.
Given a structured input, extract clinical relations between entities while strictly
conforming to the provided schema and labeling rules.

Your task:
- Identify valid entity pairs and annotate appropriate relation types between them.
- Assign Cat, Dx_Status, and Dx_Certainty labels to subject entities.
- Use the provided "candidates" field to guide your extraction and ensure spelling/casing consistency.
- For each identified relation, include:
  - subject_ent: unique index of subject entity
  - subject_cat: entity type
  - obj_ent_idx: unique index of related object
  - relation: relation type (must be one of the allowed relations)
  - sent_idx: sentence index from which the relation is derived
- Output a JSON object that conforms to the "PyraDict StructuredOutput" schema.
- Do not return natural language commentary or raw triples.

Input JSON format:
{
  "report_sections": [
    {
      "sent_idx": 1,
      "sentence": "Findings suggest possible pneumonia in the right lower lobe with opacity.",
      "candidates": [
        ["Pneumonia", ["Entity1"]],
        ["Findings", ["Entity1"]],
        ["Right lower lobe", ["Location1"]],
        ["Opacity", ["Entity1"]]
      ]
    },
    {
      "sent_idx": 2,
      "sentence": "A new small pleural effusion is seen on the left side.",
      "candidates": [
        ["Pleural effusion", ["Entity1"]],
        ["Left side", ["Location1"]],
        ["New", ["None"]],
        ["Small", ["Measurement1"]]
      ]
    }
  ]
}

Allowed Relation Types:
- Cat, Status, Location, Placement, Associate, Evidence,
  Morphology, Distribution, Measurement, Severity, Comparison,
  Onset, No Change, Improved, Worsened, Past Hx, Other Source, Assessment Limitations

Labeling Rules Summary:
- Every subject entity must be assigned exactly one of: Cat, Dx_Status, Dx_Certainty.
- Placement is used only for spatial position.
- Placement is disallowed for devices (Cat = OTH).
- Evidence relations must point from diagnoses to radiological findings.
- Attribute relations must be explicitly stated in the sentence.

Output:
- A list of structured entries containing entity and relation annotations.
- Output must be a single valid JSON object.
- Include only entities mentioned in the text but not in the candidates, following the ent_idx order of
    appearance.

For example:
Input: <related_example>
Output: <structured_report>.
```

### B.1.1 VOCABULARY MATCHING ALGORITHM

To improve consistency in entity extraction and reduce hallucinations in schema-based structuring, we implemented a vocabulary-guided span matching algorithm (see Appendix A.2 for details on vocabulary construction). This algorithm processes each section of the radiology report (e.g., findings) to identify candidate entity spans by directly matching contiguous token sequences against entries in a schema-defined vocabulary, without normalization such as lowercasing or punctuation removal. Each sentence is evaluated independently, and multiple overlapping matches are retained—e.g., "left lung" may correspond to both PF and LOCATION.

Importantly, the matched vocabulary spans are not assumed to constitute a complete or authoritative set of entities. Instead, they serve as reference cues for the LLM, which remains responsible for the final relation extraction. The LLM is expected to leverage the matched terms as guidance while retaining the flexibility to identify additional entities or values not covered by the vocabulary. This design accommodates incompleteness in the vocabulary and enables the model to make context-sensitive inferences based on both the prompt and observed patterns in the data.

The matching algorithm is summarized below:

---

**Algorithm 1** Span-Based Vocabulary Matching

---

1: **Input:** Curated vocabulary $V$; report section $T$ composed of multiple sentences.
2: **Output:** List of matched word spans in $T$, each labeled with one or more schema categories.
3: Build a dictionary $V_{\text{lookup}}$ from surface forms in $V$, mapping each to one or more associated schema categories.
4: **for** each sentence $s$ in $T$ **do**
5:    Split $s$ into a sequence of $n$ words, each with character-level start and end offsets
6:    **for** span length $l$ from $n$ down to 1 **do**
7:      **for** start index $i = 0$ to $n - l$ **do**
8:        Extract word span $s_{i:i+l}$ and its character range from original sentence
9:        Query $V_{\text{lookup}}$ for exact match of the word span
10:       **if** match found **then**
11:         **for** each schema category linked to the matched term **do**
12:          Record span text, character start/end indices, matched term, and category
13:         **end for**
14:       **end if**
15:      **end for**
16:    **end for**
17: **end for**
18: **return** List of matched spans with associated categories

---

This procedure constrains entity recognition to schema-aligned expressions, allowing the LLM to focus on inferring relational structure rather than determining precise span boundaries. By anchoring extraction to predefined lexical targets, it reduces ambiguity and ensures consistent treatment of clinically equivalent yet lexically variable expressions.

## B.2 SEQUENTIAL STRUCTURING PROMPT

**Prompt Template for Sequential Structuring**

```
You are an expert radiologist specializing in chest X-ray interpretation.
Your task is to normalize and properly group sequential CXR findings through a systematic three-step
    approach.

## TASK OVERVIEW - THREE-STEP ANALYSIS

1) GROUPING ANALYSIS (TERMINOLOGY MATCHING)
   - Purpose: Identify when different terminology describes the same underlying radiological entity.
   - Key question: "Do these terms represent the same radiological entity described differently?"

2) STATUS ANALYSIS (NORMAL/ABNORMAL DISTINCTION)
   - Purpose: Separate normal findings from abnormal findings within the same group.
   - Key question: "Is this finding normal (negative) or abnormal (positive)?"

3) EPISODE ANALYSIS (TIME INTERVAL ASSESSMENT)
   - Purpose: Determine if grouped findings occur within the same clinical episode based on time.
   - Key question: "Do these findings represent the same episode of clinical care?"

---

Grouping Criteria:
- Group together when:
  - Terminological variants are used (e.g., "opacity" = "consolidation").
  - Size or progression is described (e.g., "small effusion" ~ "resolving effusion").
  - Locations are adjacent or overlapping.
  - The same device is observed across time.

- Separate when:
  - Descriptive vs. diagnostic terms differ (e.g., "opacity" vs "pneumonia").
  - Anatomical locations or laterality differ.
  - Pathologies are distinct.
  - Different devices are involved.

---

Episode Criteria:
- Normal findings: one episode regardless of interval.
- Abnormal findings: split by resolution or long time gaps.
- Devices: one episode unless explicitly removed and reinserted.
- Symptoms: each occurrence is treated as a new episode unless continuity is stated.

---

## OUTPUT FORMAT:
Provide your analysis in this JSON format:
{
  "results": [
    {
      "group_name": "<Entity + Location + temporal descriptors>",
      "findings": [
        { "IDX": <number>, "DAY": <number>, "finding": "<description>" }
      ],
      "episodes": [
        { "episode_1": { "days": [<number>, <number>, ...] } },
        { "episode_2": { "days": [<number>, ...] } }
      ],
      "rationale": "<Concise explanation of grouping decisions>"
    }
  ]
}

---

IMPORTANT NOTES:
- Every finding must be included in exactly one group.
- Use terminology that appears in the findings.
- Name each group using "<Entity + Location + temporal descriptor>" format (e.g., "Effusion left improving",
    "Nodule right upper lobe worsening").
- For temporal descriptors, use the most recent or predominant qualifier (e.g., "improving", "worsened", "
    stable", "resolved").
- If no temporal descriptor is available in the findings, omit it from the group name.
```

## B.3 SINGLE STRUCTURING ANALYSIS

Table B.1: Ablation results of GPT-4.1 under varying prompt-shot configurations and vocabulary matching. We report precision (P), recall (R), and F1 scores for both entity-relation pair extraction and complete triplet extraction tasks.

| Shot | Vocab Usage | entity-relation | | | entity-relation-attribute | | |
|------|-------------|------|------|------|------|------|------|
| | | F1 | P | R | F1 | P | R |
| Zero | No | 0.79 | 0.65 | **1.00** | 0.52 | 0.65 | 0.44 |
| | Yes | 0.92 | 0.85 | **1.00** | 0.78 | 0.80 | 0.77 |
| 5-shot | No | 0.93 | 0.87 | **1.00** | 0.84 | 0.85 | 0.83 |
| | Yes | 0.94 | 0.89 | **1.00** | 0.87 | 0.87 | 0.86 |
| 10-shot | No | 0.94 | 0.88 | **1.00** | 0.86 | 0.86 | 0.85 |
| | Yes | **0.96** | **0.91** | **1.00** | **0.89** | **0.90** | **0.87** |

We conducted an ablation study to quantify the individual and combined effects of vocabulary matching and in-context demonstrations on single-report structuring. Using 80 radiology reports from 30 patients, previously annotated for sequential evaluation, this subset enabled consistent evaluation across controlled input conditions.

Six configurations were tested by varying two factors: (1) whether span-to-category alignment via vocabulary matching was applied, and (2) the number of in-context examples provided in the prompt (0, 5, or 10). Vocabulary matching involved matching contiguous text spans against a predefined lexicon and retrieving all associated schema categories, ensuring lexical consistency and reducing ambiguity in span interpretation, as described in Appendix B.1.1. In-context demonstrations consisted of structured examples retrieved from the gold set of structured reports using BM25 retrieval, based on textual similarity to the input report. These examples illustrate appropriate usage of entity types and relations under the schema.

As shown in Table B.1, vocabulary matching consistently enhanced performance across all prompt configurations. Under the zero-shot setting, incorporating vocabulary guidance raised the triplet-level F1 score from 0.52 to 0.78, and the entity-relation F1 from 0.79 to 0.92. When five in-context demonstrations were provided, the triplet F1 increased further—reaching 0.84 without vocabulary and 0.87 with vocabulary. The highest accuracy was achieved by combining both components: the 10-shot setting with vocabulary matching attained a triplet F1 of 0.89.

These results indicate that vocabulary matching and in-context demonstrations offer complementary benefits. Vocabulary alignment improves lexical grounding and category consistency, while prompting with examples strengthens structural fidelity across varying linguistic expressions. Together, they establish a robust configuration for producing schema-compliant structured outputs from free-text radiology reports.

To illustrate the qualitative impact of vocabulary matching and prompt-based demonstrations, we examined example outputs across configurations with and without these components. In the sentence *"there is no focal consolidation"*, the model without vocabulary and prompt guidance extracted *"focal consolidation"* as the entity, conflating the modifier and the core clinical concept. In contrast, all other configurations correctly identified *"consolidation"* as the schema-aligned entity. A similar pattern was observed in *"there are no new focal opacities concerning for pneumonia"*, where the no-guidance setup extracted *"focal opacities"*, whereas guided configurations yielded the correct entity *"opacities"*.

These examples underscore the importance of explicitly aligning model outputs to a predefined schema. Linguistically valid but structurally inconsistent extractions can hinder downstream applications, where precise interpretation and reliable information linkage are essential. By providing lexical anchoring through vocabulary and structural demonstrations via prompts, our approach ensures that model predictions are not only accurate but also semantically coherent and clinically usable.

### B.4 Sequential Structuring Analysis

We qualitatively evaluated model behavior in the sequential setting by analyzing entity grouping outputs over time. Using longitudinal chest X-ray reports from representative patients, we assessed how well the predicted entity groupings aligned with gold-standard annotations. As illustrated in Figure B.1, we examined diverse cases to understand temporal consistency and grouping granularity.

In general, clinical observations were consistently grouped across both annotations. For instance, in Patient p15881535, three lexical variants—*orthopedic side plate right clavicular unchanged*, *right clavicle hardware*, and *internal fixation hardware*—were all correctly assigned to the same entity group in both the gold standard and the model output. Although the representative phrase differed, the group identity was preserved, indicating successful recognition of referential equivalence across timepoints.

Discrepancies primarily arose from differences in granularity rather than semantic errors. In the case of Patient p15881535, temporally separated mentions of *pneumonia* were grouped together in the gold annotations but split into separate groups in the model output. Rather than a failure to track entities, this divergence suggests the model applies a stricter granularity, distinguishing findings based on specific attributes (e.g., diagnostic status or precise location) where human annotators might merge them. Similarly, regarding opacity in the right cardiophrenic sulcus, the model separated instances based on their evolving descriptions (e.g., "resolving"), prioritizing attribute precision over broad grouping.

Conversely, the model demonstrated the capability for semantic unification where appropriate. As seen in Patient p18517718, the model successfully reduced redundancy found in human annotations. For example, while the gold standard labeled specific attributes of a single medical device (e.g., feeding tube, tip location) as separate groups, the model unified them into a single coherent entity. This highlights the model's ability to identify core clinical concepts and organize fragmented descriptions effectively.

Overall, despite these variations in granularity, the grouping performance remained robust. The model preserved the essential semantic structure, balancing fine-grained distinctions for evolving pathologies with the integration of redundant observations. These findings support the reliability of our sequential annotation approach for tracking clinically meaningful entities over longitudinal report timelines.

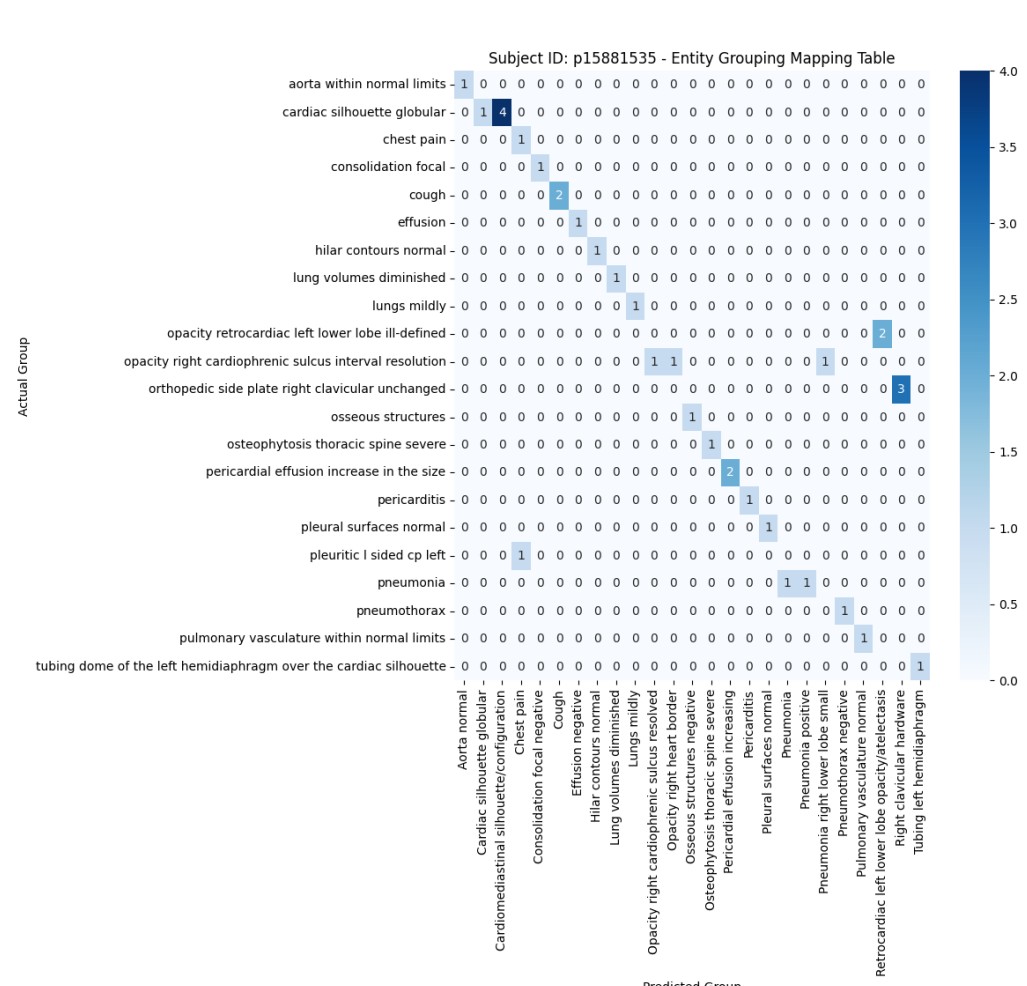

(a) Patient p15881535: Gold standard 22 groups vs Model 25 groups. Overall, the model aligns closely with the human gold standard. The slight difference in count stems from the model making more fine-grained distinctions—such as separating a diagnosis from its specific location (e.g., in Pneumonia or Cardiac Silhouette)—whereas human annotators tended to group them. These variations reflect a rigorous interpretation of the criteria while maintaining full semantic consistency.

Figure B.1: Entity grouping results for three sample patients based on sequential chest X-ray reports. The figures compare human-annotated gold-standard groupings (rows) with GPT-4.1 model predictions (columns). Numbered cells represent individual findings. Despite slight wording variations, the model demonstrates strong adherence to semantic grouping criteria. (Continued on next page)

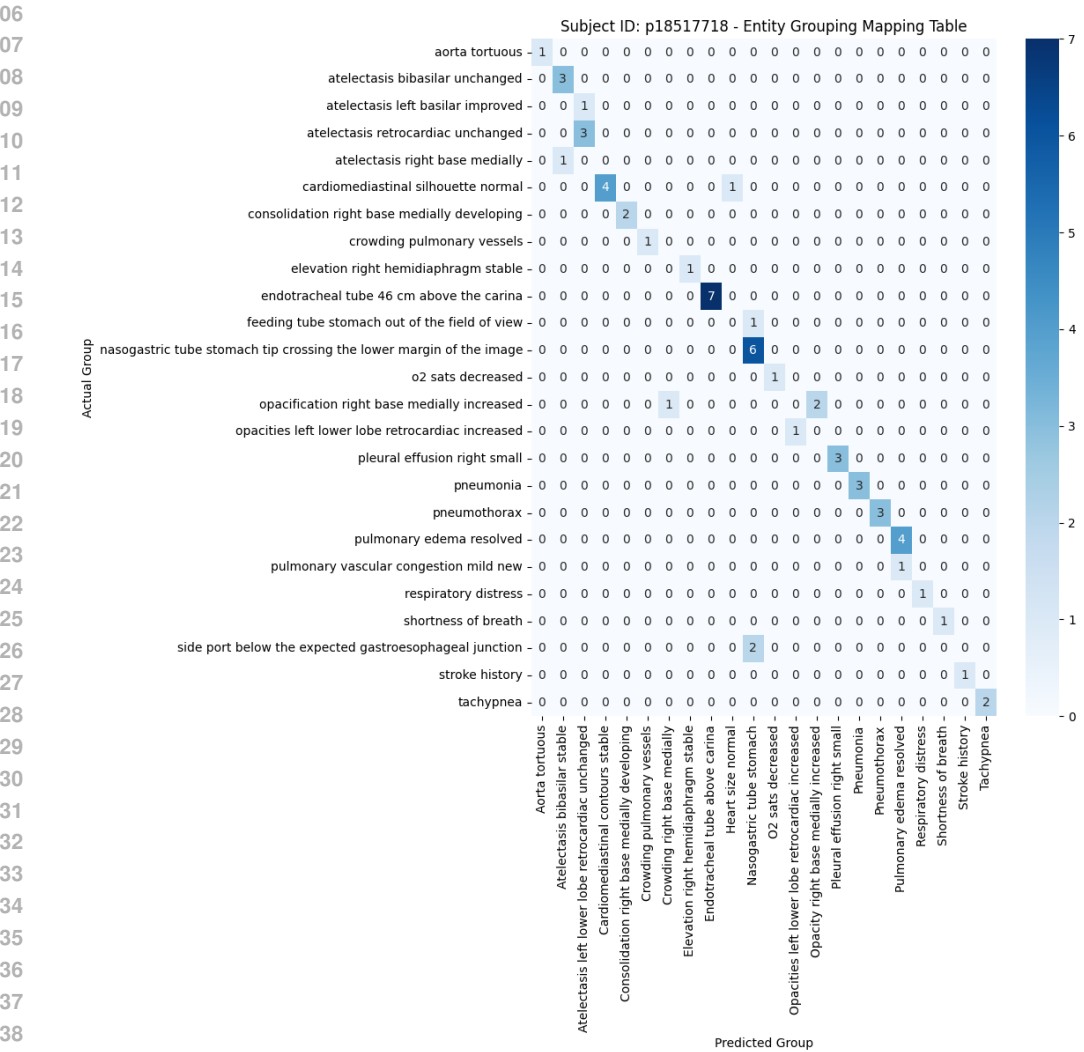

(b) Patient p18517718: Gold standard 25 groups vs Model 22 groups. While the model achieves high overall alignment with the gold standard, subtle discrepancies arise due to differences in granularity levels regarding semantically overlapping entities. In contrast to the previous case, where human annotators separated specific attributes of a single medical device (e.g., feeding tube, tip location, side port), the model interpreted them as components of a single coherent entity ('Nasogastric tube stomach'). This illustrates that mismatches often result from legitimate variations in how granularly overlapping concepts are defined, rather than semantic errors.

Figure B.1: Continued.

## C LUNGUAGESCORE DETAILS

### C.1 ATTRIBUTE WEIGHTS OF LUNGUAGESCORE

To reflect the clinical importance of structured attributes in radiology reports, LUNGUAGESCORE applies attribute-specific weights when measuring similarity between predicted and reference structures. Each comparison is performed at the level of relational triplets, jointly assessing both temporal and structural alignment. For structural attributes, we assign weights based on expert consensus from the four board-certified radiologists who participated in the data annotation process, reflecting each attribute's diagnostic significance. Although the initial weights are unnormalized, they are rescaled such that their total contribution sums to 1.0 during evaluation (see Table C.1).

For the sequential setting, temporal alignment contributes a fixed weight of 1.0, divided equally between two components: whether the predicted and reference findings belong to the same study timepoint (0.5), and whether they fall within the same temporal group (0.5).

Although our schema includes inferential relations such as ASSOCIATE and EVIDENCE, these are intentionally excluded from the evaluation metric. Such relations capture diagnostic reasoning—e.g., linking "opacity" as supporting evidence for "pneumonia"—but do not directly reflect the correctness of factual information. Scoring them would conflate interpretive inference with structural accuracy. Instead, our metric focuses on clinically grounded descriptors and attributes that define the diagnostic content of the report. Future extensions may consider integrating reasoning-based relations in settings that explicitly target causal or explanatory fidelity.

Table C.1: Weights used in LUNGUAGESCORE for evaluating structural similarity. Temporal weights apply only in the sequential setting, while structural attribute weights reflect the diagnostic importance of each relation type. All values are normalized such that their respective groups (temporal or structural) sum to 1.0 during evaluation.

| Structural Attribute Weights | Value |
| --- | --- |
| DXSTATUS | 0.50 |
| DXCERTAINTY | 0.10 |
| LOCATION | 0.20 |
| SEVERITY | 0.15 |
| ONSET | 0.15 |
| IMPROVED | 0.15 |
| WORSENED | 0.15 |
| PLACEMENT | 0.15 |
| NO CHANGE | 0.10 |
| MORPHOLOGY | 0.05 |
| DISTRIBUTION | 0.05 |
| MEASUREMENT | 0.05 |
| COMPARISON | 0.03 |
| PAST HX | 0.01 |
| OTHER SOURCE | 0.01 |
| ASSESSMENT LIMITATIONS | 0.01 |

| Temporal Weights | Value |
| --- | --- |
| Study Timepoint | 0.5 |
| Temporal Group | 0.5 |

### C.2 LUNGUAGESCORE EXAMPLES

**Single-Report Assessment**  To illustrate how LUNGUAGESCORE evaluates structured prediction quality in the single-report setting, we present detailed examples of pairwise comparisons between predicted and gold-standard structured reports. As detailed in Section 5 in the main text, each comparison is decomposed into two complementary components:

- **Semantic Score:** Computed as the cosine similarity between embedded linearized entity phrases. These phrases are formed by concatenating free-text attributes, including LOCATION, MORPHOLOGY, DISTRIBUTION, MEASUREMENT, SEVERITY, ONSET, IMPROVED, WORSENED, NO CHANGE, and PLACEMENT. This representation captures the semantic

content of the entity and its descriptive qualifiers, allowing similarity to be measured in an integrated manner.

- **Structural Score:** A weighted sum of attribute-wise comparisons. Categorical attributes (DxStatus and DxCertainty) are scored in binary fashion (1.0 for exact match, 0.0 otherwise), while all other attributes are evaluated via cosine similarity of their embeddings. The relative importance of each attribute is determined by expert-defined weights (see Table C.1).

The final similarity between a predicted and reference finding is calculated as the product of the semantic and structural scores:

$$\text{Total Score} = \text{Semantic Score} \times \text{Structural Score}$$

**Note:** Entity refers to the linearized phrase comprising the core entity and its attributes. Avg. Cosine indicates cosine similarity averaged over MedCPTJin et al. (2023) and BioLORD23Remy et al. (2024) embeddings of the phrases. Weights shown in the table reflect unnormalized values; the final Structural Score is computed by normalizing the weighted sum by the total weight of all included attributes. For a more formal explanation of the scoring method, we refer to Section 5 in the main text.

**Example 1: Moderate Match with Attribute-Level Divergence**

| Attribute | GT Value | Pred Value | Match Type | Score | Weight |
|---|---|---|---|---|---|
| Entity | effusions bilateral small | pleural effusion left-sided pleural small stable | Avg. Cosine | 0.743 | — |
| DxStatus | positive | positive | Exact match | 1.00 | 0.50 |
| DxCertainty | definitive | definitive | Exact match | 1.00 | 0.10 |
| Location | bilateral | left-sided pleural | Avg. Cosine | 0.54 | 0.20 |
| Severity | small | small | Exact match | 1.00 | 0.15 |
| Improved | — | stable | Avg. Cosine | 0.00 | 0.15 |

Semantic Score = 0.743,  Structural Score = 0.681,  Total Score = **0.506**

**Example 2: Partial Match with Location and Severity Differences**

| Attribute | GT Value | Pred Value | Match Type | Score | Weight |
|---|---|---|---|---|---|
| Entity | opacification left retrocardiac | pleural effusion left moderate | Avg. Cosine | 0.447 | — |
| DxStatus | positive | positive | Exact match | 1.00 | 0.50 |
| DxCertainty | definitive | definitive | Exact match | 1.00 | 0.10 |
| Location | left retrocardiac | left | Avg. Cosine | 0.60 | 0.20 |
| Severity | — | moderate | Avg. Cosine | 0.00 | 0.15 |

Semantic Score = 0.447,  Structural Score = 0.758,  Total Score = **0.339**

**Example 3: Strong Match with Minor Lexical Variants**

| Attribute | GT Value | Pred Value | Match Type | Score | Weight |
|---|---|---|---|---|---|
| Entity | opacity right lung base | opacity right lower lung base stable | Avg. Cosine | 0.842 | — |
| DxStatus | positive | positive | Exact match | 1.00 | 0.50 |
| DxCertainty | definitive | definitive | Exact match | 1.00 | 0.10 |
| Location | right lung base | right lower lung base | Avg. Cosine | 0.95 | 0.20 |
| Improved | — | stable | Avg. Cosine | 0.00 | 0.15 |

Semantic Score = 0.842,  Structural Score = 0.902,  Total Score = **0.759**

**Sequential-Report Assessment**    To clarify how LunguageScore computes similarity in the sequential setting, we present illustrative examples comparing gold-standard and predicted findings. Each score is computed from three components:

- **Semantic Score**: In the sequential-report setting, semantic similarity is computed between *ENTITYGROUP* representations, which group together lexically variable but conceptually equivalent findings observed at different timepoints.

- **Temporal Score**: Value of 1.0 if both findings appear in the same study timepoint and in the same TEMPORAL GROUP, or 0.5 if they belong to the same broader TEMPORAL GROUP but from different studies, or vice versa. If neither matches, the score is 0.

- **Structural Score**: Weighted average of attribute-level matches (exact for binary attributes, cosine similarity for textual ones).

The overall similarity score is computed as:

$$\text{Total Score} = \text{Semantic Score} \times \text{Temporal Score} \times \text{Structural Score}$$

Table C.2: Examples of LUNGUAGESCORE computations in the sequential setting. Each row compares a predicted finding against the corresponding ground-truth reference. Total Score is computed as the product of semantic similarity, temporal alignment, and structural accuracy. **Time** denotes the study timepoint, and **TG** indicates the assigned temporal group.

| | GT | | | Prediction | | | Explanation | Total (Sem × Temp × Str) |
|---|---|---|---|---|---|---|---|---|
| Case | EntityGroup | Time | TG | EntityGroup | Time | TG | | |
| 1 | pleural effusion subpulmonic moderate | 2 | 1 | pleural effusion right subpulmonic layering moderate stable | 2 | 1 | Minor semantic variation in anatomical modifiers and progression terms | 0.68 (0.82 × 1.0 × 0.83) |
| 2 | hilar contours stable | 3 | 1 | hilar contours unchanged | 3 | 1 | Semantically equivalent; lexical variation in stability descriptor | 0.90 (0.93 × 1.0 × 0.97) |
| 3 | atelectasis left lower lobe mild-to-moderate | 1 | 1 | atelectasis left lower lobe unchanged | 2 | 1 | Different timepoints (0.5), severity term vs. stability term mismatch | 0.35 (0.92 × 0.50 × 0.76) |
| 4 | PICC mid SVC | 2 | 1 | left PICC mid SVC | 1 | 1 | Core entity match with modifier discrepancy; higher specificity in prediction; different timepoints | 0.45 (0.90 × 0.50 × 1.00) |
| 5 | hilar contours unchanged | 2 | 1 | cardiomediastinal silhouette unchanged | 3 | 1 | Semantically related anatomical terms; timepoint mismatch (0.5) | 0.34 (0.68 × 0.50 × 1.00) |

**Final Scoring and Interpretability**  LUNGUAGESCORE calculates a TOTAL SCORE for each matched pair of predicted and reference findings by combining semantic similarity and structural alignment. In the single-report setting, the total score is defined as the product of cosine similarity over linearized entity phrases and a weighted score of attribute-level matches. In the sequential setting, the metric further incorporates a temporal alignment factor, distinguishing between exact study-time matches and broader temporal group continuity.

These component-wise scores are then aggregated across matched pairs to compute the overall F1 metric, as detailed in Section 5. Crucially, each comparison yields interpretable diagnostics: the semantic score quantifies lexical alignment of free-text descriptors; the structural score exposes attribute-level agreement or divergence; and in longitudinal contexts, the temporal score reveals whether grouping decisions respect continuity over time.

By exposing this granularity, LUNGUAGESCORE not only delivers a robust scalar evaluation, but also supports nuanced error analysis—highlighting which components of a model's output (e.g., misassigned severity, incorrect timing, lexical drift) most strongly influenced final performance. This interpretability makes the metric especially valuable to understand model's behavior.

## C.3 CLINICAL BERT MODEL SELECTION

We considered multiple clinical BERT models for computing contextual semantic embeddings. The candidate models we compared were BioLORD (Remy et al. (2024)), BiomedBERT (Gu et al. (2020)), MedCPT (Jin et al. (2023)), BioClinicalBERT (Alsentzer et al. (2019)), ClinicalBERT (Liu et al. (2025)) and BioBERT (Lee et al. (2020)). To decide which models to use in the semantic similarity step of LUNGUAGESCORE, we conducted an experiment over ReXVal, a subset of the MIMIC-CXR test set encompassing 50 randomly selected studies. We structured each individual study according to our framework described in Section 4(i), and then generated all linearized phrases derived from entity–location–attribute triplets for both the reference report and the candidate report. We then used each candidate BERT embedding model to generate an embedding for each phrase, and computed the pairwise cosine similarity for all pairs of phrases (one from the reference report and one from the candidate report). Figure C.1 shows the distribution of this similarity score for the different BERT embedding models. We find that BiomedBERT, BioClinicalBERT, ClinicalBERT and BioBERT lack variety, always scoring pairs of phrases as highly related. BioLORD manages to capture the most diversity in semantic similarity, followed by MedCPT. For this reason, we choose to use both BioLORD and MedCPT to calculate semantic similarity, by taking the average over both models.

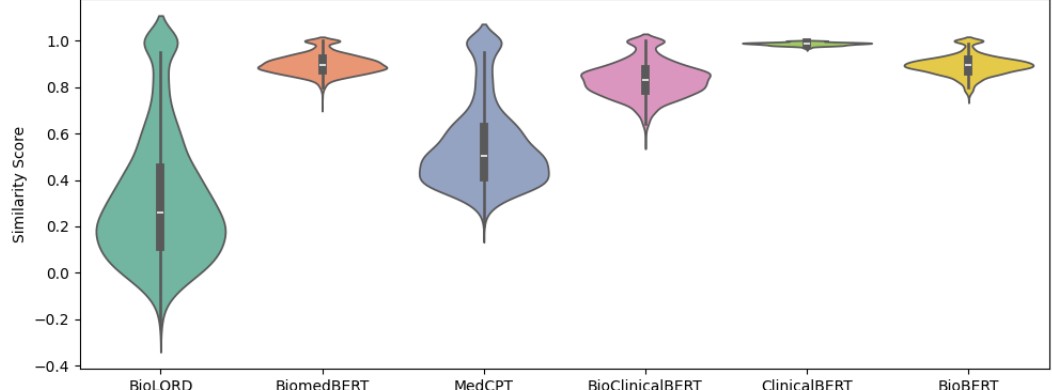

Figure C.1: Distribution of pairwise cosine similarity scores for different BERT embedding models, calculated between pairs of embedded linearized phrases taken from the ReXVal datset.

## D   METRIC VALIDATION

**Metric Implementation Details**   Whenever not further specified, we used default settings for all the metrics as provided by their respective libraries. For BLEU, we use the implementation provided in the `huggingface/evaluate` library. For BERTScore, we also use the implementation from the `huggingface/evaluate` library, with `distilroberta-base` as an embedding model. For GREEN, we use `StanfordAIMI/GREEN-radllama2-7b` as a language model. For FineRadScore, we use `GPT-4` as a language model, which responds with a list of errors each linked to a severity level. To turn this into a score, we associate each severity level with a number, and sum these scores, forming FineRadScore as proposed by (Huang et al. (2024)). In our tables, we report 1/FineRadScore, inverting the total sum to ensure that a higher score is associated with higher quality. For RaTEScore, we use their default weight matrix. Note that in their own comparison with ReXVal, the authors used a custom weight matrix trained specifically for long reports instead of the default, explaining the slight discrepancy between their reported Kendall Tau correlation with ReXVal radiologists and the one we report in Table 2. We also report results for RadGraph and its newer version RadGraph-XL, using the RG_EG setting to calculate the F1 score as proposed in (Delbrouck et al. (2022)).

**ReXVal Analysis**   To assess the consistency of our metric with established evaluation standards, we conducted a correlation analysis across the ReXVal benchmark, which includes expert-annotated radiology reports and associated error counts. Specifically, we computed pairwise Pearson correlations between all single-report metrics over the ReXVal dataset. As presented in Figure D.1, our metric exhibits strong positive correlations with BLEU (0.73), BERTScore (0.77), GREEN (0.84), RaTEScore (0.77), 1/FineRadScore (0.73), RadGraph F1 (0.80) and RadGraph-XL F1 (0.80). Notably, among all evaluated metrics, our score achieves the highest average correlation across all pairwise comparisons, indicating strong alignment with multiple evaluation perspectives and suggesting broader generalizability.

Furthermore, Figure D.2 illustrates the linear relationship between each metric and the number of radiologist-identified errors per ReXVal report. Although 1/FineRadScore shows the highest overall correlation, its relationship with error counts is not consistently linear, especially when the number of errors is low. In these cases where distinguishing between high-quality outputs is most crucial, its ability to make fine-grained distinctions is limited. In contrast, our metric not only maintains strong correlation but also demonstrates stable linear responsiveness across the full error range, underscoring its robustness and reliability as a clinically aligned evaluation measure.

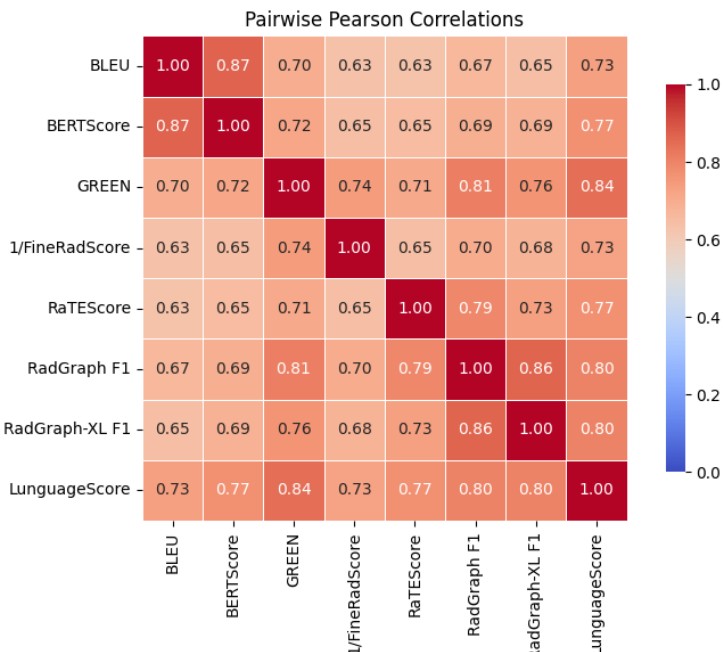

Figure D.1: Pairwise Pearson correlations between our metric (LUNGUAGESCORE), and the metrics BLEU, BERTScore, GREEN, 1/FineRadScore, RaTEScore, RadGraph F1 and RadGraph-XL F1.

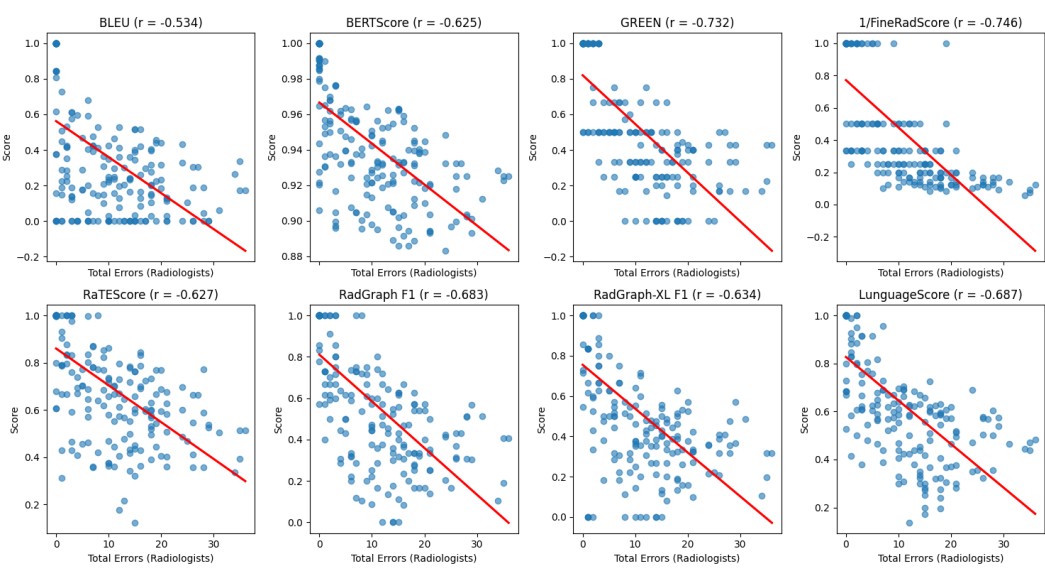

Figure D.2: Scatter plot illustrating the correlation between the total number of errors identified by radiologists per report, and each of the single-report metrics, including our LUNGUAGESCORE. $r$ indicates the Pearson correlation as reported in Table 2.

**Error Sensitivity Analysis with ReXErr (Rao et al. (2024))**   To assess the error sensitivity of our metric across diverse failure types in radiology report generation, we use the ReXErr-v1 dataset (Rao et al. (2025)), which contains synthetic reports with systematically injected clinical errors. These errors are categorized into content addition, context-dependent, and linguistic quality types, covering a broad spectrum of realistic mistakes. We focus on the subset of ReXErr aligned with our sequential structured report dataset, comprising 57 MIMIC-CXR reference reports paired with corresponding error-injected versions. Each manipulated report contains three injected errors, drawn from 12 defined error categories using a context-sensitive sampling method.

For each pair, we extract the Findings and Impression sections and evaluate them independently using our single-report LUNGUAGESCORE, along with established alternatives: GREEN, FineRadScore, and RaTEScore. Figure D.3 displays the score distributions for each of the 12 error types, relative to the average score across the subset. Our metric demonstrates differentiated sensitivity across error types, with notably larger penalizations for false predictions, incorrect negations, and changes in severity—reflecting its alignment with clinically meaningful deviations.

**Sequential Sensitivity Analysis**   We further assessed the sensitivity of LUNGUAGESCORE to clinically meaningful disruptions in temporal coherence by constructing a synthetic evaluation set in which longitudinal progression cues were deliberately inverted. Specifically, we selected 8 patient sequences from our sequential-report dataset that contained explicit temporal descriptors—such as *improved* or *worsened*—and manually reversed these attributes to simulate a contradiction in the clinical trajectory. For example, a statement like "the previously seen right lower lobe opacification has decreased substantially" was changed to "increased substantially," thereby inverting its semantic implication. Two patient sequences that lacked any such temporal expressions were excluded.

Both the single-report and sequential variants of LUNGUAGESCORE were applied to these perturbed sequences. To quantify the metric's responsiveness, we introduce the *Effect Rate*, which captures the average score reduction per flipped attribute:

$$\text{Effect Rate (\%)} = \frac{1 - \text{score}}{\#\text{flipped attributes}} \times 100$$

A perfect score of 1.0 indicates complete semantic and structural agreement with the gold standard. Deviations from this ideal reflect the metric's sensitivity to reversed temporal directionality. The normalization by the number of flipped attributes allows us to measure the per-attribute impact on the similarity score.

Table D.1: Effect Rate for each manipulated patient sequence.  W/I denotes the number of `worsened`/`improved` attributes flipped.

| Patient ID | # Attr. (W/I) | Single Score | Effect Rate (S, %) | Sequential Score | Effect Rate (Seq, %) |
|---|---|---|---|---|---|
| p10274145 | 5 (0/5) | 0.981 | 0.38 | 0.979 | 0.42 |
| p10523725 | 3 (1/2) | 0.989 | 0.37 | 0.987 | 0.43 |
| p10886362 | 8 (5/3) | 0.983 | 0.21 | 0.979 | 0.26 |
| p10959054 | 13 (9/4) | 0.967 | 0.25 | 0.963 | 0.28 |
| p12433421 | 15 (8/7) | 0.968 | 0.21 | 0.971 | 0.19 |
| p15321868 | 2 (1/1) | 0.982 | 0.90 | 0.988 | 0.60 |
| p15881535 | 1 (0/1) | 0.992 | 0.80 | 0.992 | 0.80 |
| p18079481 | 10 (2/8) | 0.976 | 0.24 | 0.980 | 0.20 |

While the absolute Effect Rates are relatively small (typically below 0.5%), they scale proportionally with the number of flipped attributes, indicating that LUNGUAGESCORE reliably captures the semantic impact of trend reversals. Notably, even sequences with a single flipped term exhibited pronounced per-attribute degradation, highlighting the metric's granularity and responsiveness. These results affirm that LUNGUAGESCORE can effectively detect inconsistencies in longitudinal directionality, even when the surface fluency of the report remains intact.

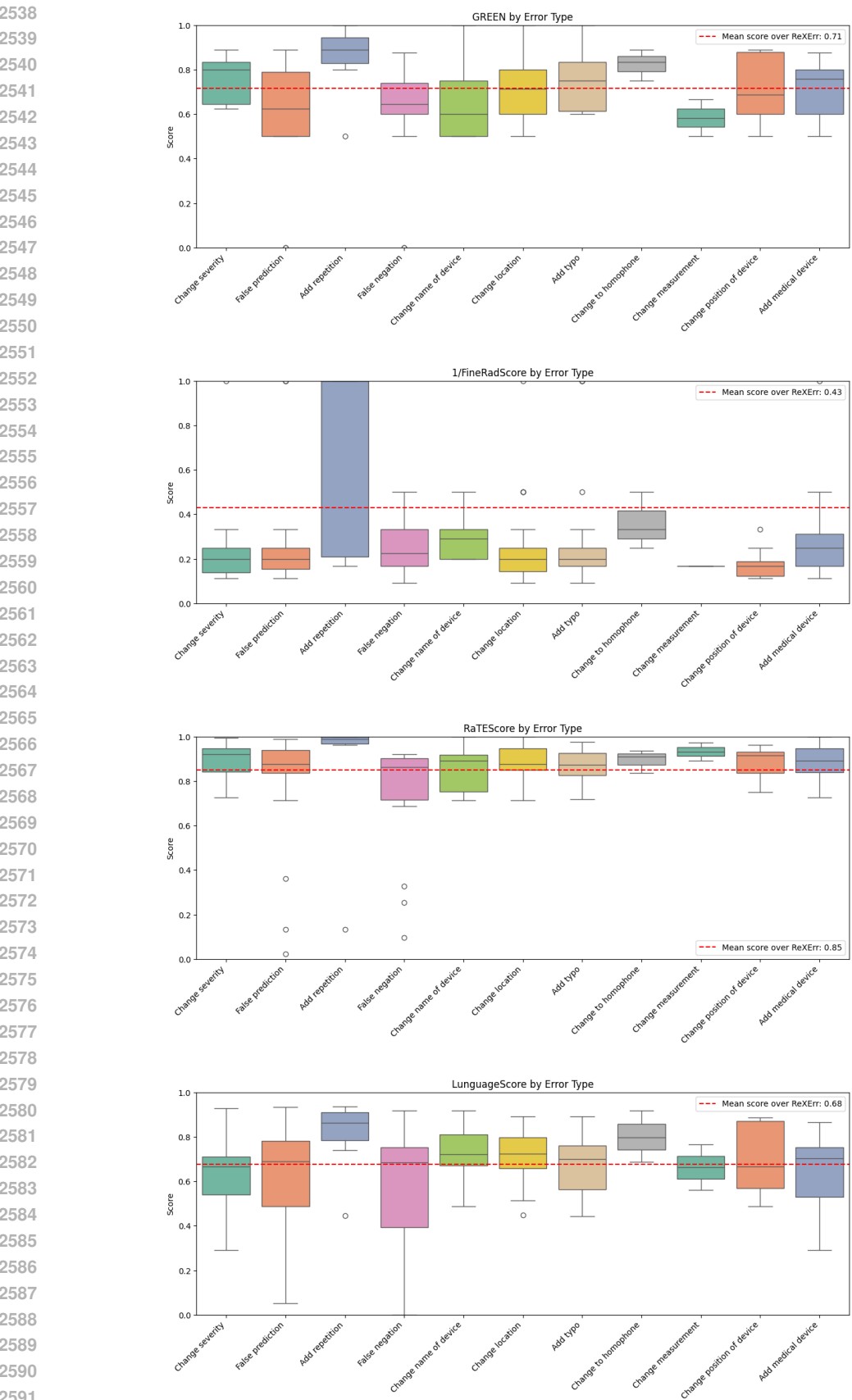

Figure D.3: Distribution of the scores for each of the twelve error types in ReXErr, relative to the average score across the 57 ReXErr reports.

# E SYNTHETIC REPORT GENERATION DETAILS

**MAIRA-2 (Bannur et al. (2024))**    At the input, we feed in a frontal chest X-ray image for the current study. If there is no frontal available for the patient, we do not generate a report. If there are multiple frontals, we randomly choose one. We also pass along a random lateral chest X-ray image for the current study, should it be available. MAIRA-2 additionally accepts the indication, technique and comparison sections. We therefore input the history for the current study in the "indication" field, if there is one. For the comparison, we input "Chest radiography dated _." if there is a previous study, to comply with the anonymised dates in the MIMIC-CXR dataset. We do not input a technique, since this field could not be reliably extracted for the MIMIC-CXR test set. We explore two distinct ways for including prior information in the generation setup. In the **standard** setting, we input the ground truth reference report that is available for the previous study. This report is structured following the template "INDICATION: <prior_history> COMPARISON: <prior_comparison> FINDINGS: <prior_findings> IMPRESSION: <prior_impression>.", where <prior_history>, <prior_impression> and <prior_findings> are all taken from the previous study's ground truth reference report, and substituted by "N/A" if they are missing. If there is no prior study, the prior report field is set to "None". If the previous study was the first one in the sequence, then <prior_comparison> is set to "N/A", otherwise it is set to "Chest radiograph dated _." In the **cascaded** setting, <prior_findings> is set to the findings report that was generated in the previous study (if there is one, otherwise the prior report field is set to "None"), while <prior_impression> is left blank (because MAIRA-2 only generates findings), and the other inputs remain the same. In both settings, we input the frontal view from the prior study, if there is one, and if there are multiple options, we choose the same one that was used to generate the previous report. We ask MAIRA-2 to generate the findings section for the current study, using their default settings, without grounding.

**Medversa (Zhou et al. (2024))**    Next to the current frontal image, we also fill in the additional input fields expected by Medversa, which are context, prompt, modality and task. For context, we follow the template "Age: None. Gender: None. Indication: <current_history>". For <current_history>, we pass along the "history" section of the reference report, should it be available, and otherwise we set it to "None". The modality and task are set to "cxr" and "report generation" respectively. All language generation parameters are left as default. The prompt is set to "Can you provide a report of <img0> with findings and impression?". Note that this is the only model with the ability to generate an impression section, and it will therefore naturally have an advantage over the other models when we compare it to the reference report, where both the findings and impression section are included based on their availability in the ground truth.

**LIBRA (Zhang et al. (2025))**    For LIBRA, a general-purpose medical vision–language model, we use the authors' public implementation in its image–text report generation mode. For each study, we provide the current frontal chest X-ray as the primary input image; if multiple frontal views are available, we randomly select one, and if no frontal is available, we do not generate a report. When a previous study exists, we additionally pass the frontal image from the most recent prior study as a second input so that LIBRA can jointly attend to the current and prior examinations; otherwise, only the current image is used. We use a fixed, generic instruction prompt asking the model to produce a detailed description of the radiographic findings, and keep all decoding hyperparameters at their default values. The resulting text is taken as the model's report for evaluation without further post-processing or templating.

**RGRG (Tanida et al. (2023)), Cvt2DistilGPT2 (Nicolson et al. (2023)), MedGemma (Sellergren et al. (2025)), ChexAgentChen et al. (2024), and Lingshu (Xu et al. (2025))**    For these models, we input the current frontal chest X-ray image, randomly selecting one when multiple views were available and skipping generation when none were present. We used the vision–language variant `google/medgemma-27b-it` for MedGemma and the MIMIC-CXR–trained version with default configuration for Cvt2DistilGPT2. RGRG and Cvt2DistilGPT2 generated only *findings* without a separate *impression* section, whereas MedGemma and Lingshu produced full reports containing both *findings* and *impressions*. The exact prompt templates for MedGemma and Lingshu, as specified in their original papers, are shown below.

---

**Prompt Templates for MedGemma and Lingshu**

```
[MedGemma]
You are an expert radiologist. Please succinctly describe
the findings for the above chest X-ray.

[Lingshu]
You are a helpful assistant. Please generate a report
for the given images, including both findings and impressions.
Return the report in the following format:
Findings: {} Impression: {}
```

---

## F  LIMITATIONS AND FUTURE DIRECTIONS

While LUNGUAGE defines the fine-grained evaluation dataset, the structuring framework produces schema-aligned representations, and LUNGUAGESCORE provides the scoring function for both single-report and longitudinal evaluation, the current work also has several limitations that suggest concrete directions for extension.

First, although LUNGUAGE provides fine-grained entity-, attribute-, and longitudinally interpreted annotations, its patient coverage remains modest: the longitudinal subset currently comprises 30 patients and 186 reports drawn from a single public MIMIC-CXR dataset. Both the dataset and the underlying schema were developed on this subset, so the current vocabulary and relation set may underrepresent findings, reporting conventions, and temporal patterns present in other institutions or populations. To support broader use, future work should scale and stress-test the schema and pipeline on the full MIMIC-CXR dataset as well as other large longitudinal datasets, and develop complementary benchmarks on multi-center, multi-country cohorts and additional imaging modalities. Because the schema, prompts, and implementation are publicly released, researchers can adapt the framework to local reporting conventions, extend the taxonomy, or substitute alternative structuring models while retaining a comparable evaluation protocol. The same tooling can also be used to generate large "silver-standard" structured sets from unlabeled reports, enabling end-to-end pipelines where models are trained and evaluated under a consistent schema, and to derive downstream resources such as QA or instruction-tuning datasets grounded in structured longitudinal trajectories. We hope this work will serve as a starting point for a broader community effort toward clinically grounded, temporally aware evaluation standards for radiology report generation.

As a second limitation and direction for future work, we note that advancing patient-centered reporting will require integrating structured EHR information alongside chest X-rays, including laboratory data, vital signs, procedures, and free-text clinical notes. Current image-based generation approaches struggle with context-rich sections such as patient history, and models that lack access to these contextual signals remain fundamentally limited in longitudinal reasoning and diagnostic continuity. A natural next step is therefore to extend our schema, structuring framework, and LUNGUAGESCORE to multimodal trajectories that couple images, reports, and EHR data, and to examine how the same design principles can be adapted to other imaging domains and healthcare settings. Key challenges for this extension include defining clinically reliable ground truth for multimodal trajectories, aligning heterogeneous temporal signals across modalities, and ensuring that extended versions of LUNGUAGESCORE remain interpretable and robust at EHR scale.

Third, in this work our structuring and evaluation operate purely at the report level. Although the schema explicitly distinguishes image-groundable entities (for example, perceptual findings and devices) from non-chest x-ray findings (for example, clinical history or laboratory results), we do not yet link these entities to the underlying images. An important next step is to ground LUNGUAGE in the pixel space by associating structured findings with spatial annotations, such as view-specific bounding boxes or pixel-level masks for lesions, devices, and other relevant regions. This would enable joint evaluation of whether a generated report is not only semantically and temporally consistent, but also spatially aligned with the visual evidence, and would support the construction of downstream vision–language tasks such as grounding, question answering, and instruction tuning based on the same structured representation.

