# OpenReview forum: "Lunguage: A Benchmark for Structured and Sequential Chest X-ray Interpretation"
_ICLR.cc/2026/Conference — ICLR 2026 Conference Withdrawn Submission_

### Official Review · Reviewer_zLzN · 2025-10-26

**Soundness:** 3
**Presentation:** 3
**Contribution:** 3
**Rating:** 4
**Confidence:** 4

**Summary:**

A benchmark dataset of sequential radiology report is proposed, containing 1,473 reports from 230 patients, including 10 patients with sequential reports. A set of clinical entities and relationships are annotated. A new evaluation metric is proposed for generated report evaluation.

The dataset construction details are presented, but the size of the dataset is small.

**Strengths:**

A benchmark dataset of sequential radiology report is proposed, where many kinds of clinical entities and relationships are annotated. Dataset construction is described with details.

**Weaknesses:**

1. More experimental results are needed for testing the temporal-aware report generation models on the proposed dataset.

2. The benchmark dataset is in small-scale. As the sequential interpretation claimed as the novelty or contributions, only 10 patients in the proposed dataset contains sequential reports (longitudinal trajectories).

**Questions:**

1. In the 3.1, there are several kinds of entities defined. Is this categorization referred from any medical textbook or handbook, or created by yourself? Any domain experts confirm that this category is reasonable? Can an entity be categorized into more than one kind?

2. In the sequential report structure, can one clinical finding in the Day 10 report be associated with more than one clinical finding in the Day 90 report? In addition, is many-to-many association between clinical findings in different sequential reports possible? If possible, how can your proposed schema represent that?

---

> ### Author Response · Authors · 2025-11-22
> **[Weakness 1]**
>
> Thank you for the comment. We added SOTA models, including a temporal-aware baseline, and evaluated them in both single-report and sequential settings. These updates will be reflected in the revised version.
>
> **Additional baselines (CheXAgent and LIBRA)**
> We added two recent chest X-ray report generation models, CheXAgent and LIBRA, as new baselines. LIBRA uses the current image together with a prior image to model multi-study context, while CheXAgent serves as a strong single-image baseline. We evaluate both models in the single-report and sequential settings using the same metrics as the main paper: RaTEScore, GREEN, 1/FineRadScore, RadGraph F1, and LunguageScore. Table 1 summarizes the results and extends the original comparison by grouping all models — including Medversa, Cvt2DistilGPT2, RGRG, MAIRA-2 (standard and cascade), MedGemma, and Lingshu — by input configuration (single vs single + prior).
>
> **Table 1.** Scores are reported as averages with 95% confidence intervals.
>
> | input image        | Model              | RaTEScore (single)            | GREEN (single)               | 1/FineRadScore (single)        | RadGraph F1 (single)            | LunguageScore (single)          | LunguageScore (sequential)      |
> |--------------------|--------------------|------------------------------|------------------------------|--------------------------------|----------------------------------|----------------------------------|----------------------------------|
> | **single**         | Cvt2DistilGPT2     | 0.491 (0.46, 0.52)           | 0.240 (0.19, 0.29)           | 0.167 (0.14, 0.20)             | 0.179 (0.15, 0.21)              | 0.367 (0.34, 0.40)              | 0.371 (0.33, 0.41)              |
> |                    | RGRG               | 0.547 (0.53, 0.57)           | 0.266 (0.23, 0.30)           | 0.139 (0.11, 0.17)             | 0.264 (0.23, 0.29)              | 0.406 (0.38, 0.43)              | 0.391 (0.36, 0.42)              |
> |                    | MedGemma-27b-it    | 0.495 (0.48, 0.51)           | 0.149 (0.10, 0.20)           | 0.127 (0.11, 0.14)             | 0.133 (0.12, 0.15)              | 0.318 (0.30, 0.34)              | 0.345 (0.32, 0.37)              |
> |                    | Lingshu            | 0.483 (0.46, 0.50)           | 0.173 (0.13, 0.22)           | 0.141 (0.11, 0.17)             | 0.150 (0.13, 0.18)              | 0.344 (0.32, 0.37)              | 0.356 (0.33, 0.38)              |
> |                    | **CheXAgent (new)**| 0.528 (0.50, 0.55)           | 0.241 (0.20, 0.29)           | 0.131 (0.12, 0.14)             | 0.228 (0.20, 0.26)              | 0.380 (0.35, 0.41)              | 0.388 (0.36, 0.42)              |
> | **single + prior** | Medversa           | 0.543 (0.52, 0.57)           | 0.314 (0.26, 0.37)           | **0.183 (0.15, 0.22)**         | 0.238 (0.21, 0.27)              | 0.409 (0.38, 0.44)              | 0.410 (0.37, 0.45)              |
> |                    | **LIBRA (new)**    | 0.526 (0.50, 0.55)           | 0.266 (0.22, 0.30)           | 0.127 (0.12, 0.14)             | 0.227 (0.20, 0.26)              | 0.414 (0.38, 0.45)              | 0.417 (0.38, 0.43)              |
> |                    | MAIRA-2 (standard) | **0.564 (0.54, 0.59)**       | **0.325 (0.28, 0.37)**       | 0.156 (0.14, 0.18)             | **0.274 (0.25, 0.30)**          | **0.429 (0.40, 0.46)**          | **0.432 (0.41, 0.46)**          |
> |                    | MAIRA-2 (cascade)  | 0.547 (0.53, 0.57)           | 0.299 (0.25, 0.34)           | 0.171 (0.13, 0.21)             | 0.233 (0.21, 0.26)              | 0.419 (0.39, 0.45)              | 0.416 (0.38, 0.45)              |
>
> In the **single-report setting**, CheXAgent and LIBRA achieve competitive performance across existing reference metrics (RaTEScore, GREEN, 1/FineRadScore, RadGraph F1), confirming that they are strong baselines under prior evaluation protocols. LunguageScore further differentiates their behavior at the structured level: for example, LIBRA attains a higher single-report LunguageScore than CheXAgent, indicating better entity- and attribute-level fidelity that is less apparent from existing reference metrics.
>
> In the **sequential setting**, both models were also evaluated under LunguageScore using patient-level trajectories. This allows us to explicitly compare their temporal reasoning capabilities, complementing the single-report analysis. Notably, across both single and sequential LunguageScore, **all models that condition on prior context** (Medversa, LIBRA, MAIRA-2 standard and cascade) outperform **all single-image-only models** (Cvt2DistilGPT2, RGRG, MedGemma, Lingshu, CheXAgent). While existing reference metrics make it difficult to clearly separate these two groups, LunguageScore naturally captures that models leveraging multi-study or historical context produce reports that are more longitudinally consistent and better aligned with the underlying clinical trajectory than those relying on single images alone.

---

> > ### Author Response · Authors · 2025-11-22
> > **[Question1, 2]**
> >
> > **[Weakness 2] Dataset Scale**
> >
> > Please also see our Official Response on “Dataset Scale”.
> >
> > ---
> >
> > **[Question 1]**
> >
> > Thank you for the question. The entity categorization in Section 3.1 was developed through an expert-guided schema design grounded in established radiology reporting standards and ontology-based terminology. We did not adopt a single textbook or ontology verbatim. Instead, we synthesized the categories by referencing the **Fleischner Society Glossary of Terms for Thoracic Imaging (Radiology, 2024)** and **UMLS-based ontologies**, so that the schema remains compatible with commonly used thoracic imaging terminology and interoperable with standard clinical vocabularies.
> >
> > The final taxonomy was refined and validated through multiple review rounds with **four board-certified radiologists**, as detailed in Appendix A.2–A.3. Each expert independently reviewed entity–category assignments and then resolved ambiguous cases through consensus. This process ensured that the categories reflect the diagnostic reasoning hierarchy used in real-world chest X-ray interpretation rather than being an ad hoc design.
> > Regarding potential overlap, our guiding principle is **mutually exclusive single categorization**. Each entity is assigned to the unique category that best represents its primary diagnostic role (for example, “pneumonia” → Clinical Finding; “opacity” → Perceptual Finding). This design avoids double counting and keeps the evaluation well defined. We acknowledge that a small subset of descriptive or temporal expressions can be linguistically ambiguous. In such cases, both during expert annotation of the ground-truth dataset and in our structuring framework, context-dependent disambiguation is applied. For instance, the term “increase” is interpreted as `Worsened` when describing lesion growth, but can map to `Improved` in contexts such as “lung volume increase” during recovery. In all cases, the surrounding clinical context is used to assign a single, definitive category, so that the resulting structure is consistent and unambiguous for evaluation.
> >
> > ---
> >
> > **[Question 2]**
> >
> > Thank you for the question. Yes, our schema explicitly supports both one-to-many (1:N) and many-to-many (N:M) associations across sequential reports. It is important to clarify, however, that these associations always represent the tracking of the **same underlying clinical finding** over time. In other words, they are not loose diagnostic correlations, but longitudinal links for a single entity that persists, evolves, or is re-described across studies.
> >
> > Concretely, a single clinical finding in the Day 10 report can be linked to multiple related mentions in the Day 90 report. For example, an earlier report may describe a lesion as “consolidation,” while a later report refers to it using multiple descriptions such as “opacity in the left lower lung” and “increased density.” Similarly, contextual coreference is handled, such as when an object explicitly named in a prior study (“ET tube”) is later referenced indirectly (“the tube seen previously on the left”). Although the surface wording changes and multiple phrases may appear in the later report, they are all treated as referential variants of the same underlying clinical entity.
> >
> > Our schema represents these associations through a two-level hierarchy defined over the full patient trajectory:
> >
> > - **EntityGroup.** This layer aggregates semantically equivalent mentions of a finding across all reports. All phrases that refer to the same concept (for example, “cardiomegaly” at Day 10 and “enlarged heart” or “increased cardiac silhouette” at Day 90) are assigned to the same EntityGroup.
> >
> > - **TemporalGroup.** This layer links EntityGroups that belong to the same diagnostic episode, based on explicit time intervals (for example, a single pneumonia episode that spans Day 10, Day 30, and Day 90).
> >
> > Many-to-many relationships between reports are represented at the **group level**: multiple mentions in Day 10 and multiple mentions in Day 90 can all be assigned to the same EntityGroup and TemporalGroup. Instead of encoding explicit pairwise edges between every Day 10 and Day 90 mention, the schema uses this shared group membership as an implicit N:M linkage. This representation naturally captures 1:N and N:M associations while preserving the continuity of each clinical finding across the entire sequence.

---

> > > ### Comment · Reviewer_zLzN · 2025-11-24
> > >
> > > Thanks for the responses. I will consider to increase my rating if no further errors are found.

---

> > > > ### Author Response · Authors · 2025-11-25
> > > > **Response to Reviewer Comment**
> > > >
> > > > Thank you again for carefully reading our rebuttal and for considering an updated rating. If any remaining issues or questions arise, please let us know and we would be happy to clarify them during the discussion.

---

### Official Review · Reviewer_u5tg · 2025-11-01

**Soundness:** 3
**Presentation:** 3
**Contribution:** 2
**Rating:** 4
**Confidence:** 4

**Summary:**

This paper presents a benchmark for structured and sequential chest x-ray interpretation. It addresses two major limitations in prior works, i.e., temporal reasoning across longitudinal reports (e.g., "no change" or "worsening" must be validated against prior findings) and
fine-grained clinical accuracy, including attributes like location, size, and morphology, which are often lost in coarse metrics.

**Strengths:**

1. This paper focuses on radiology report evaluation through the extraction of structured knowledge. In addition to single-report evaluation, the framework introduces a sequential evaluation paradigm, which has been largely underexplored in previous research.

2. The paper provides a detailed pipeline for constructing an evaluation benchmark, which can serve as a valuable resource for future research in radiology report analysis.

3. The proposed evaluation framework is comprehensive, covering a broad range of aspects, from lexical to semantic similarity. Moreover, the involvement of domain experts in annotating the structured radiology reports ensures the reliability and clinical validity of the dataset.

**Weaknesses:**

1. Despite the comprehensive evaluation design, the structure extracted from radiology reports is primarily represented as triples in a knowledge graph. However, extensive research has already explored this representation. A higher-level view of structured reports should encompass not only triples but also observations, findings, and clinical statuses. Therefore, entity-level evaluation alone may be insufficient to capture the full complexity of radiology reports. ref: [1]

2. The LUNGUAGESCORE combines both lexical and semantic similarity. However, before applying this metric, one must extract triples from reports using external tools (LLMs in this paper). While these models demonstrate good performance in triple extraction, it might be worth exploring an alternative approach, i.e., using LLM-as-a-judge for computing LUNGUAGESCORE. This approach is simple, effective, and relies only on prompting. Moreover, scoring semantic similarity through embeddings may not fully align with radiologists’ subjective preferences (as shown in Table 2).

3. The structured evaluation closely resembles KG-based evaluation, yet its performance in single-report settings is comparable to that of RadGraph, suggesting limited improvement in this aspect.

4. Regarding temporal evaluation at the semantic level, GREEN can also detect comparisons with prior findings (e.g., “(f) Omitting a comparison detailing a change from a prior study”). According to Table 2, the proposed metric appears to underperform GREEN in this regard.

[1] Automated Structured Radiology Report Generation. https://arxiv.org/pdf/2505.24223

**Questions:**

See Weakness

---

> ### Author Response · Authors · 2025-11-22
> **[Weakness 1, 2] Triples-only KG vs richer clinical structure**
>
> **[Weakness 1] Triples-only KG vs richer clinical structure**
>
> Thank you for this insightful comment. We fully agree that a robust clinical evaluation must transcend simple entity-level triples to capture higher-level observations and clinical statuses. **We clarify that our framework is explicitly designed to operate at this higher, entity-centered level.**
>
> Although we utilize triples as an intermediate serialization format for transparency, they are **not** treated as isolated evaluation units. Instead, our framework logically aggregates all triples referring to the same finding into a single, **consolidated finding representation**—effectively an entity-centered subgraph that encapsulates the observation alongside its full context.
>
> In this model, an entity is not a simple symbolic node but a **structured object** that integrates **18 clinically grounded descriptors**, including morphology, distribution, severity, diagnostic status, and temporal change. For example, the phrase *“moderate loculated right lower lobe pleural effusion worsened, related to pneumonia”* is processed not as a bag of scattered triples, but as a **coherent diagnostic cluster**:
>
> * **Entity:** Pleural Effusion
> *   ㄴ**Attributes:** `{severity: moderate, morphology: loculated, location: right lower lobe, temporal_change(worsen): worsened, evidence: pneumonia}`
>
> Crucially, **LunguageScore operates on this consolidated level.** It assesses the semantic, structural, and temporal alignment of the *entire finding* rather than evaluating isolated edges. This ensures that the metric captures the "full complexity" you emphasized—distinguishing, for instance, between a *worsening* and *resolving* condition as an intrinsic property of the finding itself. Thus, while triples serve as the underlying syntax, our metric fundamentally evaluates these **holistic clinical observations**.
>
> ---
>
> **[Weakness 2] LLM-as-a-judge**
>
> Thank you for the thoughtful suggestion. We agree that LLM-as-a-judge is an appealing alternative: it is simple to deploy, relies only on prompting, and has been adopted in recent metrics such as GREEN and Fine-RadScore. In LunguageScore, however, we intentionally follow a different design philosophy that **decouples structuring from scoring**.
>
> Radiology reports are inherently dense, often entangling multiple findings, attributes, and temporal cues within a single sentence. Asking an LLM to parse this complexity and produce a single holistic judgment in one “black-box” step is prone to oversight and instability, and it offers limited insight into *why* a particular score was assigned. To address this, we restrict the role of the LLM to what it does best: using contextual understanding to map raw reports into a structured schema of entities, attributes, and relations. Once structured, the burden of evaluation is shifted to a **predefined scoring pipeline** that combines lexical, semantic, structural, and temporal components according to a mathematically specified procedure. In this way, the LLM focuses on representation, while the scoring logic remains fixed, inspectable, and repeatable across models and settings.
>
> This modular framework reduces the error modes inherent in generative judgments and, importantly, enables **precise error localization**. Because scores are decomposed across components (for example, entity status, laterality, severity, temporal trend), we can identify exactly which aspect of a prediction caused a discrepancy. This yields traceable, actionable feedback for model development, which is difficult to obtain from a single holistic judge-style score.
>
> Regarding alignment with clinical judgment, we agree that semantic embeddings alone do not fully capture radiologist preferences. LunguageScore therefore does *not* rely on embeddings in isolation. Instead, it combines semantic similarity with **expert-derived attribute weighting** and explicit structural checks. The relative weights on key attributes (such as diagnostic status, laterality, morphology, and temporal change) were determined through consensus among board-certified radiologists, so that clinically critical discrepancies are penalized more heavily than minor lexical or paraphrasing differences. This balances linguistic flexibility (via semantics) with structural rigor (via weighted attributes), while keeping the procedure well defined and reproducible.
>
> Finally, our empirical analyses with ReXVal (Table 2) and ReXErr (Appendix D) show that this structured, decoupled design aligns closely with radiologist assessments and is particularly sensitive to subtle but clinically meaningful errors that purely embedding-based or judge-only baselines often miss. We view LLM-as-a-judge as a valuable complementary baseline, but the goal of LunguageScore is to provide a **transparent, factorized evaluation framework** that can be audited and refined at the level of specific clinical attributes, rather than relying solely on a single holistic judgment.

---

> > ### Author Response · Authors · 2025-11-22
> > **[Weakness 3&4]**
> >
> > Thank you for the careful observations.
> >
> > We see two related concerns here:
> >
> > - (i) in the single-report setting, our structured evaluation may appear similar to prior KG-based approaches while achieving performance comparable to RadGraph, and
> >
> > - (ii) GREEN appears stronger in Table 2.
> >
> > First, regarding the relation to KG-based evaluation, we agree that any structured evaluation of radiology reports will, at some level, involve entities and relations that can be serialized as triples. Our contribution is not to move away from triples altogether, but to **change the evaluation unit** from individual edges to richer, clinically grounded finding objects that support both single-report and sequential analysis.
> >
> > - **Granularity and robustness.** As summarized in Table A.3, when all schemas are re-aligned under a unified NER/RE view, our schema covers a larger set of core entity types, richer entity–entity relations (e.g., location, associate, evidence), and more fine-grained entity–attribute relations (e.g., status, certainty, location, severity, morphology), and it is the only one that explicitly models sequential relations across multiple reports. In contrast, RadGraph evaluates relatively coarse semantic units using a small number of entity categories (about 5 types) and edge-level F1. LunguageScore instead aggregates triples into **consolidated finding objects** that bundle these attributes, and the metric scores these findings jointly at the semantic and attribute level rather than evaluating isolated triples.
> >
> > - **Interpretation of comparable performance.** Achieving performance comparable to or better than RadGraph-based metrics in the ReXVal single-report benchmark, while operating on this more fine-grained schema, suggests that we can introduce a richer, clinically aligned representation **without sacrificing the stability** that makes KG-style metrics attractive. We view this as a positive validation that our structured framework remains reliable in the static setting, while also providing the structural foundation required for more complex patient-level longitudinal evaluation.
> >
> > Second, in the comparison with GREEN, we acknowledge that GREEN achieves slightly higher overall alignment with ReXVal in Table 2. This is expected, since GREEN is explicitly designed to follow the ReXVal error taxonomy and uses an instruction-following LLM that is prompted to act as another radiologist when detecting errors on ReXVal cases.
> >
> > Our design makes a different, deliberate choice about **where** temporal reasoning is validated:
> >
> > **Single-report setting.** In the single-report regime, expressions such as “no change”, “unchanged”, “improved”, or “worsened” are treated as textual attributes of the current finding. LunguageScore evaluates whether the model reproduces these attributes and encodes them correctly in the structured representation, in the same spirit as GREEN’s handling of error category (f), where a penalty is assigned if phrases like “no change” are omitted from the generated report when they are present in the reference. We do not attempt to determine whether these comparison statements are clinically correct in this setting, since the relevant prior or future studies are not part of the single-report input. Instead, we only assess whether the comparison language is faithfully captured here, and reserve the assessment of temporal consistency for the sequential setting where longitudinal context is available.
> >
> > - **Sequential setting.** In the sequential setting, we explicitly evaluate temporal behavior over the full clinical trajectory. LunguageScore uses EntityGroups and TemporalGroups to align findings across time and assess whether the predicted trajectory is consistent with the longitudinal evidence (for example, whether a reported change belongs to the correct diagnostic episode). Temporal evaluation is thus grounded in patient-level sequences rather than only in local comparison wording within a single report. In this regime, LunguageScore can directly compare diagnostic changes in disease status against the structured history derived from previous and subsequent reports, whereas GREEN operates on single reports and does not condition its judgments on actual prior studies. This allows our sequential setting to make more meaningful assessments of temporal consistency than is possible in purely single-report, text-only evaluation.
> >
> > In summary, while GREEN focuses on detecting temporal comparison phrasing in single reports, LunguageScore focuses on a complementary goal: providing a richer, entity-centered structured representation that (i) matches the reliability of KG-style metrics such as RadGraph in the single-report regime, and (ii) is explicitly designed to extend beyond single-report comparisons to **patient-level longitudinal verification**. We will clarify these design choices and their implications for single-report versus sequential evaluation in the revised version.

---

### Official Review · Reviewer_XQC1 · 2025-11-04

**Soundness:** 3
**Presentation:** 3
**Contribution:** 3
**Rating:** 6
**Confidence:** 4

**Summary:**

This paper introduces LUNGUAGE, a fine-grained structured and longitudinal evaluation benchmark tailored for chest X-ray reports. The authors design and implement an automatic structuring framework aligned with the proposed schema, enabling entity and relation extraction and grouping across both single and longitudinal multi-report settings. The framework demonstrates relatively high agreement with human annotations. Furthermore, the paper proposes LUNGUAGESCORE, an interpretable evaluation metric that jointly assesses semantic fidelity, structural consistency, and temporal coherence, thereby establishing a systematic foundation for the evaluation of longitudinal radiology reports generation.

**Strengths:**

- The paper introduces LUNGUAGE, a longitudinal and structured benchmark for chest X-ray report generation. It offers fine-grained report structuring and pays specific attention to the evaluation of longitudinal reports. The dataset has been reviewed by radiologists, enhancing its clinical reliability.

- The authors release their structuring algorithm, prompting templates, and the expert-validated vocabulary, which collectively facilitate standardized, automated, and reproducible extensions of this work.

- The paper introduces LUNGUAGESCORE, integrates semantic, structural, and temporal dimensions into a unified scoring framework. This design enables more effective discrimination among models, especially in terms of their temporal consistency.

**Weaknesses:**

- The overall dataset size in this work is relatively small. Although a key contribution is the focus on longitudinal report sequence evaluation, the number of such longitudinal sequences is limited, which makes the evaluation somewhat insufficient. While the authors acknowledge this limitation in the conclusion, I would strongly encourage them to expand the benchmark with more cases if feasible.

- The study primarily uses the MIMIC-CXR dataset and its subsets, which exhibit a fairly homogeneous reporting style. The evaluation would be more convincing if it were extended to include an additional, distinct dataset with different reporting conventions.

- The evaluation relies on the LLM’s performance. Minor errors in entity extraction, relation identification, or grouping can cascade, since MatchScore = Semantic × (Temporal if T > 1) × Structural, leading to unstable or unreliable results.

- Several concerns regarding specific details, as outlined in **Questions**.

**Questions:**

I fully appreciate the substantial effort behind this work and its significance for longitudinal report evaluation. However, a few details remain unclear to me:

- In single-report evaluation, it is common for reports to already contain longitudinal comparison cues (e.g., “no change,” “unchanged,” etc.). How does the method interpret and evaluate such explicitly comparative attributes when assessing a single report in isolation?
- The semantic similarity metric is somewhat confusing. Are you using the fully structured phrases (e.g., “opacity–left lung–nodular–slightly increased”) directly as input for embeddings, or are you using the original unstructured free-text sentences? If structured phrases, essentially sequences of discrete terms, are embedded directly, is the resulting semantic similarity truly meaningful? In contrast, structural similarity appears more reliable and interpretable in this context.
- In the bipartite matching step, does the approach account for one-to-many alignments? For instance, a single phrase in the reference report might correspond to two or more phrases in the generated report. If such cases arise, how does the method handle them?
- The paper mentions grouping certain entities. Do the authors account for view-specific groupings? Some findings are only observable in specific radiographic views (e.g., PA vs. lateral), and longitudinal sequences may occasionally miss certain views altogether. How does the proposed method handle and evaluate such view-dependent and potentially incomplete scenarios?
- In longitudinal evaluation, some follow-up reports may be overly concise and omit entities previously mentioned (e.g., a finding described in an earlier report is absent in a later one without explicit negation). How does the method treat such omissions? Could this affect the final evaluation scores, especially if the absence is interpreted as a negative finding or simply ignored?

Suggestions for Future Work:

- While the current focus is on textual structuring, it would be a valuable extension to ground each structured phrase with corresponding bounding boxes in the chest X-ray images. This would enable more fine-grained, vision-language alignment and enhance clinical utility.
- Given the benchmark’s small size, its main role is evaluation. It would be more impactful if the authors could leverage their automated structuring pipeline to scale up this dataset for training more robust chest X-ray report generation models. But I also have a concern, if without expert validation, automated expansion may introduce hallucinations. Although the paper notes high consistency when leveraging domain-specific lexicons, medical applications demand high reliability, and unverified synthetic labels could negatively impact downstream model training. It is another challenge that warrants careful consideration.

Overall, I  hope authors will continue to expand and refine this work, which can benefit the communities.

**Details Of Ethics Concerns:**

If the authors release their data and codes, I have no ethics concerns.

---

> ### Author Response · Authors · 2025-11-22
> **[Weakness 2, 3]**
>
> **[Weakness 1] Dataset Scale**
>
> Please also see our Official Response on “Dataset Scale”.
>
> ---
>
> **[Weakness 2] Dataset homogeneity**
>
> We appreciate the reviewer's suggestion regarding dataset diversity. We acknowledge that our experiments are restricted to a single public dataset and that a multi-institutional evaluation would be a valuable extension. However, our exclusive reliance on MIMIC-CXR was a deliberate choice necessitated by the specific requirements of our longitudinal reasoning task.
>
> **1. Scarcity of Longitudinal Public Data.**
> Our study requires tracking patient trajectories over time. As documented in the dataset description, MIMIC-CXR is designed to “preserve the relative chronology of patient information,” linking all chest X-ray studies for patients between 2011 and 2016. To the best of our knowledge, among publicly available chest X-ray datasets with free-text reports, MIMIC-CXR is the only large-scale resource that provides multi-year linked studies with explicit temporal ordering. Most other datasets are effectively cross-sectional and lack the continuous patient timelines required to validate our sequential setting.
>
> **2. Internal stylistic diversity.**
> Although sourced from a single institution, MIMIC-CXR exhibits meaningful internal variability in reporting. According to the dataset documentation, reporting templates are “not enforced by the user interface and can be overridden by the user,” which leads to substantial inter-reporter variability. Furthermore, the documentation notes a “drift over time” in reporting styles due to template amendments over the six-year collection period. This indicates that the dataset contains a wide spectrum of linguistic patterns and is not strictly homogeneous, providing a non-trivial testbed for robustness under different reporting practices within the constraints of publicly available longitudinal data.
>
> ---
>
> **[Weakness 3] LLM pipeline robustness & score stability**
>
> Thank you for raising this concern. We understand that a multiplicative formulation $ (\text{Semantic} \times \text{Temporal} \times \text{Structural}) $ can appear vulnerable to cascading errors. Our design, however, includes several safeguards so that noise in the LLM-based structuring leads to gradual degradation rather than unstable collapse.
>
> **1. Component-wise stability**
>
> Even under structural imperfections (for example, omitted attributes or imperfect grouping), each component is designed to be soft and bounded:
>
> - **Semantic.** Because we embed linearized findings, minor formatting or attribute-placement errors do not automatically zero out the semantic score. As long as the core clinical meaning is preserved, the embedding similarity remains moderately high, so semantic noise reduces the score rather than annihilating it.
>
> - **Temporal.** The temporal term is computed additively as
>   $$0.5 \times \text{SameTimepoint} + 0.5 \times \texttt{TemporalGroup}.$$
>   When a finding is correctly placed in the current study but mis-grouped across time, it still satisfies the SameTimepoint criterion and retains roughly half of the temporal credit instead of dropping to zero.
>
> - **Structural.** Structural similarity is a weighted sum over attributes, not a product. An error in one attribute (for example, severity) only reduces the score proportionally to its weight, while correctly predicted attributes (for example, entity type, location, status) continue to contribute.
>
> As a result, a mistake in one aspect of the structuring reduces the score for that finding but does not fully suppress it.
>
> **2. Localizing errors**
>
> LunguageScore operates at the level of individual findings ( i.e., entity-location-attributes) and then aggregates. An extraction or grouping error affects only the score of the corresponding finding; it does not propagate to unrelated entities within the same report. This prevents a single structuring error from causing a catastrophic failure of the overall score.
>
> **3. Reliable inputs**
>
> Finally, we explicitly minimize the occurrence of structuring errors at the source. As reported in Table 1, our two-stage structuring pipeline achieves high precision and recall against expert annotations, so the majority of inputs to the scoring function are already high-fidelity. The robustness of the scoring design and the quality of the structured inputs together help ensure that LunguageScore behaves stably even in the presence of occasional LLM errors.

---

> > ### Author Response · Authors · 2025-11-22
> > **[Question 1] Longitudinal cues in single reports**
> >
> > Thank you for the question. In the single-report setting ($T=1$), LunguageScore does not perform cross-study temporal alignment due to the absence of neighboring reports (e.g., $T-1$, $T+1$). Consequently, the metric naturally reduces to evaluating Semantic $\times$ Structural similarity.
> >
> > In this context, temporal change expressions such as “no change,” “unchanged,” “improved,” or “worsened” are treated as **intrinsic descriptive attributes** of the finding itself, aligning with established methodologies like RadGraph, GREEN, and RateScore. These comparison cues are integrated into the structured representation similarly to standard attributes like morphology or severity. They are evaluated textually: influencing **Semantic similarity** via the linearized finding representation, and contributing to **Structural similarity** via attribute-level matching. For example, if the reference says “no change” and the prediction says “unchanged,” both phrases are normalized into the same *no-change* temporal slot. Within that slot, we compute a cosine embedding similarity between the two descriptions, then multiply this value by the radiologist-defined weight for the temporal-change attribute; this weighted contribution is added to the Structural score for that finding. In contrast, if the reference says “no change” and the prediction says “improved,” the two phrases populate different temporal slots (the *no-change* column vs. the *improved* column). Since there is no valid pair to compare in either slot, both contribute 0 (0 × attribute weight) to the Structural score, so the total Structural similarity for that finding is reduced by exactly the weights associated with the mismatched attributes. Finally, the Structural similarity is obtained as the normalized sum of these weighted attribute-wise contributions, and in the $T=1$ setting the final match score for a finding is computed as Semantic $\times$ Structural.
> >
> >
> > The distinction in our framework lies in how we validate temporal logic. Radiology reports are often elliptical or concise, frequently omitting findings that are stable or referencing context that exists only in the patient's history. In the single-report setting, we can evaluate whether the model generated the phrase "no change" (attribute extraction), but we cannot verify whether that description is clinically accurate regarding the patient's actual history.
> >
> > Therefore, we reserve true temporal validation for the **Sequential setting**. In this setting, we evaluate the full clinical trajectory by modeling two distinct grouping structures:
> >
> > * **Entity Groups:** We identify whether findings across the sequence represent the same underlying pathology. This grouping is subsequently used for the Semantic evaluation, ensuring that the model is consistently tracking the same disease entity (e.g., linking "opacity" in report A to "consolidation" in report B if they refer to the same lesion).
> >
> > * **Temporal Groups:** We determine whether these grouped entities belong to the same diagnostic episode based on explicit time intervals (days). This grouping verifies the Temporal validity, ensuring that the model correctly recognizes the continuity and progression of the episode over time.
> >
> > Thus, our approach ensures that the single-report metric accurately evaluates the capture of **explicit comparative descriptions as textual features**, while the sequential metric evaluates the **longitudinal clinical dynamics**—verifying both entity consistency (Semantic) and episodic continuity (Temporal) derived from the patient's complete history.

---

> > > ### Author Response · Authors · 2025-11-22
> > > **[Question 2] Structured vs free-text embeddings**
> > >
> > > Thank you for this insightful question.
> > >
> > > To clarify, for semantic similarity, we deliberately embed the linearized structured representation of each finding (for example, “opacity – left lung – nodular – slightly increased”) rather than the raw free-text sentences.
> > >
> > > This choice addresses the inherent complexity of radiology reports, where free-text sentences frequently entangle multiple findings, anatomical locations, and temporal cues within a single clause. For instance, a sentence such as “the right basilar opacity is no longer present, but nodular opacity in the left lung appears slightly increased” mixes distinct clinical concepts. Embedding such sentences directly introduces substantial noise, as encoders tend to emphasize the broader contextual overlap of the entire sentence rather than the specific finding under comparison. Linearization resolves this by isolating the essential clinical elements of each finding into a clean, focused unit.
> > >
> > > Regarding your concern about embedding sequences of discrete terms, modern clinical encoders (for example, MedCPT, BioLORD) are generally effective at handling such keyword-rich inputs. Even without grammatical connectives, self-attention mechanisms aggregate the semantic contribution of the component terms, producing vector representations that yield high similarity for semantically equivalent descriptions (for example, recognizing that “cardiomegaly” and “enlarged cardiac silhouette” are synonymous), even in linearized formats.
> > >
> > > We fully agree, however, that structural similarity offers superior reliability for precise attribute verification. Semantic embeddings, while powerful for handling linguistic variability and synonyms, can under-penalize critical distinctions such as laterality (left vs. right). Therefore, LunguageScore adopts a complementary design: semantic similarity captures linguistic variability and paraphrasing at a high level, while structural similarity enforces strict attribute-level accuracy. In practice, the overall similarity is computed by combining these components so that a high semantic score cannot override a structurally incorrect match. By coupling these dimensions, the metric remains flexible enough to accommodate phrasing variations while maintaining the rigor needed to penalize clinically meaningful attribute errors.

---

> > > > ### Author Response · Authors · 2025-11-22
> > > > **[Question 3] One-to-many under bipartite**
> > > >
> > > > Thank you for the question.
> > > >
> > > > Our matching procedure utilizes **one-to-one bipartite assignment** to prevent score inflation, but the scoring formulation is explicitly designed to retain information from **many-to-many relations** (semantic $\times$ temporal $\times$ structural). This ensures that fragmented predictions do not artificially inflate the score, while still receiving proportional partial credit based on their proximity to the ground truth.
> > > >
> > > > Our goal is not to classify findings as simply binary matches, but to **capture how close each predicted finding is to the gold reference**, regardless of whether it is ultimately chosen in the optimal bipartite match.
> > > >
> > > > **1. Continuous Scoring Mechanism**
> > > > To support this, we compute the full pairwise similarity between every predicted finding ($i$) and gold finding ($j$). Each similarity value is the product of semantic, temporal, and structural similarity without binary thresholding:
> > > > $$
> > > > s_{ij} = \text{Semantic}(i,j) \times \text{Temporal}(i,j) \times \text{Structural}(i,j)
> > > > $$
> > > > We then perform one-to-one bipartite assignment over this matrix. Only the optimal pairs are matched to calculate True Positives (TP), but the **residual similarities** of unmatched items still influence the final score:
> > > >
> > > > * **Matched Pairs (Proportional Credit):**
> > > >     Instead of a binary count ($TP=1$), the credit is the similarity score itself ($s_{mn}$). The remaining divergence ($1 - s_{mn}$) is treated as residual error and assigned symmetrically to FP and FN. This reflects that a match indicates the closest pairing, but not necessarily a perfect one.
> > > >     $$TP = \sum_{(m,n) \in \text{matched}} s_{mn}$$
> > > >
> > > > * **Unmatched Items (Soft Penalties):**
> > > >     Unmatched predictions and gold findings are penalized, but **not arbitrarily**. To account for potential many-to-many overlaps (e.g., fragmentation), the penalty is reduced by the *best available similarity* in the matrix, even if that link wasn't chosen for the 1:1 match.
> > > >     $$FP_u = 1 - \max_j s_{uj} \quad (\text{for unmatched prediction } u)$$
> > > >     $$FN_v = 1 - \max_i s_{iv} \quad (\text{for unmatched gold } v)$$
> > > >
> > > > **2. Concrete Example**
> > > > Consider a scenario where the Gold Standard (GT) has 3 findings and the Prediction (Pred) has 5. This illustrates how we handle N:M relationships (fragmentation) without duplicate credit.
> > > >
> > > > **A. Full Similarity Matrix:**
> > > > | | Pred0 | Pred1 | Pred2 | Pred3 | Pred4 |
> > > > | :--- | :---: | :---: | :---: | :---: | :---: |
> > > > | **GT0** | 0.3 | 0.4 | 0.1 | 0.2 | **0.8** |
> > > > | **GT1** | **0.8** | 0.1 | 0.2 | 0.1 | 0.3 |
> > > > | **GT2** | 0.1 | 0.2 | 0.3 | **0.9** | 0.1 |
> > > >
> > > > **B. Bipartite Matching (1:1 Assignment):**
> > > > * $GT_0 \leftrightarrow Pred_4$ ($s=0.8$)
> > > > * $GT_1 \leftrightarrow Pred_0$ ($s=0.8$)
> > > > * $GT_2 \leftrightarrow Pred_3$ ($s=0.9$)
> > > >
> > > > **C. Scoring Breakdown:**
> > > > 1.  **True Positives (TP):** Sum of matched similarities.
> > > >     $$TP = 0.8 + 0.8 + 0.9 = 2.5$$
> > > >
> > > > 2.  **Residual Error from Matches:** Accounts for imperfect matches.
> > > >     $$\text{Residual } FP = \text{Residual } FN = (1-0.8) + (1-0.8) + (1-0.9) = 0.5$$
> > > >
> > > > 3.  **Unmatched Predictions (Pred1, Pred2):**
> > > >     Although `Pred1` and `Pred2` were not matched (to prevent double counting), their penalties are reduced because they capture some semantic overlap (0.4 and 0.3 respectively) with the GT.
> > > >     * $Pred_1$ Penalty: $1 - \max(0.4, 0.1, 0.2) = 0.6$
> > > >     * $Pred_2$ Penalty: $1 - \max(0.1, 0.2, 0.3) = 0.7$
> > > >     $$FP_{\text{unmatched}} = 0.6 + 0.7 = 1.3$$
> > > >
> > > > **D. Final Counts / Score:**
> > > > * **Total TP:** 2.5
> > > > * **Total FP:** $0.5 \text{ (residual)} + 1.3 \text{ (unmatched)} = 1.8$
> > > > * **Total FN:** $0.5 \text{ (residual)} + 0 \text{ (no unmatched GT)} = 0.5$
> > > >
> > > > From these, we obtain:
> > > >
> > > > * **Precision** $= \dfrac{TP}{TP + FP} = \dfrac{2.5}{2.5 + 1.8} \approx 0.58$
> > > > * **Recall** $= \dfrac{TP}{TP + FN} = \dfrac{2.5}{2.5 + 0.5} \approx 0.83$
> > > > * **F1** $= \dfrac{2 \cdot \text{Precision} \cdot \text{Recall}}{\text{Precision} + \text{Recall}} \approx 0.68$
> > > >
> > > > Here, only one predicted fragment aligns with each gold finding, preventing duplication of credit (inflation). Yet, the unmatched fragments (`Pred1`, `Pred2`) receive reduced penalties proportional to their similarity. This formulation allows the metric to treat findings on a **continuous scale**, acknowledging partial correctness and robustness against verbosity or fragmentation without collapsing into binary notions of right or wrong.

---

> > > > > ### Author Response · Authors · 2025-11-22
> > > > > **[Question 4] View-specific grouping & missing views**
> > > > >
> > > > > Thank you for raising this important question regarding view dependencies. While we do not explicitly model view metadata, our EntityGrouping is designed to handle both view-specific variations and incomplete sequences by leveraging the full patient trajectory and prioritizing the underlying clinical essence.
> > > > >
> > > > > In our longitudinal setting, we leverage the fact that many clinically important disease processes (for example, chronic cardiac or pulmonary findings) persist over time, even if radiographic protocols vary or specific views are occasionally missing. EntityGrouping addresses these inconsistencies not by evaluating each visit in isolation, but by analyzing findings across the entire history collectively.
> > > > >
> > > > > For instance, consider a patient trajectory with three sequences where visits 1 and 3 are PA views, while visit 2 is an AP view or lacks a particular projection. Although the description of “cardiomegaly” may differ between PA and AP views, our method unifies these mentions based on their semantic similarity within the longitudinal context, so they are treated as a single clinical entity rather than as separate or conflicting findings. When a finding is less observable or omitted in visit 2 due to view limitations, the clear evidence from visits 1 and 3 allows EntityGrouping to maintain a continuous entity across the timeline. We do not interpret such omissions as explicit negatives; hallucinations and contradictions are instead determined by whether the predicted trajectory is inconsistent with the overall longitudinal pattern.
> > > > >
> > > > > Consequently, our metric focuses on evaluating the correctness of the disease trajectory and reduces the impact of surface-level inconsistencies or partial occlusions caused by view dependencies, even without explicit view-level annotations.

---

> > > > > > ### Author Response · Authors · 2025-11-22
> > > > > > **[Question 5] Omissions in follow-ups**
> > > > > >
> > > > > > Thank you for raising this important point regarding reporting bias. We explicitly designed our framework to handle the common clinical scenario where follow-up reports are concise and omit previously established findings to avoid redundancy. In standard single-report evaluations, if a model correctly tracks a persistent finding that the radiologist omits for brevity (e.g., present at $T_1$, omitted at $T_2$), it is typically penalized as a hallucination simply because the finding is textually absent in the reference.
> > > > > >
> > > > > > LunguageScore is designed to resolve this dilemma by evaluating the **longitudinal diagnostic episode** rather than just the isolated text snapshot. Even if the reference report at $T_2$ omits the finding, our Entity Grouping mechanism recognizes that the model’s prediction refers to the same pathology established in the patient’s history (at $T_1$).
> > > > > >
> > > > > > Crucially, the **Temporal Scoring component** ($0.5 \times \text{Sequence} + 0.5 \times \text{Episode}$) ensures the model still receives partial credit for this clinical accuracy. Since the finding is textually absent in the current reference, the model does not receive credit for the **Sequence Alignment** component (the 0.5 associated with the same timepoint study). However, because the finding is correctly identified as part of a valid, ongoing **Diagnostic Episode**, it satisfies the episode alignment criteria, retaining the remaining 0.5 credit.
> > > > > >
> > > > > > As a result, the evaluation becomes more robust to reporting variability. Instead of penalizing valid clinical tracking as a "hallucination" with a zero score, the metric validates the finding against the patient’s longitudinal history. By distinguishing predictions grounded in prior context from baseless over-generations, LunguageScore aims to provide a fair evaluation that targets actual contradictions while accommodating clinically valid but textually omitted findings.

---

> > > > > > > ### Author Response · Authors · 2025-11-22
> > > > > > > **[Suggestion 1, 2]**
> > > > > > >
> > > > > > > **[Suggestion 1] Phrase-level bounding boxes**
> > > > > > >
> > > > > > > We fully agree that grounding structured phrases in the underlying chest X-ray images would be a highly valuable extension. In this work, we intentionally focused on the textual side: analyzing reports from multiple perspectives, defining a clinically grounded schema, and showing how structured representations enable fine-grained and application-oriented evaluation of report generation and longitudinal reasoning. We view this as one half of the broader vision-language problem.
> > > > > > >
> > > > > > > Importantly, our entity schema is already designed to distinguish which parts of the report are plausibly image-groundable and which are not. We separate **Chest X-ray Findings**, which can be directly visualized or inferred primarily from the image, from **Non Chest X-ray Findings**, which require external clinical context:
> > > > > > >
> > > > > > > - Within **Chest X-ray Findings**:
> > > > > > >   - **PF (Perceptual Findings)** capture visual features that are explicitly visible on the image (for example, “opacity”, “pleural effusion”, “pneumothorax”),
> > > > > > >   - **CF (Contextual Findings)** represent diagnoses that rely on interpreting visual findings with limited contextual knowledge (for example, “pneumonia”, “congestive heart failure”), and
> > > > > > >   - **OTH (Other Objects)** describe non-anatomic elements such as tubes, lines, and hardware (for example, “endotracheal tube”, “central venous catheter”).
> > > > > > >   These categories are natural candidates for spatial grounding, especially PF and OTH, and in many cases CF where the diagnosis is tightly coupled to specific image findings.
> > > > > > >
> > > > > > > - In contrast, **Non Chest X-ray Findings**, including **COF (Clinical Objective Findings)** such as lab values and vitals, **NCD (Non-CXR Diagnosis)** originating from other modalities (for example, CT or MRI diagnoses), and **PATIENT INFO** such as symptoms and history, cannot be determined from the image alone and are not appropriate targets for bounding-box annotation.
> > > > > > >
> > > > > > > This schema therefore provides a principled blueprint for future image grounding: we can link PF and OTH entities (and selected CF entities) to localized image regions via bounding boxes or segmentation masks, while keeping COF, NCD, and PATIENT INFO as text-only contextual signals. Because MIMIC-CXR is linkable to per-patient EHR in MIMIC-IV, this grounded representation can be further extended to incorporate structured clinical trajectories, enabling multimodal alignment between image, report, and EHR. Overall, we see Lunguage as an initial but reliable core: a clinically grounded representation that can be directly extended with image grounding and EHR integration, providing a stable foundation for the next generation of trustworthy vision–language models in medical AI.
> > > > > > >
> > > > > > >
> > > > > > > ---
> > > > > > >
> > > > > > > **[Suggestion 2] Auto-structuring for training; expert validation**
> > > > > > >
> > > > > > > Thank you for this suggestion and for highlighting the trade-off between scale and reliability. In the present work, we intentionally positioned Lunguage primarily as a **gold-standard evaluation benchmark**, with expert-validated annotations that can be trusted for fine-grained analysis of model behavior. That said, we fully agree that our structuring framework naturally lends itself to constructing larger **silver-standard datasets** for training more robust chest X-ray report generation models.
> > > > > > >
> > > > > > > Concretely, the same schema and two-stage structuring pipeline can be applied to the broader MIMIC-CXR corpus to generate large-scale structured labels that are aligned with the Lunguage representation. In this setting, the gold annotations in Lunguage can serve as a calibration and evaluation set: they provide a reference distribution to tune structuring prompts, thresholds, and post-processing rules, and to estimate error rates in different entity and attribute types.
> > > > > > >
> > > > > > > We fully share the reviewer’s concern that automated expansion without expert validation may introduce hallucinations or systematic biases, which would be problematic if treated as ground truth. Our view is that such silver-standard labels should be used as **auxiliary training signals** rather than as primary supervision for safety-critical components, for example as a pretraining target before fine-tuning on smaller expert-validated subsets.

---

> ### Comment · Reviewer_XQC1 · 2025-11-24
> **Final Decision**
>
> I appreciate the authors’ thorough responses. After careful consideration, I’ve decided to keep my original score.

---

> > ### Author Response · Authors · 2025-11-24
> > **Response to Reviewer Comments**
> >
> > Thank you again for carefully reading our rebuttal. We believe we have addressed all of your concerns, especially the first, by annotating 20 additional patients. We would be very grateful if you could consider updating your score. If there are remaining concerns that prevent this, please let us know so we can address them as much as possible during the discussion period.

---

### Official Review · Reviewer_mMcS · 2025-11-04

**Soundness:** 3
**Presentation:** 3
**Contribution:** 3
**Rating:** 6
**Confidence:** 4

**Summary:**

This paper presents LUNGUAGE, a benchmark for structured and longitudinal radiology report generation. It includes 1,473 expert-annotated chest X-ray reports, with 80 capturing disease progression over time. The authors propose a two-stage framework for converting reports into schema-aligned representations and introduce LUNGUAGESCORE, a metric that evaluates entity, relation, and attribute consistency across time. This work enables fine-grained and temporally-aware assessment of radiology report generation.

**Strengths:**

* The paper introduces LUNGUAGE, the benchmark specifically designed for both single-report and longitudinal structured radiology report evaluation. It addresses a key gap in temporal and schema-aligned medical language evaluation.
* The proposed LUNGUAGESCORE enables structured, interpretable, and temporally aware evaluation of generated reports, addressing the limitations of coarse or purely lexical metrics.

**Weaknesses:**

I have previously reviewed an earlier version of this work, and I acknowledge that the authors have addressed several prior concerns, particularly in terms of annotation quality and clinical relevance. However, one major concern still remains:

While the annotation quality and clinical depth are commendable, the overall dataset scale—particularly the limited number of annotated reports and longitudinal cases—remains insufficient to support the benchmark’s claimed generalizability and broader impact. The small number of patients and annotated reports restricts the diversity of pathologies, temporal patterns, and linguistic variations, making it difficult to serve as a comprehensive resource for training or evaluating robust, general-purpose longitudinal models.

Given that this benchmark is positioned as a potential standard for evaluating structured report generation and temporal consistency, I would have expected either significantly broader coverage or stronger justification that the current scope is sufficient.

I would also be interested in seeing how other reviewers weigh the dataset’s scale versus its clinical design, to better understand the consensus on its impact and readiness for adoption.

**Questions:**

N/A

---

> ### Author Response · Authors · 2025-11-22
> **Dataset Scale**
>
> [Weakness] Dataset Scale
>
> Please also see our Official Response on “Dataset Scale”.

---

### Author Response · Authors · 2025-11-22
**Addressing Comments from Reviewers mMcS, XQC1, and zLzN on Dataset Scale**

We thank Reviewers mMcS, XQC1, and zLzN for their feedback on dataset scale, particularly the limited number of longitudinal cases in the initial submission. While we agree that dataset size is an important consideration, our primary goal in this work is not to build the largest possible dataset, but to develop a fine-grained evaluation framework for single and longitudinal chest X-ray interpretation. The benchmark is therefore designed first and foremost as a **high-quality evaluation dataset** for structured report generation and temporal consistency, rather than as a **large-scale training corpus**. Because our metric operates at entity- and attribute-level resolution over full patient trajectories, it requires exhaustively structured reports with dense expert supervision, which is challenging to extend to hundreds of thousands of studies by manual annotation alone. Within these practical constraints, we chose to construct a dense, expert-verified core and expand it as far as is feasible.

During the rebuttal period, we therefore conducted and completed a **second annotation phase** focused on the longitudinal setting. All board-certified radiologists who participated in the initial longitudinal annotation intensively annotated **20 additional patients** with multi-study trajectories, with cases allocated so that each radiologist was responsible for 4 patients. This expansion **triples the number of patients in the longitudinal benchmark (from 10 to 30 patients)**, increasing longitudinal reports from **80 to 186** and temporally aligned entity pairs from about 40,000 to over 90,000. The expanded set has been annotated using the same protocol as in the initial version and is undergoing final consistency checks and integration into the benchmark; the updated longitudinal statistics will be reflected in the revised manuscript.

Overall, the final benchmark is dense at both the report and episode level. The 1,473 single reports contain over 17,000 structured entities and 23,000 attribute–relation pairs, while the 30 longitudinal patients contribute more than 90,000 temporally aligned entity pairs obtained by aligning all possible entities across time. Despite the modest number of patients, this dense structure provides a fine-grained, clinically interpretable testbed for evaluating semantic and temporal consistency in longitudinal report generation.

Finally, the benchmark is designed with extensibility and broader community use in mind. We will **publicly release** the vocabulary, structured reports, annotation prompts, and implementation code, enabling others to adopt and expand our methodology. The structured and modular nature of the framework allows future scaling using large language models, while preserving the clarity and precision of the expert-annotated core.

---

### Author Response · Authors · 2025-11-26
**Response to All Reviewers**

We sincerely appreciate the time and effort you devoted to reviewing our paper. Your constructive feedback has been invaluable, and we have made substantial revisions and clarifications to further strengthen the paper.

The reviewers consistently highlight several strengths of our work:

- **(i) Novelty and Significance:** The introduction of LUNGUAGE as a benchmark specifically tailored for fine-grained report structuring and longitudinal radiology report evaluation, addressing a critical gap in temporal and schema-aligned medical language research **(All reviewers: mMcS, XQC1, u5tg, zLzN)**.

- **(ii) Methodological Rigor:** The proposed LUNGUAGESCORE, which effectively integrates semantic, structural, and temporal dimensions into a unified metric, providing a comprehensive framework covering aspects from lexical to semantic similarity **(Reviewers mMcS, XQC1, u5tg)**.

- **(iii) Data Quality:** The rigorous construction of the dataset involving board-certified radiologists, ensuring reliability and clinical validity of the dataset **(Reviewers mMcS, XQC1, u5tg)** and the description of the construction details **(Reviewer zLzN)**.

- **(iv) Potential Impact:** The broad applicability of the benchmark and the automated framework for future research, serving as a valuable resource to the community **(Reviewers XQC1, u5tg)**.


At the same time, we have carefully responded to the reviewers’ concerns by expanding the dataset, adding new baselines, and providing point-by-point clarifications. We summarize these revisions below:

- **1. Expansion of the Longitudinal Benchmark (Reviewers mMcS, XQC1, zLzN).**

  To address concerns regarding dataset scale, especially for the longitudinal setting, we completed a second annotation phase using the same rigorous quality-control protocol as the initial dataset construction. This expansion triples the longitudinal benchmark from **10 to 30 patients**, yielding **186 reports and 95,404 pairwise observation comparisons**. We have updated the **Introduction** and **Section 3.2** to reflect these changes, re-conducted all sequential setting experiments, and updated **Table 1**. Additionally, we revised **Appendix A.3.2 (Sequential Report Annotation Details)** to describe the construction process and **Figure A.1** to visualize the expanded dataset statistics. Furthermore, to provide a deeper understanding into the structuring logic, we added a qualitative analysis of sequential structuring results in **Appendix B.4**.


- **2. Additional SOTA Baselines (Reviewer zLzN).**

  We incorporated two recent state-of-the-art models, **CheXAgent and LIBRA**, and evaluated them across all metrics (RaTEScore, GREEN, RadGraph, FineRadScore, RadGraph F1) and LUNGUAGESCORE, in both single-report and sequential settings. As shown in Table 3, LUNGUAGESCORE clearly separates single-image models (e.g., CheXAgent) from context-aware models that leverage prior images or history (e.g., MAIRA-2, LIBRA). While single-image models remain competitive in the single-report setting, they consistently lag behind in the sequential setting, underscoring the importance of our sequential evaluation framework for assessing longitudinal consistency.

- **3. Clarification on Metric (Reviewers u5tg, XQC1).**

  To address concerns regarding metric stability and design choices, we have clarified the stability of LUNGUAGESCORE against cascading errors **(Reviewer XQC1)** and justified our structured design over "LLM-as-a-judge" approaches **(Reviewer u5tg)**. We emphasize that our framework operates at an integrated **"entity-centered subgraph level"**, aggregating 18 clinically grounded descriptors, rather than isolated knowledge graph triples, thereby preserving the holistic clinical context. Furthermore, we detailed the **"soft penalty"** mechanism within our bipartite matching process, which ensures that minor structuring noise results in gradual score degradation rather than instability.

- **4. Refined Conclusion and Expanded Limitations.**

  We have revised the **Conclusion** to provide a clearer and more comprehensive summary of the key contributions offered by the LUNGUAGE benchmark, the automated framework, and LUNGUAGESCORE. Additionally, **inspired by the insightful suggestions from Reviewer XQC1 regarding future directions**, we have expanded and structured the discussion of limitations into a dedicated **Appendix F (Limitations and Future Directions)**. This section outlines specific avenues for future research, including scaling to multi-institutional datasets, generating "silver-standard" structured reports, integrating multimodal EHR data (e.g., labs, vitals) for patient-centered reporting, and grounding structured entities in pixel space to enhance vision-language alignment.

We hope these revisions and point-by-point responses address the reviewers’ concerns and clarify our work. We remain open to further discussion.

---

### Author Response · Authors · 2025-12-01
**Summary of Reviewer Consensus and Final Status**

**Dear Area Chair,**

Following our detailed "Response to All Reviewers" (posted Nov 26), we provide a brief update on the current status of the review process and the resolution of key concerns to assist in your final decision.

### **1. Reviewer Status & Engagement**

We have addressed the primary shared weakness (Dataset Scale) and are actively engaging with reviewers for final consensus.

**[Group A: Positive Signals & Active Engagement]**

* **Reviewer zLzN (4 $\rightarrow$ Considering Score Increase):**
After reviewing our new SOTA experiments and tripled dataset, they explicitly commented on Nov 24: **“Thanks for the responses. I will consider to increase my rating if no further errors are found.”** This indicates that their concerns regarding baselines and scale have been substantively resolved.

* **Reviewer XQC1 (6 $\rightarrow$ Score Increase Requested):**
In their final comment, they explicitly acknowledged the **“thoroughness of our responses.”** While they initially decided to keep the original score, we have subsequently **requested a score update**, as we believe their primary concern regarding dataset scale has been fully addressed by our dataset expansion. We remain hopeful for a positive outcome given their acknowledgment of our effort.

**[Group B: Concerns Addressed (Pending Follow-up)]**

* **Reviewer mMcS (6):**
Their review identified **dataset scale as the sole major weakness** in an otherwise positive assessment. Since we have tripled the longitudinal scale (>95k pairs), we believe we have fully resolved their only stated objection.

* **Reviewer u5tg (4):**
They raised concerns that KG triples miss higher-level clinical context and suggested LLM-as-a-judge. We clarified that this stems from a **misunderstanding**; our framework aggregates **triples into entity-centered subgraphs (with 18 clinically grounded attributes) to fully represent clinical context**. We also justified our decoupled design (structuring vs. scoring) for its superior auditability and error localization compared to black-box LLM judging. We believe this detailed response resolves the conceptual concern.

---

### **2. Summary of Key Resolutions**

We briefly map the core concerns to our completed revisions.

| Concern | Resolution (Completed) | Status |
| :--- | :--- | :--- |
| **c1. Dataset Scale** (mMcS, XQC1, zLzN) | **Tripled Longitudinal Benchmark:** Expanded to 30 patients, 186 reports, **95,404 expert comparisons**. | **Resolved** |
| **c2. Baselines** (zLzN) | **New SOTA Models:** Added CheXAgent & LIBRA. Validated metric's discriminative power. | **Resolved** (Ack. by zLzN) |
| **c3. Robustness** (XQC1) | **Soft Penalty Design:** Clarified finding-level error localization to prevent score collapse. | **Addressed** (Ack. by XQC1) |
| **c4. Structure** (u5tg) | **Entity Subgraphs:** Clarified use of rich attributes (18 types) vs. simple triples. | **Clarified** |

---

### **3. Conclusion**

The rebuttal process has resulted in a **tripled dataset** and **empirical validation with new SOTA models**, directly addressing the empirical concerns of Reviewers mMcS, zLzN, and XQC1. With Reviewer zLzN signaling a potential score increase and our active request for reconsideration to Reviewer XQC1, we hope the paper is ready for acceptance.


Best regards,
The Authors

---

### Note · Authors · 2026-01-29

I have read and agree with the venue's withdrawal policy on behalf of myself and my co-authors.

---

### Meta-Review · Area_Chair_FNuu · 2026-01-03

**Summary:**

This paper proposes an evaluation benchmark for radiology report generation that aims to address key limitations of existing approaches, including the focus on single-report settings, coarse-grained clinical semantics, and the lack of consideration of temporal dependencies in longitudinal reports. The benchmark dataset is carefully annotated by medical professionals following a well-designed protocol and contains 1,473 annotated chest X-ray reports from 230 patients. In addition, the work introduces a structuring framework that extracts clinically relevant information from free-text reports to enable automatic benchmarking. A composite metric is further proposed to assess the quality of generated reports with respect to reference reports, taking into account semantic accuracy, structural fidelity, and temporal coherence. The benchmark annotation process is well documented, and experiments conducted on the dataset demonstrate the functionality of the proposed framework and metric.

The reviewers identify several notable strengths of the work, including its focus on a key gap in medical language evaluation; its ability to move beyond coarse or purely lexical metrics; the development of a fine-grained benchmark that supports longitudinal reports with enhanced clinical reliability; the integration of multiple evaluation dimensions into a unified scoring framework; its value as a reusable resource for future research; and the overall comprehensiveness of the proposed evaluation framework.

At the same time, the reviewers raise a number of concerns. The most significant issue—shared by three of the four reviewers—relates to the overall dataset scale, particularly the limited number of annotated reports for longitudinal cases (30 patients and 186 reports after expansion), despite longitudinal evaluation being a central motivation of the work. This limitation restricts the benchmark’s ability to serve as a comprehensive resource for training or evaluating robust, general-purpose longitudinal models. Additional concerns include the reliance on a single dataset (MIMIC-CXR), which reflects a relatively homogeneous reporting style; the dependence of the evaluation on large language model performance, where minor errors may propagate and affect reliability; the potential insufficiency of entity-level evaluation alone; the need to explore alternative approaches to triple extraction; the close resemblance to existing knowledge-graph-based evaluation methods; underperformance relative to established metrics; and several issues related to specific implementation and experimental details.

The authors make commendable efforts to address the reviewers’ concerns, and the response provides additional annotations, experiments, clarifications, and justifications that resolve many of the raised issues, particularly those related to implementation details. The substantial effort devoted to fine-grained annotation represents a valuable contribution of this work. However, concerns regarding the limited data scale—and thus the representativeness and diversity of the benchmark—remain. Moreover, the consistency of LUNGUAGESCORE with expert judgment is only concretely evaluated for the single-report setting via ReXVal, even though longitudinal evaluation is a key motivation of the work, and this consistency is weaker than that of some existing metrics such as GREEN and 1/FineRadScore. While the design of the annotation protocol and the composite metric is thoughtful, it remains largely heuristic. In addition, the LLM-based structuring framework is primarily an engineering solution and offers limited technical novelty.

Taking all factors into consideration, the Area Chair believes that the two reviewers (mMcS and XQC1) who initially assigned scores of 6 are unlikely to increase their scores, while Reviewers u5tg and zLzN, who initially gave scores of 4, may increase their scores to 6.

In summary, while the work demonstrates clear merits and addresses an important problem, it also exhibits notable limitations, and its current contributions are insufficient for acceptance at this highly competitive conference. Therefore, the Area Chair cannot recommend acceptance of the paper in its present form. It is hoped that the reviewers’ comments will help further strengthen this promising line of work in future submissions.

**Reviewer Concerns:**

The authors make commendable efforts to address the reviewers’ concerns, and the response provides additional annotations, experiments, clarifications, and justifications that resolve many of the raised issues, particularly those related to implementation details. The substantial effort devoted to fine-grained annotation represents a valuable contribution of this work. However, concerns regarding the limited data scale—and thus the representativeness and diversity of the benchmark—remain. Moreover, the consistency of LUNGUAGESCORE with expert judgment is only concretely evaluated for the single-report setting via ReXVal, even though longitudinal evaluation is a key motivation of the work, and this consistency is weaker than that of some existing metrics such as GREEN and 1/FineRadScore. While the design of the annotation protocol and the composite metric is thoughtful, it remains largely heuristic. In addition, the LLM-based structuring framework is primarily an engineering solution and offers limited technical novelty.

**Reviewer Scores:**

Taking all factors into consideration, the Area Chair believes that the two reviewers (mMcS and XQC1) who initially assigned scores of 6 are unlikely to increase their scores, while Reviewers u5tg and zLzN, who initially gave scores of 4, may increase their scores to 6.

---

### Decision · Program_Chairs · 2026-01-26

Reject